# On the SDEs and Scaling Rules for Adaptive Gradient Algorithms

**Sadhika Malladi**[*]    **Kaifeng Lyu**[*]    **Abhishek Panigrahi**    **Sanjeev Arora**
Department of Computer Science
Princeton University
{smalladi,klyu,ap34,arora}@cs.princeton.edu

## Abstract

Approximating Stochastic Gradient Descent (SGD) as a Stochastic Differential Equation (SDE) has allowed researchers to enjoy the benefits of studying a continuous optimization trajectory while carefully preserving the stochasticity of SGD. Analogous study of adaptive gradient methods, such as RMSprop and Adam, has been challenging because there were no rigorously proven SDE approximations for these methods. This paper derives the SDE approximations for RMSprop and Adam, giving theoretical guarantees of their correctness as well as experimental validation of their applicability to common large-scaling vision and language settings. A key practical result is the derivation of a *square root scaling rule* to adjust the optimization hyperparameters of RMSprop and Adam when changing batch size, and its empirical validation in deep learning settings.

## 1 Introduction

Distributed synchronous optimization environments have enabled very rapid training of models (in terms of wall-clock time) by allowing a large batch size. Understanding large-batch stochastic optimization is crucial to enjoying the speed-up of this setting. In this context, Krizhevsky (2014); Goyal et al. (2017) empirically discovered the *linear scaling rule (LSR)* for Stochastic Gradient Descent (SGD). It calls for scaling learning rate proportionately to the batch size while fixing the number of epochs. It was recognized that the validity of this scaling rule stems from the benefits to generalization due to noise from mini-batch gradient estimation. But naive analysis, as done in Hoffer et al. (2017), suggested that the scaling rule for SGD ought to be *square root* instead of linear. Subsequently, Jastrzębski et al. (2017) pointed out that since the phenomenon involves varying the LR even down to zero, the correct analysis should invoke a continuous view, namely a stochastic differential equation (SDE). The SDE view helps identify the correct scaling of the noise and leads to a derivation of the linear scaling rule (see Section 2.2).

However, extending the SDE approach—i.e., continuous-time approximations—to popular adaptive optimization algorithms, like RMSprop (Tieleman and Hinton, 2012) and Adam (Kingma and Ba, 2015), has been challenging due to their use of coordinate-wise normalization. By ignoring gradient noise, Ma et al. (2022) derived intuitive continuous approximations for full-batch RMSprop and Adam; however, this deterministic view precludes a scaling rule.

Recent papers have suggested a *square root* scaling rule for adaptive algorithms: set the learning rate proportionately to the *square root* of the batch size while fixing the number of epochs. Based on perturbation theory, Granziol et al. (2022) proposed the square root scaling rule for RMSprop and Adam but could only reason about optimization behavior near convergence, not along the entire trajectory. A square root scaling rule was also empirically discovered for another adaptive gradient

---

[*]Equal Contribution.

36th Conference on Neural Information Processing Systems (NeurIPS 2022).

method called *LAMB* (You et al., 2020), which is an optimization method with layerwise adaptive learning rates, designed for better optimization and generalization in large-batch training. Instead of tuning learning rates while increasing batch size, LAMB used the square root scaling rule to automatically adjust the learning rate and achieve strong performance on vision and language tasks.

In this paper, we make the following contributions.

1. We propose new SDE approximations for two popular adaptive optimization algorithms, RMSprop and Adam (Definitions 4.1 and 4.4) in Theorems 4.2 and 4.5. We prove that these approximations are *1st-order weak approximations* (Definition 2.4), providing a calculus-based guarantee of the approximation strength between the SDEs and their corresponding discrete processes as was done for SGD and its SDE in Li et al. (2019).

2. Our SDE approximations immediately yield square-root scaling rules (Definitions 5.1 and 5.2) for adjusting the optimization hyperparameters of Adam and RMSprop when changing batch size to ensure that the resulting discrete trajectories are all 1st-order weak approximations of the same SDE (Theorem 5.3). Experiments (Figures 1 and 2 and Appendix I) validate the scaling rules in the vision and language modeling domains.

3. We provide efficient experimental verification of the validity of the SDE approximation for the adaptive algorithms in realistic deep nets (Definitions 5.1 and 5.2). Direct simulation of the SDE, e.g., Euler-Maruyama, is prohibitively expensive because it requires computing the full gradient and noise covariance at fine-grained intervals. Hence we adapt (Definition 6.2) the new and efficient *SVAG simulation* for SGD (Li et al., 2021) for use with our proposed SDEs and rigorously prove its correctness (Theorem 6.3). Using SVAG, we provide evidence that the proposed SDE approximations track the analogous discrete trajectories in many common large-scale vision and language settings (Figure 3 and Appendix H).

## 2 Preliminaries

We use $\boldsymbol{v} \odot \boldsymbol{u}$, $\boldsymbol{v}^2$, $\sqrt{\boldsymbol{v}}$ to denote coordinate-wise operators for multiplication, taking squares, taking square roots. For ease of exposition we modify RMSprop and Adam to use $\boldsymbol{v}_k$ in the update for $\boldsymbol{\theta}_k$ instead of using $\boldsymbol{v}_{k+1}$.[2] We also assume that $\boldsymbol{v}_0$ is nonzero if $\epsilon$ is 0 to avoid division by zero.

**Definition 2.1.** RMSprop (Tieleman and Hinton, 2012) is an algorithm that updates $\boldsymbol{\theta}_k$ as follows,

$$\boldsymbol{\theta}_{k+1} = \boldsymbol{\theta}_k - \eta \boldsymbol{g}_k \odot (\sqrt{\boldsymbol{v}_k} + \epsilon)^{-1}, \quad \boldsymbol{v}_{k+1} = \beta \boldsymbol{v}_k + (1-\beta)\boldsymbol{g}_k^2,$$

where $\boldsymbol{\theta}_k$ is the parameter, $\boldsymbol{g}_k$ is the stochastic gradient at step $k$, and $\boldsymbol{v}_k$ is an estimate for the second moment of $\boldsymbol{g}_k$.

**Definition 2.2.** Adam (Kingma and Ba, 2015) is an algorithm that updates $\boldsymbol{\theta}_k$ as follows,

$$\boldsymbol{m}_{k+1} = \beta_1 \boldsymbol{m}_k + (1-\beta_1)\boldsymbol{g}_k, \qquad\qquad \boldsymbol{v}_{k+1} = \beta_2 \boldsymbol{v}_k + (1-\beta_2)\boldsymbol{g}_k^2,$$
$$\widehat{\boldsymbol{m}}_{k+1} = \boldsymbol{m}_{k+1} \odot (1-\beta_1^{k+1})^{-1}, \qquad\qquad \widehat{\boldsymbol{v}}_{k+1} = \boldsymbol{v}_{k+1} \odot (1-\beta_2^{k+1})^{-1},$$
$$\boldsymbol{\theta}_{k+1} = \boldsymbol{\theta}_k - \eta \widehat{\boldsymbol{m}}_{k+1} \odot (\sqrt{\widehat{\boldsymbol{v}}_k} + \epsilon)^{-1},$$

where $\boldsymbol{\theta}_k$ is the parameter, $\boldsymbol{g}_k$ is the stochastic gradient at step $k$, $\boldsymbol{m}_k$ is the momentum, and $\boldsymbol{v}_k$ is an estimate for the second moment of $\boldsymbol{g}_k$.

### 2.1 Noisy Gradient Oracle with Scale Parameter

We abstract the stochastic gradient as being provided by a *noisy* oracle for the full gradient. We formulate the oracle to highlight the phenomenon of interest: changing the batch size affects the scale of the noise.

**Definition 2.3.** A *Noisy Gradient Oracle with Scale Parameter* (NGOS) is characterized by a tuple $\mathcal{G}_\sigma = (f, \boldsymbol{\Sigma}, \mathcal{Z}_\sigma)$. Given a (noise) scale parameter $\sigma > 0$, $\mathcal{G}_\sigma$ takes an input $\boldsymbol{\theta}$ and returns $\boldsymbol{g} = \nabla f(\boldsymbol{\theta}) + \sigma \boldsymbol{z}$, where $\nabla f(\boldsymbol{\theta})$ is the gradient of $f$ at $\boldsymbol{\theta}$, $\boldsymbol{z}$ is the gradient noise drawn from the probability distribution $\mathcal{Z}_\sigma(\boldsymbol{\theta})$ with mean zero and covariance matrix $\boldsymbol{\Sigma}(\boldsymbol{\theta})$. We use $\mathcal{G}_\sigma(\boldsymbol{\theta})$ to denote the distribution of $\boldsymbol{g}$ given $\sigma$ and $\boldsymbol{\theta}$. The probability distribution $\mathcal{Z}_\sigma(\boldsymbol{\theta})$ can change with the scale $\sigma$, but the covariance matrix $\boldsymbol{\Sigma}(\boldsymbol{\theta})$ is fixed across different noise scales.

---

[2]Experiments in Appendix G.2 show that this change does not significantly impact performance.

For all $\boldsymbol{g}_k$ in Definitions 2.1 and 2.2, we assume that $\boldsymbol{g}_k$ is drawn from a noisy gradient oracle $\mathcal{G}_\sigma$. In our setting, as is common when batches are sampled with replacement, $\sigma$ is primarily controlled through the batch size; in particular, $\sigma \sim 1/\sqrt{B}$ (see Appendix F.1 for a derivation). For sampling with replacement on a finite dataset of size $n$, where $f_1(\boldsymbol{\theta}), \ldots, f_n(\boldsymbol{\theta})$ are the loss functions for the $n$ data points (and the average of these $n$ functions is $f(\boldsymbol{\theta})$), this covariance matrix for a given parameter $\boldsymbol{\theta}$ can be explicitly written as

$$\boldsymbol{\Sigma}(\boldsymbol{\theta}) = \frac{1}{n} \sum_{i=1}^{n} (\nabla f_i(\boldsymbol{\theta}) - \nabla f(\boldsymbol{\theta}))(\nabla f_i(\boldsymbol{\theta}) - \nabla f(\boldsymbol{\theta}))^\top. \tag{1}$$

## 2.2 SDE Approximation and Scaling Rules

Under appropriate conditions it becomes possible to approximate SGD via an Itô SDE, which uses Brownian motion to model the noise and has the following general form, where $W_t$ is a Wiener process: $\mathrm{d}\boldsymbol{X}_t = \boldsymbol{b}(\boldsymbol{X}_t)\mathrm{d}t + \boldsymbol{\sigma}(\boldsymbol{X}_t)\mathrm{d}W_t$. The SGD update rule for a loss $f$ is $\boldsymbol{x}_{k+1} = \boldsymbol{x}_k - \eta\boldsymbol{g}_k$, where $\eta$ is the learning rate and $\boldsymbol{g}_k$ is given by the NGOS on input $\boldsymbol{x}_k$. The following is the canonical interpretation of SGD as an SDE:

$$\mathrm{d}\boldsymbol{X}_t = -\nabla f(\boldsymbol{X}_t)\mathrm{d}t + \sqrt{\eta}\boldsymbol{\Sigma}^{1/2}(\boldsymbol{X}_t)\mathrm{d}\boldsymbol{W}_t. \tag{2}$$

Equation (2) hints at a relationship between learning rate and gradient noise—specifically, the *linear scaling rule*—since scaling batch size by factor $\kappa$ scales the noise covariance by $1/\kappa$, which can be canceled by scaling $\eta$ by $\kappa$ as well (Jastrzębski et al., 2017). Therefore, the linear scaling rule ensures the SDE approximation does not change when using a different batch size. With the same methodology, the current paper studies the SDE approximations for adaptive gradient algorithms to derive the square root scaling rule for them.

## 2.3 Quality of SDE Approximation and Theoretical Assumptions

The quality of the SDE approximation can be measured empirically (Section 6) and bounded theoretically using a calculus-based guarantee, which was initiated in the context of deep learning in Li et al. (2019). In particular, the theoretical guarantee uses the following notion of approximation between discrete and continuous stochastic processes by comparing iteration $k$ in the discrete process with continuous time $k\eta_\mathrm{e}$, where $\eta_\mathrm{e} > 0$ is the (effective) step size of the discrete process.

**Definition 2.4** (Order-1 Weak Approximation, Li et al. (2019))**.** Let $\{\boldsymbol{X}_t^{\eta_\mathrm{e}} : t \in [0, T]\}$ and $\{\boldsymbol{x}_k^{\eta_\mathrm{e}}\}_{k=0}^{\lfloor T/\eta_\mathrm{e} \rfloor}$ be families of continuous and discrete stochastic processes parametrized by $\eta_\mathrm{e}$. We say $\{\boldsymbol{X}_t^{\eta_\mathrm{e}}\}$ and $\{\boldsymbol{x}_k^{\eta_\mathrm{e}}\}$ are order-1 weak approximations of each other if for every test function $g$ with at most polynomial growth (Definition B.1), there exists a constant $C > 0$ independent of $\eta_\mathrm{e}$ such that

$$\max_{k=0,\ldots,\lfloor T/\eta_\mathrm{e} \rfloor} |\mathbb{E}g(\boldsymbol{x}_k^{\eta_\mathrm{e}}) - \mathbb{E}g(\boldsymbol{X}_{k\eta_\mathrm{e}}^{\eta_\mathrm{e}})| \leq C\eta_\mathrm{e}$$

A function $g \colon \mathbb{R}^d \to \mathbb{R}$ is said to have *polynomial growth* if there exist positive integers $\kappa_1, \kappa_2 > 0$ such that $|g(\boldsymbol{x})| \leq \kappa_1(1 + \|\boldsymbol{x}\|_2^{2\kappa_2})$ for all $\boldsymbol{x} \in \mathbb{R}^d$ (Definition B.1). The above definition measures the strength of the approximation by the closeness of a test function $g$ computed on the iterates of both trajectories. The approximation becomes stronger in this sense as $\eta_\mathrm{e}$ becomes smaller. In the SDE approximation of SGD, $\eta_\mathrm{e} = \eta$ and $k$ steps correspond to continuous time $k\eta$. A key difference between SGD and adaptive algorithms is that $\eta_\mathrm{e} = \eta^2$ for both RMSprop and Adam, which means $k$ steps correspond to continuous time $k\eta^2$. We validate this time-scaling theoretically in Section 4.

Now we formalize the assumptions needed in the theory. Since our analysis framework is based upon calculus, it becomes necessary to assume differentiability conditions on the NGOS (Definition 2.5). Similar differentiability conditions also appear in prior SDE works (Li et al., 2019, 2021), and we note that lately it has become clear that restricting to differentiable losses (via differentiable node activations such as Swish (Ramachandran et al., 2017)) does not hinder good performance.

**Definition 2.5** (Well-behaved NGOS)**.** The loss function $f$ and covariance matrix function $\boldsymbol{\Sigma}$ in a NGOS $\mathcal{G}_\sigma$ are *well-behaved* if they satisfy[3]: (1) $\nabla f(\boldsymbol{\theta})$ is Lipschitz and $\mathcal{C}^\infty$-smooth; (2) The square root of covariance matrix $\boldsymbol{\Sigma}^{1/2}(\boldsymbol{\theta})$ is bounded, Lipschitz, and $\mathcal{C}^\infty$-smooth; and (3) All partial

---

[3]Note: $\mathcal{C}^\infty$-smoothness can be relaxed using the mollification technique from Li et al. (2019).

derivatives of $\nabla f(\boldsymbol{\theta})$ and $\boldsymbol{\Sigma}^{1/2}(\boldsymbol{\theta})$ up to and including the 4-th order have polynomial growth. We also say that the NGOS $\mathcal{G}_\sigma$ is well-behaved if $f$ and $\boldsymbol{\Sigma}$ are well-behaved.

Deriving an SDE approximation also requires conditions on the noise distribution in the NGOS. It is allowed to be non-Gaussian, but not heavy-tailed. We require an upper bound on the third moment of the noise so that the distribution is not very skewed. For other higher order moments, we require $\mathbb{E}_{\boldsymbol{z} \sim \mathcal{Z}_\sigma}[\|\boldsymbol{z}\|_2^p]^{1/p}$, namely the $L^p$-norm of random variable $\|\boldsymbol{z}\|_2$, to grow at most linearly as a function of $\boldsymbol{\theta}$ (which is implied by ensuring an upper bound on all even order moments). We note that the following conditions are standard in prior work using Itô SDEs to study SGD.

**Definition 2.6** (Low Skewness Condition). The NGOS $\mathcal{G}_\sigma$ is said to satisfy the *low skewness condition* if there is a function $K_3(\boldsymbol{\theta})$ of polynomial growth (independent of $\sigma$) such that $|\mathbb{E}_{\boldsymbol{z} \sim \mathcal{Z}_\sigma(\boldsymbol{\theta})}[\boldsymbol{z}^{\otimes 3}]| \leq K_3(\boldsymbol{\theta})/\sigma$ for all $\boldsymbol{\theta} \in \mathbb{R}^d$ and all noise scale parameters $\sigma$.

**Definition 2.7** (Bounded Moments Condition). The NGOS $\mathcal{G}_\sigma$ is said to satisfy the *bounded moments condition* if for all integers $m \geq 1$ and all noise scale parameters $\sigma$, there exists a constant $C_{2m}$ (independent of $\sigma$) such that $\mathbb{E}_{\boldsymbol{z} \sim \mathcal{Z}_\sigma(\boldsymbol{\theta})}[\|\boldsymbol{z}\|_2^{2m}]^{\frac{1}{2m}} \leq C_{2m}(1 + \|\boldsymbol{\theta}\|_2)$ for all $\boldsymbol{\theta} \in \mathbb{R}^d$.

For well-behaved loss $f(\boldsymbol{\theta})$ and covariance $\boldsymbol{\Sigma}(\boldsymbol{\theta})$, the above two conditions are satisfied when $\mathcal{Z}_\sigma$ is the Gaussian distribution with mean zero and covariance $\boldsymbol{\Sigma}(\boldsymbol{\theta})$. That is, the Gaussian NGOS $\boldsymbol{g} \sim \mathcal{N}(\nabla f(\boldsymbol{\theta}), \sigma^2 \boldsymbol{\Sigma}(\boldsymbol{\theta}))$ satisfies the low skewness and bounded moments conditions. The low skewness condition holds because the odd moments of a Gaussian are all zeros, and the bounded moments condition can be verified since $\mathbb{E}_{\boldsymbol{z} \sim \mathcal{N}(\mathbf{0}, \boldsymbol{\Sigma}(\boldsymbol{\theta}))}[\|\boldsymbol{z}\|_2^{2m}]^{\frac{1}{2m}} \leq \mathbb{E}_{\boldsymbol{w} \sim \mathcal{N}(\mathbf{0}, \boldsymbol{I})}[\|\boldsymbol{w}\|_2^{2m}]^{\frac{1}{2m}} \cdot \|\boldsymbol{\Sigma}^{1/2}(\boldsymbol{\theta})\|_2$ and $\boldsymbol{\Sigma}^{1/2}(\boldsymbol{\theta})$ is Lipschitz.

The Gaussian NGOS with $\sigma = \frac{1}{\sqrt{B}}$ can be seen as an approximation of the gradient noise in a mini-batch training with batch size $B$, if $\boldsymbol{\Sigma}(\boldsymbol{\theta})$ is set to match with (1). But this does not directly imply that the gradient noise in mini-batch training satisfies the low skewness and bounded moments conditions, as the noise is not exactly Gaussian. Assuming that the gradient of the loss function $f_i(\boldsymbol{\theta})$ at every data point is Lipschitz, these two conditions can indeed be verified for all batch sizes $B \geq 1$.

### 2.4 Discussion on Heavy-Tailed Gradient Noise

We note that Definitions 2.6 and 2.7 allow some non-Gaussianity in the noise, but $K_3(\boldsymbol{\theta})$ and $C_{2m}$ could be large in practice. In this case, higher order moments of the gradient noise have non-negligible effects on training that the Itô SDE cannot capture. Zhang et al. (2020) and Simsekli et al. (2019) presented experimental evidence that the noise is heavy-tailed. This motivated Zhou et al. (2020) to consider a Lévy SDE approximation (instead of Itô SDE) to study the (worse) generalization behavior of Adam. However, the quality of the Lévy SDE approximation was not formally guaranteed (e.g., in the sense of Definition 2.4), so finding a guaranteed approximation for adaptive optimization algorithms remains an open problem. Moreover, Li et al. (2021); Xie et al. (2021) highlighted issues with the evidence, noting that the measurements used in Simsekli et al. (2019) are intended only for scalar values. When applied to vector valued distributions the measurement can (incorrectly) identify a multidimensional Gaussian distribution as heavy-tailed too (Li et al., 2021). It is in general difficult to estimate the moments of the noise distribution from samples, so the heavy-tailedness of real-world gradient noise remains an open question.

Our empirical results suggest that our assumptions in Definitions 2.6 and 2.7 are not too strong. In Section 6, we efficiently simulate the Itô SDE using an algorithm analogous to SVAG (Li et al., 2021). The simulation closely approximates the test accuracy achieved by the discrete trajectory, suggesting that even if heavy-tailed noise is present during training, it is not crucial for good generalization (Appendix H). We remain interested in exploring the heavy-tailed analogs of our Itô SDEs. However, efficient simulation of such SDEs remains intractable and formal approximation guarantees are difficult to prove, so we are limited in assessing the utility of such approximations. We leave it for future work to investigate if and how heavy-tailed noise plays a role in adaptive optimization.

## 3 Related Work

We defer a full discussion of empirical and theoretical works on adaptive gradient methods to Appendix A.1 and only discuss works relevant to SDEs and scaling rules here.

**Applications of SDE Approximations.** There are applications of the SDE approximation beyond the derivation of a scaling rule. Li et al. (2020) and Kunin et al. (2021) assumed that the loss has some symmetry (i.e., scale invariance) and studied the resulting dynamics. Furthermore, Li et al. (2020) used this property to explain the phenomenon of sudden rising error after LR decay in training. Xie et al. (2021) analyzed why SGD favors flat minima using an SDE-motivated diffusion model.

**Past Work on Square Root Scaling Rule.** As mentioned before, square root scaling was incorrectly believed for a few years to be theoretically justified for SGD. Granziol et al. (2022) decomposed the stochastic Hessian during batch training into a deterministic Hessian and stochastic sampling perturbation and assumes one of the components to be low rank to propose a square root scaling rule. (You et al., 2020) empirically discovered a square root scaling rule for language models trained by LAMB, a layer-wise variant of Adam. Xie et al. (2022) heuristically derived, but did not show an approximation bound for, a second-order SDE for approximating Adam, and they applied the SDE to study the time needed for Adam to escape sharp minima. Xie et al. (2022) did not discuss a scaling rule, though their proposed SDE may admit one. Similarly, Zhou et al. (2020) derived a Lévy SDE for Adam, but no approximation bounds are given in the paper.

## 4 SDEs for Adaptive Algorithms

An SDE approximation operates in continuous time and thus implicitly considers the limit $\eta \to 0$. In adaptive algorithms, the moment averaging parameters $\beta, \beta_1, \beta_2$ and $\eta$ must be taken to limits such that the adaptivity and stochasticity can still be studied. For example, if $\beta, \beta_1, \beta_2$ remain fixed when $\eta \to 0$, then the algorithm computes the moving averages in a very small neighborhood, which averages out the effects of gradient noise and gradient history, causing the flow to turn into deterministic SignGD (Ma et al., 2022). We will need to assume $\beta, \beta_1, \beta_2 \to 1$ as $\eta \to 0$, which implies the impact of the history grows as the learning rate decreases, and thus the adaptive features of these algorithms can still be studied in the continuous approximation (Ma et al., 2022). To keep the stochastic nature of the flow, we require the noise from mini-batching dominate the gradient updates.

### 4.1 Warm-Up: Linear loss

To build intuition for the SDE and the scaling rule, we first study a simplified setting. In particular, consider a linear loss $f(\boldsymbol{\theta}) = \langle \boldsymbol{\theta}, \bar{\boldsymbol{g}} \rangle$ and isotropic noise in the NGOS, namely $\boldsymbol{g}_k \sim \mathcal{N}(\bar{\boldsymbol{g}}, \sigma^2 \boldsymbol{I})$. The RMSprop update $\boldsymbol{v}_{k+1} = \beta \boldsymbol{v}_k + (1-\beta)\boldsymbol{g}_k^2$ can be expanded as $\boldsymbol{v}_k = \beta^k \boldsymbol{v}_0 + (1-\beta)\sum_{j=0}^{k-1}\beta^j \boldsymbol{g}_j^2$. By linearity of expectation,

$$\mathbb{E}[\boldsymbol{v}_k] = \beta^k \boldsymbol{v}_0 + (1-\beta)\sum_{j=0}^{k-1}\beta^j(\bar{\boldsymbol{g}}^2 + \sigma^2 \mathbf{1}) = \beta^k \boldsymbol{v}_0 + (1-\beta^k)(\bar{\boldsymbol{g}}^2 + \sigma^2 \mathbf{1}).$$

This suggests that $\mathbb{E}[\boldsymbol{v}_k]$ is approximately $\bar{\boldsymbol{g}}^2 + \sigma^2 \mathbf{1}$ after a sufficient number of steps. Setting $\boldsymbol{v}_0 = \bar{\boldsymbol{g}}^2 + \sigma^2 \mathbf{1}$, we see that the approximation $\mathbb{E}[\boldsymbol{v}_k] = \bar{\boldsymbol{g}}^2 + \sigma^2 \mathbf{1}$ becomes exact for all $k \geq 0$.

Using the linearity of variance (for independent variables), the standard deviation of each coordinate of $\boldsymbol{v}_k$ is of scale $\mathcal{O}((1-\beta)\sigma^2)$. Moreover, the expectation $\mathbb{E}[\boldsymbol{v}_k]$ is of scale $\mathcal{O}(\sigma^2)$, so we know that $\boldsymbol{v}_k$ is nearly deterministic and concentrates around its expectation $\bar{\boldsymbol{g}}^2 + \sigma^2 \mathbf{1}$ when $\beta$ is close to 1. Therefore, we take the approximation $\boldsymbol{v}_k \approx \bar{\boldsymbol{g}}^2 + \sigma^2 \mathbf{1}$ for all $k \geq 0$. Ignoring $\epsilon$, the RMSprop update rule becomes:

$$\boldsymbol{\theta}_{k+1} \approx \boldsymbol{\theta}_k - \eta \boldsymbol{g}_k \odot (\bar{\boldsymbol{g}}^2 + \sigma^2 \mathbf{1})^{-1/2}. \tag{3}$$

These dynamics depend on the relative magnitudes of $\bar{\boldsymbol{g}}$ and $\sigma$. We show that when $\sigma \ll \|\bar{\boldsymbol{g}}\|$, no SDE approximation can exist in Appendix F.2. Here, we explore the case where $\sigma \gg \|\bar{\boldsymbol{g}}\|$ which implies $\boldsymbol{\theta}_{k+1} \approx \boldsymbol{\theta}_k - \frac{\eta}{\sigma}\boldsymbol{g}_k$. Noting that $\boldsymbol{g}_k \sim \mathcal{N}(\bar{\boldsymbol{g}}, \sigma^2 \boldsymbol{I})$ gives $\boldsymbol{\theta}_{k+1} - \boldsymbol{\theta}_k \sim \mathcal{N}(\frac{\eta}{\sigma}\bar{\boldsymbol{g}}, \eta^2 \boldsymbol{I})$ approximately. Since Gaussian variables are additive, we can take a telescoping sum to obtain the marginal distribution of $\boldsymbol{\theta}_k$: $\boldsymbol{\theta}_k \sim \mathcal{N}\left((k\eta/\sigma)\bar{\boldsymbol{g}}, k\eta^2 \boldsymbol{I}\right)$ approximately.

If an SDE approximation of RMSprop exists, then $\boldsymbol{\theta}_k$ can be closely approximated by a *fixed* random variable from the corresponding stochastic process at a fixed (continuous) time $t$. Thus, to keep the SDE unchanged upon changing the noise scale $\sigma$, the hyperparameters must be adjusted to keep $\frac{k\eta}{\sigma}$ and $k\eta^2$ unchanged, which implies $\eta \sim \frac{1}{\sigma}$ and $k \sim \frac{1}{\eta^2}$. These observations intuitively yield

the square root scaling rule: noting that $\sigma$ changes with mini-batch size $B$ as $\sigma \sim 1/\sqrt{B}$ suggests $\eta \sim \sqrt{B}$, and $k \sim 1/B$.

## 4.2 SDE Approximations for Adaptive Algorithms

Having established intuition in a highly simplified setting for adaptive algorithms, we now return to a more general and realistic setting. We derive the SDEs that are order-1 approximations of the discrete RMSprop and Adam algorithms under Definition 2.5, where the SDE states consist of both the parameters $\boldsymbol{\theta}$ and moment estimates. From the example of Section 4.1, we see that SDE approximations may exist if $\sigma \sim 1/\eta^2$ and $\sigma \gg \|\bar{\boldsymbol{g}}\|$ (see Appendix G.1 for empirical validation of this assumption). In this case, we can prove that $k \sim \eta^2$ is true not only for the simple setting above but also in general. This is a key difference to the SDE for SGD — each step in RMSprop or Adam corresponds to a time interval of $\eta^2$ in SDEs, but each SGD step corresponds to a time interval of $\eta$. In Section 4.1, $\boldsymbol{v} \sim \sigma^2 \sim 1/\eta^2$ grows to infinity as $\eta \to 0$. This also happens in the general setting, so we track $\boldsymbol{u}_k \triangleq \boldsymbol{v}_k/\sigma^2$ (instead of $\boldsymbol{v}_k$ directly) in the SDEs.

**Definition 4.1** (SDE for RMSprop). Let $\sigma_0 \triangleq \sigma\eta$, $\epsilon_0 \triangleq \epsilon\eta$, and $c_2 \triangleq (1-\beta)/\eta^2$. Define the state of the SDE as $\boldsymbol{X}_t = (\boldsymbol{\theta}_t, \boldsymbol{u}_t)$ and the dynamics as

$$d\boldsymbol{\theta}_t = -\boldsymbol{P}_t^{-1}\left(\nabla f(\boldsymbol{\theta}_t)dt + \sigma_0\boldsymbol{\Sigma}^{1/2}(\boldsymbol{\theta}_t)d\boldsymbol{W}_t\right), \qquad d\boldsymbol{u}_t = c_2(\mathrm{diag}(\boldsymbol{\Sigma}(\boldsymbol{\theta}_t)) - \boldsymbol{u}_t)dt$$

where $\boldsymbol{P}_t := \sigma_0\mathrm{diag}(\boldsymbol{u}_t)^{1/2} + \epsilon_0\boldsymbol{I}$ is a preconditioner matrix, and $\boldsymbol{W}_t$ is the Wiener process.

**Theorem 4.2** (Informal version of Theorem C.2). *Let $\boldsymbol{u}_k \triangleq \boldsymbol{v}_k/\sigma^2$ and define the state of the discrete RMSprop trajectory with hyperparameters $\eta, \beta, \epsilon$ (Definition 2.1) as $\boldsymbol{x}_k = (\boldsymbol{\theta}_k, \boldsymbol{u}_k)$. Then, for a well-behaved NGOS (Definition 2.3) satisfying the skewness and bounded moments conditions (Definitions 2.6 and 2.7), the SDE in Definition 4.1 satisfies*

$$\max_{k=0,\ldots,\lfloor T/\eta^2 \rfloor} |\mathbb{E}g(\boldsymbol{x}_k) - \mathbb{E}g(\boldsymbol{X}_{k\eta^2})| \le C\eta^2$$

*where $g$ and $T$ are defined as in Definition 2.4 and the initial condition of the SDE is $\boldsymbol{X}_0 = \boldsymbol{x}_0$.*

*Remark* 4.3. Section 4.1 suggested that the SDE approximation can only exist when $\sigma \gg \|\bar{\boldsymbol{g}}\|$. This condition is reflected by keeping $\sigma_0 = \sigma\eta$ a constant and $C$ depends on $\sigma_0$. When $\eta \to 0$, $\sigma$ scales as $1/\eta$, so $\sigma \gg \|\boldsymbol{g}\|_2$.

We need to find continuous approximations of the normalization constants in Adam (Definition 2.2). As in the RMSprop case, we take $(1-\beta_2)/\eta^2 = c_2$. Then, we can estimate the normalization constant $1 - \beta_2^k$ in continuous time $t = k\eta^2$ as $1 - \beta_2^k = 1 - (1 - c_2\eta^2)^{t/\eta^2} \approx 1 - \exp(-c_2 t)$. We can do the analogous approximation for the other normalization constant $1 - \beta_1^k$ in Adam. Taking $(1-\beta_1)/\eta^2 = c_1$, we can approximate it as $1 - \beta_1^k \approx 1 - \exp(-c_1 t)$. This is a heuristic approach to deal with the normalization constants, but we can indeed justify it in theory.

**Definition 4.4** (Adam SDE). Let $c_1 \triangleq (1-\beta_1)/\eta^2, c_2 \triangleq (1-\beta_2)/\eta^2$ and define $\sigma_0, \epsilon_0$ as in Definition 4.1. Let $\gamma_1(t) \triangleq 1 - \exp(-c_1 t)$ and $\gamma_2(t) \triangleq 1 - \exp(-c_2 t)$. Define the state of the SDE as $\boldsymbol{X}_t = (\boldsymbol{\theta}_t, \boldsymbol{m}_t, \boldsymbol{u}_t)$ and the dynamics as

$$d\boldsymbol{\theta}_t = -\frac{\sqrt{\gamma_2(t)}}{\gamma_1(t)}\boldsymbol{P}_t^{-1}\boldsymbol{m}_t dt, \qquad d\boldsymbol{m}_t = c_1(\nabla f(\boldsymbol{\theta}_t) - \boldsymbol{m}_t)dt + \sigma_0 c_1 \boldsymbol{\Sigma}^{1/2}(\boldsymbol{\theta}_t)dW_t,$$

$$d\boldsymbol{u}_t = c_2(\mathrm{diag}(\boldsymbol{\Sigma}(\boldsymbol{\theta}_t)) - \boldsymbol{u}_t)dt,$$

where $\boldsymbol{P}_t := \sigma_0\mathrm{diag}(\boldsymbol{u}_t)^{1/2} + \epsilon_0\sqrt{\gamma_2(t)}\boldsymbol{I}$ is a preconditioner matrix, $\boldsymbol{W}_t$ is the Wiener process.

Our main theorem for Adam is given below. The initial steps of the discrete Adam trajectory can be discontinuous and noisy because of the normalization constants changing drastically. Hence, we introduce a constant $t_0$ and show that for any $t_0$, we can construct an SDE to be a weak approximation for Adam starting from the $\lceil t_0/\eta^2 \rceil$-th step of Adam.

**Theorem 4.5** (Informal version of Theorem D.2). *Define $\boldsymbol{u}_k = \boldsymbol{v}_k/\sigma^2$ and let $\boldsymbol{x}_k = (\boldsymbol{\theta}_k, \boldsymbol{m}_k, \boldsymbol{u}_k) \in \mathbb{R}^{3d}$ be the state of the discrete Adam trajectory with hyperparameters $\eta, \beta_1, \beta_2, \epsilon$. Then, for a well-behaved NGOS (Definition 2.3) satisfying the skewness and bounded moments conditions (Definitions 2.6 and 2.7) and any $t_0 > 0$, the SDE in Definition 4.4 satisfies*

$$\max_{k=\lceil t_0/\eta^2 \rceil,\ldots,\lfloor T/\eta^2 \rfloor} |\mathbb{E}g(\boldsymbol{x}_k) - \mathbb{E}g(\boldsymbol{X}_{k\eta^2})| \le C\eta^2$$

*where $g$ and $T$ are defined as in Definition 2.4 and the initial condition of the SDE is $\boldsymbol{X}_{t_0} = \boldsymbol{x}_{\lceil t_0/\eta^2 \rceil}$.*

*Proof Sketch.* We provide a proof sketch for our SDE approximations here and defer the technical details to Theorems C.2 and D.2. The proof follows the same steps as Li et al. (2019): first, we compute the approximation error of the continuous trajectory after one step in discrete time. Then, we use the single step error to bound the error over a finite interval of time. The proof extends standard SDE techniques in several ways. The given SDEs do not satisfy the Lipschitzness and smoothness conditions because the denominator can be unbounded. We thus construct an auxiliary SDE with an equivalent distribution to the desired SDE (Theorem C.5) but with better smoothness conditions, and we prove this SDE to be an order-1 weak approximation of the discrete trajectory. Moreover, the SDE coefficients are time-dependent for Adam, unlike the ones for SGD, so we need to extend existing results to cover this case (see Appendix B.1). □

# 5 Square Root Scaling Rule

The SDE approximations in Definitions 4.1 and 4.4 motivate scaling rules for how to adjust the optimization hyperparameters when changing the batch size. In order for $\sigma_0, c_1, c_2$, and $\epsilon_0$ to remain constant, as required by the SDEs, one needs to change $\eta, \beta, \beta_1, \beta_2$, and $\epsilon$ accordingly.

**Definition 5.1** (RMSprop Scaling Rule). When running RMSprop (Definition 2.1) with batch size $B' = \kappa B$, use the hyperparameters $\eta' = \eta\sqrt{\kappa}$, $\beta' = 1 - \kappa(1 - \beta)$, and $\epsilon' = \frac{\epsilon}{\sqrt{\kappa}}$.

**Definition 5.2** (Adam Scaling Rule). When running Adam (Definition 2.2) with batch size $B' = \kappa B$, use the hyperparameters $\eta' = \eta\sqrt{\kappa}$, $\beta_1' = 1 - \kappa(1 - \beta_1)$, $\beta_2' = 1 - \kappa(1 - \beta_2)$, and $\epsilon' = \frac{\epsilon}{\sqrt{\kappa}}$.

**Theorem 5.3** (Validity of the Scaling Rules). *Suppose we have a well-behaved NGOS satisfying the low skewness and bounded moments conditions.*

1. *Let $\boldsymbol{x}_k^{(B)}$ be the discrete RMSprop (Definition 2.1) trajectory with batch size $B$ and hyperparameters $\eta, \beta$, and $\epsilon$. Let $\boldsymbol{x}_k^{(\kappa B)}$ be the trajectory with batch size $\kappa B$ and hyperparameters adjusted according to Definition 5.1. If $\boldsymbol{x}_k^{(B)}$ and $\boldsymbol{x}_k^{(\kappa B)}$ start from the same initial point, then with g and T defined as in Definition 2.4,*

$$\max_{k=0,\ldots,\lfloor T/\eta^2 \rfloor} \left| \mathbb{E}g(\boldsymbol{x}_k^{(B)}) - \mathbb{E}g(\boldsymbol{x}_{\lfloor k/\kappa \rfloor}^{(\kappa B)}) \right| \leq C(1 + \kappa)\eta^2.$$

2. *Fix $t_0 > 0$. Let $\boldsymbol{x}_k^{(B)}$ be the discrete Adam (Definition 2.2) trajectory with batch size $B$ and hyperparameters $\eta, \beta_1, \beta_2$, and $\epsilon$. Let $\boldsymbol{x}_k^{(\kappa B)}$ be the trajectory with batch size $\kappa B$ and hyperparameters adjusted according to Definition 5.2. If $\boldsymbol{x}_{\lceil t_0/\eta^2 \rceil}^{(B)}$ and $\boldsymbol{x}_{\lceil \kappa t_0/\eta^2 \rceil}^{(\kappa B)}$ are equal, then with g and T defined as in Definition 2.4,*

$$\max_{k=\lceil t_0/\eta^2 \rceil,\ldots,\lfloor T/\eta^2 \rfloor} |\mathbb{E}g(\boldsymbol{x}_k^{(B)}) - \mathbb{E}g(\boldsymbol{x}_{\lfloor k/\kappa \rfloor}^{(\kappa B)})| \leq C(1 + \kappa)\eta^2.$$

*Proof.* By the linearity of covariance, scaling the batch size by $\kappa$ only modifies the NGOS by scaling $\sigma$ by $1/\sqrt{\kappa}$. Hence, both scaling rules ensure that $\sigma_0, c_1, c_2$, and $\epsilon_0$ (and thus, the SDEs) are unchanged when the batch size changes. The approximation bounds in Theorems 4.2 and 4.5 are in terms of $\eta^2$, and since $\eta$ is scaled here by $\sqrt{\kappa}$, the same method gives an upper bound $C\kappa\eta^2$. Adding the approximation bounds for $\eta$ and $\sqrt{\kappa}\eta$ together gives $C(1 + \kappa)\eta^2$. □

*Remark 5.4.* The $t_0$ condition on the Adam rule, a holdover from the condition in Theorem 4.5, implies that our theory only directly applies when there is a warm-start phase of $\lceil t_0/\eta^2 \rceil$, where the marginal distribution of the trainable parameters at the end of the phase is the same across different learning rates $\eta$. Regardless, the scaling rules are shown to work in practice even without this phase.

The scaling rules depend on maintaining the same SDE approximation, so the bounded moments and low skewness conditions are sufficient (but not necessary) for this scaling rule to work. Li et al. (2021) provided an analogous discussion for SGD, and they show the scaling rule can hold even if there is heavy-tailed noise. We leave a study of heavy-tailed gradient noise in adaptive algorithms as future work.

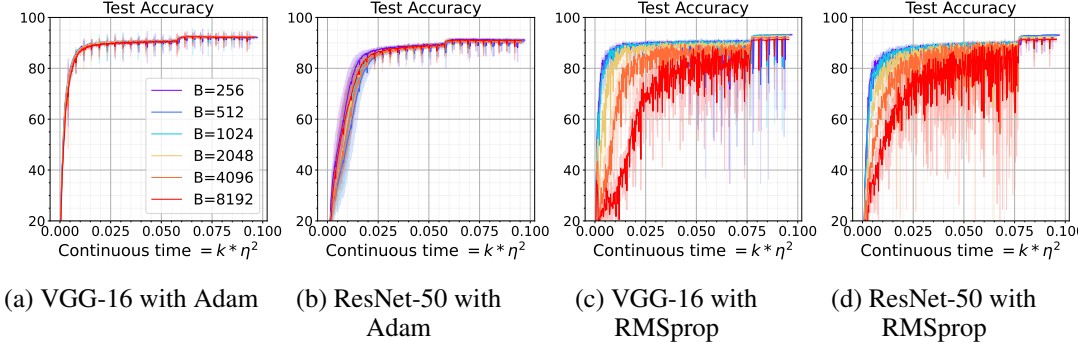

(a) VGG-16 with Adam    (b) ResNet-50 with Adam    (c) VGG-16 with RMSprop    (d) ResNet-50 with RMSprop

Figure 1: Square root scaling rule experiments on CIFAR-10 with VGG-16 and ResNet-50 (details in Appendix J). We plot the mean and variance of 3 random seeds. Same color legend has been used across all the plots. The performance gap between $B = 256$ and $B = 8192$ is at most $3\%$ in all cases.

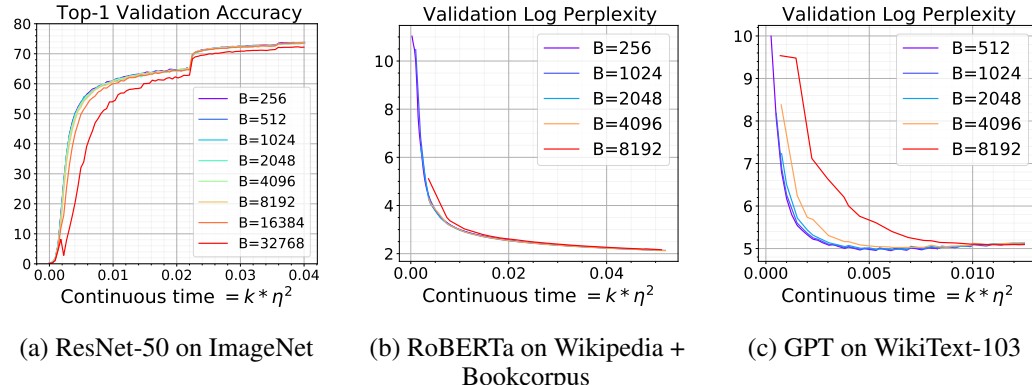

(a) ResNet-50 on ImageNet    (b) RoBERTa on Wikipedia + Bookcorpus    (c) GPT on WikiText-103

Figure 2: Large scale square root scaling rule experiments (details in Appendix J). Small and large batch models differ by at most $1.5\%$ test accuracy in vision and $0.5$ perplexity in language.

**Experiments.** Figures 1 and 2 show the square root scaling rule applied to ResNet-50 (He et al., 2016) and VGG-16 (Simonyan and Zisserman, 2014) trained on CIFAR-10 (Krizhevsky et al.), RoBERTa-large (Liu et al., 2019) trained on the Wiki+Books corpus (Zhu et al., 2015), 12-layer GPT (Brown et al., 2020) on WikiText-103 (Merity et al., 2017) and ResNet-50 trained on ImageNet (Deng et al., 2009). We use the efficient language model pre-training recipe outlined in Izsak et al. (2021). Appendix I has many additional settings, including ablations against other scaling rules (Appendix I.1).

## 6 SVAG for Adaptive Algorithms

Validating the approximation strength captured in Definition 2.4 involves comparing the discrete algorithms and their SDEs on a set of test functions. However, obtaining the SDE solution through traditional simulations, e.g., Euler-Maruyama, is computationally intractable.[4]

Li et al. (2021) proposed an efficient simulation, SVAG, of the SDE for SGD in the finite LR regime: scale the constant LR by $1/\ell$ and take the limit $\ell \to \infty$. In practice the simulation converges for a small value of $\ell$. We adapt SVAG technique to simulate our proposed SDEs, which requires additionally adjusting the moment averaging hyperparameters (i.e., $\beta$, $\beta_1$, $\beta_2$) and $\epsilon$.

**Definition 6.1** (SVAG Operator). Given an NGOS $\mathcal{G}_\sigma = (f, \boldsymbol{\Sigma}, \mathcal{Z}_\sigma)$ with scale $\sigma$ (Definition 2.3) and hyperparameter $\ell \geq 1$, the SVAG operator transforms $\mathcal{G}_\sigma$ into an NGOS $\widehat{\mathcal{G}}_{\ell\sigma} = (f, \boldsymbol{\Sigma}, \widehat{\mathcal{Z}}_{\ell\sigma})$ with scale $\ell\sigma$. The NGOS $\widehat{\mathcal{G}}_{\ell\sigma}$ takes an input $\boldsymbol{\theta}$ and returns $\hat{\boldsymbol{g}} = r_1(\ell)\boldsymbol{g}_1 + r_2(\ell)\boldsymbol{g}_2$, where $\boldsymbol{g}_1, \boldsymbol{g}_2$ are two

---

[4]One can also simulate the SDE by constructing 1st-order weak approximations while taking $\eta \to 0$ along the scaling rules, but the batch size cannot be smaller than 1 and hence $\eta$ cannot go arbitrarily close to the limit.

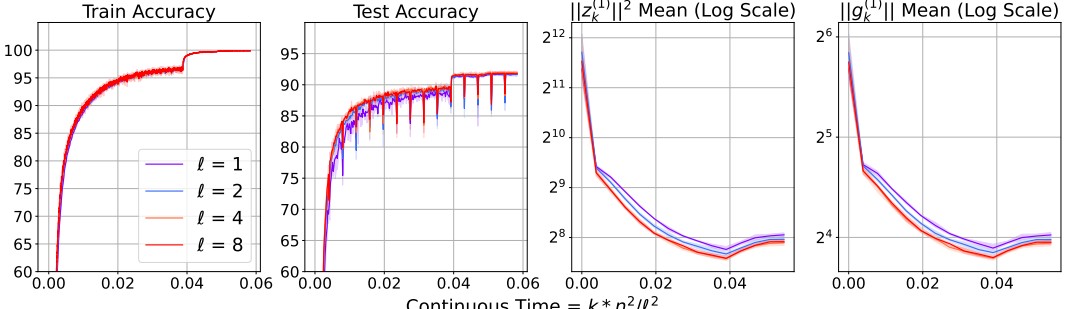

Figure 3: SVAG on the Adam trajectory when training ResNet-50 on CIFAR-10 matches the discrete trajectory ($\ell = 1$) on various test functions (see Appendix J for details). The closeness of the trajectories with respect to various test functions for different values of $\ell$ implies the SDE approximation (Definition 4.4) is a 1st-order weak approximation of Adam (Theorem 4.5).

stochastic gradients from $\mathcal{G}_\sigma(\boldsymbol{\theta})$ and $r_i(\ell) = \frac{1}{2}(1 + (-1)^i \sqrt{2\ell^2 - 1})$ for $i \in \{1, 2\}$. The probability distribution $\widehat{\mathcal{Z}}_{\ell\sigma}$ is defined such that $\hat{\boldsymbol{g}}$ has the same distribution as $\nabla f(\boldsymbol{\theta}) + \ell\sigma\boldsymbol{z}$ when $\boldsymbol{z} \sim \widehat{\mathcal{Z}}_{\ell\sigma}(\boldsymbol{\theta})$.

Lemma E.1 verifies that $\widehat{G}_{\ell\sigma}$ does indeed compute stochastic gradients for $f$ with covariance $\boldsymbol{\Sigma}$. Applying the SVAG operator to mini-batch training amplifies the noise scale by $\ell$. We then apply the square root scaling rule to adjust the learning rates and other hyperparameters accordingly, which yields the SVAG algorithm.

**Definition 6.2** (SVAG Algorithm). For a loss $f$, SVAG operator hyperparameter $\ell > 0$, and optimization hyperparameters $\eta, \beta, \beta_1, \beta_2$, and $\epsilon$, compute the stochastic gradient as $\hat{\boldsymbol{g}} = r_1(\ell)\boldsymbol{g}_{\gamma_1} + r_2(\ell)\boldsymbol{g}_{\gamma_2}$, where $r_1$ and $r_2$ are defined as in Definition 6.1, and scale the optimization hyperparameters:

1. For RMSprop, set $\eta \leftarrow \eta/\ell$, $\beta \leftarrow 1 - (1-\beta)/\ell^2$, and $\epsilon \leftarrow \epsilon\ell$ and apply updates as in Definition 2.1.

2. For Adam, set $\eta \leftarrow \eta/\ell$, $\beta_1 \leftarrow 1 - (1 - \beta_1)/\ell^2$, $\beta_2 \leftarrow 1 - (1 - \beta_2)/\ell^2$ and $\epsilon \leftarrow \epsilon\ell$ and apply updates as in Definition 2.2.

The SVAG algorithm describes a discrete trajectory that is a 1st-order approximation of the corresponding SDE (Definitions 4.1 and 4.4), thereby providing an efficient simulation of the SDEs.

**Theorem 6.3** (SVAG algorithm approximates SDE). *Assume the NGOS is well-behaved and satisfies the bounded moments condition (Definitions 2.5 and 2.7).*

1. *Let $\boldsymbol{X}_t$ be the state of the RMSprop SDE (Definition 4.1) with hyperparameters $\eta$, $\beta$, and $\epsilon$. Let $\boldsymbol{x}_k$ be the state of the analogous discrete SVAG algorithm with hyperparameter $\ell$.*

2. *Let $\boldsymbol{X}_t$ be the state of the Adam SDE (Definition 4.4) with hyperparameters $\eta$, $\beta_1$, $\beta_2$, and $\epsilon$. Let $\boldsymbol{x}_k$ be the state of the analogous discrete SVAG algorithm with hyperparameter $\ell$.*

*In both 1 and 2, following holds for $g$ and $T$ as in Definition 2.4.*

$$\max_{k=0,\ldots,\lfloor \ell^2 T/\eta^2 \rfloor} |\mathbb{E}g(\boldsymbol{x}_k) - \mathbb{E}g(\boldsymbol{X}_{k\eta^2/\ell^2})| \leq C\eta^2/\ell^2$$

*Proof.* The main idea of the proof is to show that the SVAG operator transforms the noise distribution of a well-behaved NGOS satisfying the bounded moments condition into one that is well-behaved and satisfies the bounded moments and the low skewness conditions (Lemma E.2). With these three conditions satisfied, we can directly apply Theorems 4.2 and 4.5 to complete the proof. □

Because the SDE scales time as $k = t/\eta^2$, we must run SVAG for $\ell^2$ steps to match a single step of the discrete trajectories. Nevertheless, we note that in our setting and in Li et al. (2021), the approximation guarantee is strong enough for small $\ell$, so this simulation is still more efficient than Euler-Maruyama.

**Experiments.** Figure 3 compares the Adam SVAG trajectories (Definition 6.2) up to $\ell = 8$ to the discrete one ($\ell = 1$) on CIFAR-10 (Krizhevsky et al.) with ResNet-50 (He et al., 2015b). We use $\text{Tr}(\mathbf{\Sigma}(\boldsymbol{\theta}_k))$ and $\|\boldsymbol{g}_k\|$ as mathematically well-behaved test functions to test the approximation strength (see Definition 2.4). We also measure the train and test accuracies, even though they are not differentiable (and hence, not covered by the theory). The converged SVAG trajectories are close to the discrete ones under these test functions, suggesting the SDE approximations are applicable to realistic deep learning settings. Additional details and settings, including large language models, are in Appendix H.

## 7    Conclusion

We derive SDEs that are provable 1st-order approximations of the RMSprop and Adam trajectories, immediately yielding formal derivations of square root scaling rules: increase the learning rate by $\sqrt{\kappa}$ and adjust the adaptive hyperparameters when increasing batch size by $\kappa$. Experiments in the vision and language domains verify that applying these rules ensures that the values of several test functions, including test accuracy, are preserved. We furthermore design an efficient simulation for the SDEs, allowing us to directly validate the applicability of these SDEs to common vision and language settings. These SDEs can lead to a deeper understanding of how adaptivity and stochasticity impact optimization and generalization, and we hope to extend our results to formal identification of necessary and sufficient conditions for the approximation and its consequences to hold.

## Acknowledgements

We thank Zhiyuan Li and Chao Ma for helpful discussion. We also thank Tianyu Gao and Alexander Wettig for helping us run language modeling experiments. This work is funded by NSF, ONR, Simons Foundation, DARPA and SRC.

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
