# A  Contextualizing our Work

## A.1  Additional Recent Works

Variations of Adam have been proposed to improve its speed of convergence, generalization, and stability during training. Reddi et al. (2018) observed that Adam does not collect long-term memory of past gradients and therefore the effective learning rate could be increasing in some cases. Hence, they propose AMSGrad that maintains a maximum over the exponential running average of the squared gradients. Zaheer et al. (2018) proposed a more controlled increase in the effective learning rate by switching to additive updates, using a more refined version of AdaGrad (Duchi et al., 2011). Chen et al. (2021) unified the generalization ability of vanilla SGD and the convergence speed of Adam by introducing a new adaptive parameter $p \in (0, \frac{1}{2}]$ that can be hypertuned for each setting. Other variations include (a) Nadam (Dozat, 2016) that uses Nesterov momentum, (b) AdamW (Loshchilov and Hutter, 2019) that decouples the weight decay from the optimization step, (c) AdaBound (Luo et al., 2019) that maintains a dynamic upper and lower bound on the step size, (d) AdaBelief (Zhuang et al., 2020) uses a decaying average of estimated variance in the gradient in place of the running average of the squared gradients, (e) QHAdam (Ma and Yarats, 2019) that replaces both momentum estimators in Adam with quasi-hyperbolic terms, etc. LAMB (You et al., 2020) used a layerwise adaptive version of Adam to pretrain large language models efficiently.

## A.2  Broader Impact

Our work is primarily theoretical in nature, but we discuss its broader impacts here. Strubell et al. (2020) highlighted the environmental impact of training large language models. Formal scaling rules remove the need to grid search over hyperparameters: in the case of adaptive algorithms, the grid search is over an even larger space because of the additional adaptivity hyperparameters. Hence, our work reduces the number of times researchers need to train very large models to find ones that work just as well as their smaller, slower counterparts. At the same time, we recognize that the presence of a formal scaling rule may encourage people who were otherwise discouraged by the daunting grid search to train very large models.

# B  SDE Approximation Theorem

In this section, we introduce a theorem in aid of our analysis for approximating Stochastic Gradient Algorithm (SGA) with Stochastic Differential Equation (SDE).

We consider Stochastic Gradient Algorithm (SGA) of the following form:

$$\boldsymbol{x}_{k+1} = \boldsymbol{x}_k + \eta_{\mathrm{e}} \boldsymbol{h}_k(\boldsymbol{x}_k, \boldsymbol{\xi}_k, \eta_{\mathrm{e}}), \tag{4}$$

where $\boldsymbol{x}_k \in \mathbb{R}^D$ is the parameter vector, $\eta_{\mathrm{e}}$ is the learning rate, $\boldsymbol{h}_k$ is a vector-valued function that can depend on the current step $k$, the current parameter $\boldsymbol{x}_k$, a random vector $\boldsymbol{\xi}_k$, and the learning rate $\eta_{\mathrm{e}}$. The random vector $\boldsymbol{\xi}_k$ is sampled from a certain distribution $\Xi(\boldsymbol{x}_k, \eta_{\mathrm{e}})$ in every step.

We consider Stochastic Differential Equation (SDE) of the following form:

$$\mathrm{d}\boldsymbol{X}_t = \boldsymbol{b}(X_t, t)\mathrm{d}t + \boldsymbol{\sigma}(X_t, t)\mathrm{d}\boldsymbol{W}_t, \tag{5}$$

where $\boldsymbol{b} : \mathbb{R}^D \times \mathbb{R} \to \mathbb{R}^D$ is the drift vector function, $\boldsymbol{\sigma} : \mathbb{R}^D \times \mathbb{R} \to \mathbb{R}^{D \times D}$ is the diffusion matrix function.

Let $\mathcal{P}_X(\boldsymbol{x}, s, t)$ be the probability distribution of $\boldsymbol{X}_t$ in (5) when the initial condition is $\boldsymbol{X}_s = \boldsymbol{x}$. Now we define the following random variables to characterize the one-step changes of SGA and SDE.

$$\boldsymbol{\Delta}(\boldsymbol{x}, k) := \eta_{\mathrm{e}} \boldsymbol{h}_k(\boldsymbol{x}, \boldsymbol{\xi}, \eta_{\mathrm{e}}), \qquad \text{where} \quad \boldsymbol{\xi} \sim \Xi(\boldsymbol{x}, \eta_{\mathrm{e}}). \tag{6}$$

$$\tilde{\boldsymbol{\Delta}}(\boldsymbol{x}, k) := \boldsymbol{X}_{(k+1)\eta_{\mathrm{e}}} - \boldsymbol{x}, \qquad \text{where} \quad \boldsymbol{X}_{(k+1)\eta_{\mathrm{e}}} \sim \mathcal{P}_X(\boldsymbol{x}, k\eta_{\mathrm{e}}, (k+1)\eta_{\mathrm{e}}). \tag{7}$$

The following regularity condition is needed for our main theorem.

**Definition B.1.** A function $g \colon \mathbb{R}^d \to \mathbb{R}$ is said to have *polynomial growth* if there exist positive integers $\kappa_1, \kappa_2 > 0$ such that

$$|g(\boldsymbol{x})| \leq \kappa_1(1 + \|\boldsymbol{x}\|_2^{2\kappa_2}),$$

for all $\boldsymbol{x} \in \mathbb{R}^d$. We let $G$ denote the set of all such functions. For each integer $\alpha \geq 1$, $G^\alpha$ denotes the set of $\alpha$-times continuously differentiable functions $\mathbb{R}^d \to \mathbb{R}$ which, together with its partial derivatives up to and including order $\alpha$, belong to $G$. If $g$ depends on additional parameters, we say $g \in G^\alpha$ uniformly if the constants $\kappa_1, \kappa_2$ are independent of those parameters.

Now we are ready to introduce the main theorem for the order of approximation by modeling SGA as SDE. We follow the same proof strategy as Li et al. (2019). First, we show that the moments of the one-step difference for the SGA and for the SDE of our interest are close to each other. Then, under some regularity conditions, we translate the one-step error to an error over a finite interval of time to get the final result. The following is our main theorem to achieve such translation, and it is adapted from Theorem 3 in Li et al. (2019) with slightly different conditions to match our need in the analysis of adaptive gradient methods.

**Theorem B.2** (Adaption of Theorem 3 in Li et al. (2019)). *Let $T > 0$, $\eta_{\mathrm{e}} \in (0, 1 \wedge T)$ and $N = \lfloor T/\eta_{\mathrm{e}} \rfloor$. Consider an SDE with drift vector $\boldsymbol{b}(\boldsymbol{x}, \eta_{\mathrm{e}})$ and diffusion matrix $\boldsymbol{\sigma}(\boldsymbol{x}, \eta_{\mathrm{e}})$, and a stochastic gradient algorithm with initial point $\boldsymbol{x}_0 \in \mathbb{R}^D$ and update rule $\boldsymbol{x}_{k+1} = \boldsymbol{x}_k + \eta_{\mathrm{e}} \boldsymbol{h}(\boldsymbol{x}_k, \boldsymbol{\xi}_k, \eta_{\mathrm{e}})$. Let $\mathcal{X}_k$ be the support of the random variable $\boldsymbol{x}_k$ given $\boldsymbol{x}_0$. Assume the following conditions hold:*

*(1). The drift function $\boldsymbol{b}$ and diffusion function $\boldsymbol{\sigma}$ are Lipschitz and belong to $G^4$;*

*(2). The moments of $\boldsymbol{\Delta} \in \mathbb{R}^D$ and $\tilde{\boldsymbol{\Delta}} \in \mathbb{R}^D$ satisfy the following. There is a function $K_1 \in G$ (independent of $\eta_{\mathrm{e}}$) such that for all $\boldsymbol{x} \in \mathcal{X}_k$ and $1 \leq i, j, l \leq D$,*

$$\left| \mathbb{E}[\Delta_i(\boldsymbol{x}, k) - \tilde{\Delta}_i(\boldsymbol{x}, k)] \right| \leq K_1(\boldsymbol{x})\eta_{\mathrm{e}}^2,$$

$$\left| \mathbb{E}[\Delta_i(\boldsymbol{x}, k)\Delta_j(\boldsymbol{x}, k) - \tilde{\Delta}_i(\boldsymbol{x}, k)\tilde{\Delta}_j(\boldsymbol{x}, k)] \right| \leq K_1(\boldsymbol{x})\eta_{\mathrm{e}}^2,$$

$$\left| \mathbb{E}[\Delta_i(\boldsymbol{x}, k)\Delta_j(\boldsymbol{x}, k)\Delta_l(\boldsymbol{x}, k) - \tilde{\Delta}_i(\boldsymbol{x}, k)\tilde{\Delta}_j(\boldsymbol{x}, k)\tilde{\Delta}_l(\boldsymbol{x}, k)] \right| \leq K_1(\boldsymbol{x})\eta_{\mathrm{e}}^2.$$

*(3). There exists a subset $P$ of the index set $\{1, 2, \ldots, D\}$ such that the following holds. Below we use the notations $\|\boldsymbol{x}\|_{\mathrm{P}} := \sqrt{\sum_{i \in P} x_i^2}$ and $\|\boldsymbol{x}\|_{\mathrm{R}} := \sqrt{\sum_{i \notin P} x_i^2}$.*

*   *There are constants $C_1 > 0, \omega_1 > 0$ (independent of $\eta_{\mathrm{e}}$) so that for all $k \leq N$ and $\boldsymbol{x} \in \mathcal{X}_k$,*

$$\|\mathbb{E}\boldsymbol{\Delta}(\boldsymbol{x}, k)\|_{\mathrm{P}} \leq C_1\eta_{\mathrm{e}}(1 + \|\boldsymbol{x}\|_{\mathrm{P}}),$$
$$\|\mathbb{E}\boldsymbol{\Delta}(\boldsymbol{x}, k)\|_{\mathrm{R}} \leq C_1\eta_{\mathrm{e}}(1 + \|\boldsymbol{x}\|_{\mathrm{P}}^{\omega_1})(1 + \|\boldsymbol{x}\|_{\mathrm{R}}).$$

*   *For all $m \geq 1$, there are constants $C_{2m}, \omega_{2m} > 0$ (independent of $\eta_{\mathrm{e}}$) so that for all $k \leq N$ and $\boldsymbol{x} \in \mathcal{X}_k$,*

$$\mathbb{E}\|\boldsymbol{\Delta}(\boldsymbol{x}, k)\|_{\mathrm{P}}^{2m} \leq C_{2m}\eta_{\mathrm{e}}^m(1 + \|\boldsymbol{x}\|_{\mathrm{P}}^{2m}),$$
$$\mathbb{E}\|\boldsymbol{\Delta}(\boldsymbol{x}, k)\|_{\mathrm{R}}^{2m} \leq C_{2m}\eta_{\mathrm{e}}^m(1 + \|\boldsymbol{x}\|_{\mathrm{P}}^{\omega_{2m}})(1 + \|\boldsymbol{x}\|_{\mathrm{R}}^{2m}).$$

*Then for each function $g \in G^4$, there exists a constant $C > 0$ (independent of $\eta_{\mathrm{e}}$) such that*

$$\max_{0 \leq k \leq N} |\mathbb{E}[g(\boldsymbol{x}_k)] - \mathbb{E}[g(\boldsymbol{X}_{k\eta_{\mathrm{e}}})]| \leq C\eta_{\mathrm{e}}, \tag{8}$$

*when the SDE starts from the same initial point $\boldsymbol{x}_0$ as SGA. That is, the SGA is a first-order weak approximation of the SDE.*

To apply Theorem B.2, the major condition to verify is that $\boldsymbol{\Delta}$ and $\tilde{\boldsymbol{\Delta}}$ match in moments. The following lemma computes the moments for $\tilde{\boldsymbol{\Delta}}$:

**Lemma B.3.** *For drift function $b$ and diffusion function $\boldsymbol{\sigma}$ that belong to $G^4$, there is a function $K_2 \in G$ (independent of $\eta_{\mathrm{e}}$) such that for all $\boldsymbol{x} \in \mathbb{R}^D$ and $1 \leq i, j, l \leq D$,*

$$\left| \mathbb{E}[\tilde{\Delta}_i(\boldsymbol{x}, k)] - \eta_{\mathrm{e}}b_i(\boldsymbol{x}, k\eta_{\mathrm{e}}) \right| \leq K_2(\boldsymbol{x})\eta_{\mathrm{e}}^2,$$

$$\left| \mathbb{E}[\tilde{\Delta}_i(\boldsymbol{x}, k)\tilde{\Delta}_j(\boldsymbol{x}, k)] - \eta_{\mathrm{e}}^2 \sum_{k=1}^{D} \sigma_{i,k}(\boldsymbol{x}, k\eta_{\mathrm{e}})\sigma_{j,k}(\boldsymbol{x}, k\eta_{\mathrm{e}}) \right| \leq K_2(\boldsymbol{x})\eta_{\mathrm{e}}^2,$$

$$\left| \mathbb{E}[\tilde{\Delta}_i(\boldsymbol{x}, k)\tilde{\Delta}_j(\boldsymbol{x}, k)\tilde{\Delta}_l(\boldsymbol{x}, k)] \right| \leq K_2(\boldsymbol{x})\eta_{\mathrm{e}}^2.$$

## B.1  Proof for Theorem B.2

To prove Theorem B.2, we need the following lemma from Li et al. (2019). However, the original version of the lemma does not apply to time-dependent SDEs, i.e., $\boldsymbol{b}$ and $\boldsymbol{\sigma}$ cannot change with time $t$. By carefully scrutinizing the proof, we find that the proof is indeed applicable to time-dependent SDEs.

**Lemma B.4** (Adaption of Proposition 25, Li et al. (2019)). *Suppose that the drift function $\boldsymbol{b}$ and diffusion function $\boldsymbol{\sigma}$ are Lipschitz and belong to $G^\alpha$ for some $\alpha \geq 1$. Let $s \in [0, T]$ and $g \in G^\alpha$. For $t \in [s, T]$, define*

$$u(\boldsymbol{x}, s, t) := \mathbb{E}_{\boldsymbol{X}_t \sim \mathcal{P}_X(\boldsymbol{x}, s, t)}[g(\boldsymbol{X}_t)],$$

*Then $u(\cdot, s, t) \in G^\alpha$ uniformly in $s, t$.*

We need the following lemma to bound the growth of $\boldsymbol{x}_k$.

**Lemma B.5.** *Under Condition (3) of Theorem B.2, given the initial point $\boldsymbol{x}_0$ and any $m \geq 1$, there exists a constant $C'_{2m} > 0$ (depending on $\boldsymbol{x}_0$ but independent of $\eta_\mathrm{e}$) such that the parameters $\{\boldsymbol{x}_k\}$ of SGA starting from $\boldsymbol{x}_0$ can be uniformly bounded by $\mathbb{E}[\|\boldsymbol{x}_k\|_2^{2m}] \leq C'_{2m}$ for all $k \leq N := \lfloor T/\eta_\mathrm{e}\rfloor$.*

*Proof.* It suffices to show that both $\mathbb{E}[(1 + \|\boldsymbol{x}_k\|_\mathrm{P}^2)^m]$ and $\mathbb{E}[(1 + \|\boldsymbol{x}_k\|_\mathrm{R}^2)^m]$ are uniformly bounded.

First, we show that for $\mathbb{E}[(1 + \|\boldsymbol{x}_k\|_\mathrm{P}^2)^m]$. For all $2 \leq j \leq 2m$, by Jensen's inequality,

$$\mathbb{E}\left[\|\boldsymbol{\Delta}(\boldsymbol{x}_k, k)\|_\mathrm{P}^j \mid \boldsymbol{x}_k\right] \leq \mathbb{E}\left[\|\boldsymbol{\Delta}(\boldsymbol{x}_k, k)\|_\mathrm{P}^{2m} \mid \boldsymbol{x}_k\right]^{\frac{j}{2m}} \leq C_{2m}^{\frac{j}{2m}} \eta_\mathrm{e}^{j/2}(1 + \|\boldsymbol{x}_k\|_\mathrm{P}^{2m})^{\frac{j}{2m}}.$$

Then there exists a constant $\hat{C}_1 > 0$ so that

$$\mathbb{E}\left[\|\boldsymbol{\Delta}(\boldsymbol{x}_k, k)\|_\mathrm{P}^j \mid \boldsymbol{x}_k\right] \leq \hat{C}_1 \eta_\mathrm{e}(1 + \|\boldsymbol{x}_k\|_\mathrm{P}^j). \tag{9}$$

For any two vectors $\boldsymbol{x}, \boldsymbol{y} \in \mathbb{R}^D$, we use the notation $\langle \boldsymbol{x}, \boldsymbol{y}\rangle_\mathrm{P} := \sum_{i\in P} x_i y_i$ to denote the inner product of $\boldsymbol{x}$ and $\boldsymbol{y}$ restricting on coordinates in $P$. Now we expand $\left(1 + \|\boldsymbol{x}_{k+1}\|_\mathrm{P}^2\right)^m$ using the update rule. Let $\delta_k := 2\langle \boldsymbol{x}_k, \boldsymbol{\Delta}(\boldsymbol{x}_k, k)\rangle_\mathrm{P} + \|\boldsymbol{\Delta}(\boldsymbol{x}_k, k)\|_\mathrm{P}^2$. Then by the binomial theorem,

$$\begin{aligned}
\mathbb{E}\left[\left(1 + \|\boldsymbol{x}_{k+1}\|_\mathrm{P}^2\right)^m \mid \boldsymbol{x}_k\right] &= \mathbb{E}\left[\left(1 + \|\boldsymbol{x}_k\|_\mathrm{P}^2 + \delta_k\right)^m \mid \boldsymbol{x}_k\right] \\
&= (1 + \|\boldsymbol{x}_k\|_\mathrm{P}^2)^m + m\mathbb{E}[\delta_k \mid \boldsymbol{x}_k](1 + \|\boldsymbol{x}_k\|_\mathrm{P}^2)^{m-1} \\
&\quad + \sum_{j=2}^m \binom{m}{j}\mathbb{E}[\delta_k^j \mid \boldsymbol{x}_k](1 + \|\boldsymbol{x}_k\|_\mathrm{P}^2)^{m-j}.
\end{aligned}$$

By (9), it can be shown that there is a constant $\hat{C}_2$ such that $\mathbb{E}[\delta_k^j \mid \boldsymbol{x}_k] \leq \hat{C}_2 \eta_\mathrm{e}(1 + \|\boldsymbol{x}_k\|_\mathrm{P}^2)^j$ for $j \geq 2$. Then there exists constant $\hat{C}_3, \hat{C}_4$ such that

$$\begin{aligned}
\mathbb{E}\left[\left(1 + \|\boldsymbol{x}_{k+1}\|_\mathrm{P}^2\right)^m \mid \boldsymbol{x}_k\right] &\leq (1 + \|\boldsymbol{x}_k\|_\mathrm{P}^2)^m + m\mathbb{E}[\delta_k \mid \boldsymbol{x}_k](1 + \|\boldsymbol{x}_k\|_\mathrm{P}^2)^{m-1} \\
&\quad + \hat{C}_3 \eta_\mathrm{e}(1 + \|\boldsymbol{x}_k\|_\mathrm{P}^2)^m \\
&\leq (1 + \|\boldsymbol{x}_k\|_\mathrm{P}^2)^m + 2m\left\langle \boldsymbol{x}_k, \mathbb{E}[\boldsymbol{\Delta}(\boldsymbol{x}_k, k) \mid \boldsymbol{x}_k]\right\rangle_\mathrm{P}(1 + \|\boldsymbol{x}_k\|_\mathrm{P}^2)^{m-1} \\
&\quad + \hat{C}_4 \eta_\mathrm{e}(1 + \|\boldsymbol{x}_k\|_\mathrm{P}^2)^m.
\end{aligned}$$

Recall that $\|\mathbb{E}[\boldsymbol{\Delta}(\boldsymbol{x}_k, k) \mid \boldsymbol{x}_k]\|_\mathrm{P} \leq C_1 \eta_\mathrm{e}(1 + \|\boldsymbol{x}_k\|_\mathrm{P})$ by Condition (3). Thus, there exists a constant $\hat{C}_5$ (independent of $\eta_\mathrm{e}, \|\boldsymbol{x}_k\|_\mathrm{P}$) such that

$$\mathbb{E}\left[\left(1 + \|\boldsymbol{x}_{k+1}\|_\mathrm{P}^2\right)^m \mid \boldsymbol{x}_k\right] \leq (1 + \hat{C}_5\eta_\mathrm{e})(1 + \|\boldsymbol{x}_k\|_\mathrm{P}^2)^m.$$

Taking the expectation over $\boldsymbol{x}_k$ gives

$$\mathbb{E}\left[\left(1 + \|\boldsymbol{x}_{k+1}\|_\mathrm{P}^2\right)^m\right] \leq (1 + \hat{C}_5\eta_\mathrm{e})\mathbb{E}[(1 + \|\boldsymbol{x}_k\|_\mathrm{P}^2)^m].$$

Then taking a telescoping product proves that for all $k \leq N$,

$$\begin{aligned}
\mathbb{E}[(1 + \|\boldsymbol{x}_k\|_\mathrm{P}^2)^m] \leq (1 + \hat{C}_5\eta_\mathrm{e})^k(1 + \|\boldsymbol{x}_0\|_\mathrm{P}^2)^m &\leq \exp(\hat{C}_5 k\eta_\mathrm{e})(1 + \|\boldsymbol{x}_0\|_\mathrm{P}^2)^m \\
&\leq \exp(\hat{C}_5 T)(1 + \|\boldsymbol{x}_0\|_\mathrm{P}^2)^m.
\end{aligned}$$

So $\mathbb{E}[(1 + \|\boldsymbol{x}_k\|_{\mathrm{P}}^2)^m]$ is uniformly bounded (independent of $\eta_{\mathrm{e}}, k$).

Now we show that $\mathbb{E}[(1 + \|\boldsymbol{x}_k\|_{\mathrm{R}}^2)^m]$ is uniformly bounded. We can repeat the argument above to bound $\mathbb{E}[(1 + \|\boldsymbol{x}_k\|_{\mathrm{R}}^2)^m]$, while utilizing the bounds $\|\mathbb{E}\boldsymbol{\Delta}(\boldsymbol{x}, k)\|_{\mathrm{R}} \leq C_1 \eta_{\mathrm{e}}(1 + \|\boldsymbol{x}\|_{\mathrm{P}}^{\omega_1})(1 + \|\boldsymbol{x}\|_{\mathrm{R}})$, $\mathbb{E}\|\boldsymbol{\Delta}(\boldsymbol{x}, k)\|_{\mathrm{R}}^{2m} \leq C_{2m} \eta_{\mathrm{e}}^m (1 + \|\boldsymbol{x}\|_{\mathrm{P}}^{\omega_{2m}})(1 + \|\boldsymbol{x}\|_{\mathrm{R}}^{2m})$. In the end we can obtain the following for some real constant $\hat{C}_6 > 0$ and some integer constant $\hat{\omega} > 0$:

$$\mathbb{E}\left[\left(1 + \|\boldsymbol{x}_{k+1}\|_{\mathrm{R}}^2\right)^m\right] \leq (1 + \hat{C}_6 \eta_{\mathrm{e}} \mathbb{E}[1 + \|\boldsymbol{x}_k\|_{\mathrm{P}}^{2\hat{\omega}}]) \mathbb{E}\left[\left(1 + \|\boldsymbol{x}_k\|_{\mathrm{R}}^2\right)^m\right].$$

As we have shown, $\mathbb{E}[1 + \|\boldsymbol{x}_k\|_{\mathrm{P}}^{2\hat{\omega}}]$ is uniformly bounded by a constant. Taking a telescoping product proves that $\mathbb{E}[(1 + \|\boldsymbol{x}_k\|_{\mathrm{R}}^2)^m]$ is uniformly bounded (independent of $\eta_{\mathrm{e}}, k$). $\qquad\square$

We also need the following lemma adapted from Lemma C.2, Li et al. (2021).

**Lemma B.6** (Adaption of Lemma C.2, Li et al. (2021)). *Let $u_1, \ldots, u_N$ be a set of functions that belong to $G^4$ uniformly. Under Conditions (2), (3) in Theorem B.2, if $\boldsymbol{b}$ and $\boldsymbol{\sigma}$ are Lipschitz, then there exists a function $K_1' \in G$ (independent of $\eta_{\mathrm{e}}$) such that*

$$\left|\mathbb{E}[u_j(\boldsymbol{x} + \boldsymbol{\Delta}(\boldsymbol{x}, k))] - \mathbb{E}[u_j(\boldsymbol{x} + \tilde{\boldsymbol{\Delta}}(\boldsymbol{x}, k))]\right| \leq K_1'(\boldsymbol{x}) \eta_{\mathrm{e}}^2,$$

*for all $1 \leq j \leq N$, $1 \leq k \leq N$ and $\boldsymbol{x} \in \mathcal{X}_k$.*

*Proof.* Since $u_1, \ldots, u_N \in G^4$ uniformly, we can find $K_0 \in G$ such that $u_j(\boldsymbol{x})$ is bounded by $K_0(\boldsymbol{x})$ and so are all the partial derivatives of $u_j$ up to order 4.

By Taylor's Theorem with Lagrange Remainder, for all $1 \leq j \leq N$, $1 \leq k \leq N$ we have

$$u_j(\boldsymbol{x} + \boldsymbol{\Delta}(\boldsymbol{x}, k)) - u_j(\boldsymbol{x} + \tilde{\boldsymbol{\Delta}}(\boldsymbol{x}, k))$$
$$= \underbrace{\sum_{s=1}^3 \frac{1}{s!} \sum_{1 \leq i_1, \ldots, i_s \leq D} \frac{\partial^s u_j}{\partial x_{i_1} \cdots \partial x_{i_s}}(\boldsymbol{x}) \left(\prod_{r=1}^s \Delta_{i_r}(\boldsymbol{x}, k) - \prod_{r=1}^s \tilde{\Delta}_{i_r}(\boldsymbol{x}, k)\right)}_{=:M_j} + R_j - \tilde{R}_j,$$

where the remainders $R_j, \tilde{R}_j$ are

$$R_j := \frac{1}{4!} \sum_{1 \leq i_1, \ldots, i_4 \leq D} \frac{\partial^4 u_j}{\partial x_{i_1} \cdots \partial x_{i_4}}(\boldsymbol{x} + a\boldsymbol{\Delta}(\boldsymbol{x}, k)) \prod_{r=1}^4 \Delta_{i_r}(\boldsymbol{x}, k).$$

$$\tilde{R}_j := \frac{1}{4!} \sum_{1 \leq i_1, \ldots, i_4 \leq D} \frac{\partial^4 u_j}{\partial x_{i_1} \cdots \partial x_{i_4}}(\boldsymbol{x} + \tilde{a}\tilde{\boldsymbol{\Delta}}(\boldsymbol{x}, k)) \prod_{r=1}^4 \tilde{\Delta}_{i_r}(\boldsymbol{x}, k).$$

for some $a, \tilde{a} \in [0, 1]$.

By Condition (2), the expectation of $M_j$ can be bounded by

$$\mathbb{E}[M_j] \leq \sum_{s=1}^3 \frac{1}{s!} \sum_{1 \leq i_1, \ldots, i_s \leq D} \left|\frac{\partial^s u_j}{\partial x_{i_1} \cdots \partial x_{i_s}}(\boldsymbol{x})\right| \cdot K_1(\boldsymbol{x}) \eta_{\mathrm{e}}^2 \leq \sum_{s=1}^3 \frac{D^s}{s!} K_0(\boldsymbol{x}) K_1(\boldsymbol{x}) \eta_{\mathrm{e}}^2,$$

so $\frac{1}{\eta_{\mathrm{e}}^2} \mathbb{E}[M_j]$ is uniformly bounded by a function in $G$.

Now let $\kappa_0, m$ be the constants so that $K_0(\boldsymbol{x})^2 \leq \kappa_0^2(1 + \|\boldsymbol{x}\|_2^{2m})$. For $R_j$, by Cauchy-Schwarz inequality we have

$$\mathbb{E}[R_j] \leq \frac{1}{4!} \left(\sum_{i_1, \ldots, i_4} \mathbb{E}\left[\left|\frac{\partial^4 u_j}{\partial x_{i_1} \cdots \partial x_{i_4}}(\boldsymbol{x} + a\boldsymbol{\Delta}(\boldsymbol{x}, k))\right|^2\right]\right)^{1/2} \cdot \left(\sum_{i_1, \ldots, i_4} \mathbb{E}\left[\left|\prod_{r=1}^4 \Delta_{i_r}(\boldsymbol{x}, k)\right|^2\right]\right)^{1/2}$$

$$\leq \frac{1}{4!} \left(\sum_{i_1, \ldots, i_4} \mathbb{E}\left[\left|\frac{\partial^4 u_j}{\partial x_{i_1} \cdots \partial x_{i_4}}(\boldsymbol{x} + a\boldsymbol{\Delta}(\boldsymbol{x}, k))\right|^2\right]\right)^{1/2} \cdot \mathbb{E}\left[\|\boldsymbol{\Delta}(\boldsymbol{x}, k)\|_2^8\right]^{1/2}$$

$$\leq \frac{1}{4!} \left(D^2 \cdot K_0(\boldsymbol{x} + a\boldsymbol{\Delta}(\boldsymbol{x}, k))\right) \cdot \left(\eta_{\mathrm{e}}^4 K_8(\boldsymbol{x})\right)^{1/2},$$

where the last line uses Condition (3) and $K_8$ is a function of polynomial growth. For $K_0(\boldsymbol{x} + a\boldsymbol{\Delta}(\boldsymbol{x}, k))$, we can bound its expectation by

$$\mathbb{E}[K_0(\boldsymbol{x} + a\boldsymbol{\Delta}(\boldsymbol{x}, k))] \leq \kappa_0 \mathbb{E}\left[1 + \|\boldsymbol{x} + a\boldsymbol{\Delta}(\boldsymbol{x}, k)\|_2^{2m}\right]^{1/2}$$
$$\leq \kappa_0 \left(1 + 2^{2m-1}\mathbb{E}[\|\boldsymbol{x}\|_2^{2m} + \mathbb{E}\|\boldsymbol{\Delta}(\boldsymbol{x}, k)\|_2^{2m}]\right)^{1/2}$$
$$\leq \kappa_0 \left(1 + 2^{2m-1}(\|\boldsymbol{x}\|_2^{2m} + C_{2m}\eta_e^{2m}(1 + \|\boldsymbol{x}\|_2^{2m}))\right)^{1/2}.$$

Combining this with our bound for $\mathbb{E}[R_j]$ proves that $\frac{1}{\eta_e^2}\mathbb{E}[R_j]$ is uniformly bounded by a function in $G$:

$$\mathbb{E}[R_j] \leq \eta_e^2 \cdot \frac{1}{4!} \cdot D^4 \cdot \kappa_0 \left(1 + 2^{2m-1}(\|\boldsymbol{x}\|_2^{2m} + C_{2m}\eta_e^{2m}(1 + \|\boldsymbol{x}\|_2^{2m}))\right)^{1/2} \cdot K_8^{1/2}(\boldsymbol{x}).$$

Then we can repeat the above argument for $R_j$ while replacing $\Delta$ as $\tilde{\Delta}$, and conclude that $\frac{1}{\eta_e^2}\mathbb{E}[\tilde{R}_j]$ is also uniformly bounded by a function in $G$. To do so, we note that $\boldsymbol{b}, \boldsymbol{\sigma}$ are Lipschitz, and apply a similar argument as in Lemma 26 of Li et al. (2019) to show that for all $s \geq 1$ there exists a function $\tilde{K} \in G$ such that for all $1 \leq i_1, \ldots, i_s \leq D$,

$$\mathbb{E}\left[\left|\prod_{r=1}^{s} \tilde{\Delta}_{i_r}(\boldsymbol{x}, k)\right|\right] \leq \tilde{K}(\boldsymbol{x})\eta_e^s.$$

Finally, we can find a function $K_1' \in G$ such that $\mathbb{E}[M_j] \leq \frac{1}{3}K_1'(\boldsymbol{x})\eta_e^2$, $\mathbb{E}[R_j] \leq \frac{1}{3}K_1'(\boldsymbol{x})\eta_e^2$, $\mathbb{E}[\tilde{R}_j] \leq \frac{1}{3}K_1'(\boldsymbol{x})\eta_e^2$. Then $\mathbb{E}[u_j(\boldsymbol{x} + \boldsymbol{\Delta}(\boldsymbol{x}, k))] - \mathbb{E}[u_j(\boldsymbol{x} + \tilde{\boldsymbol{\Delta}}(\boldsymbol{x}, k))] \leq K_1'(\boldsymbol{x})\eta_e^2$. $\qquad\square$

Now we are ready to present our proof for Theorem C.2.

*Proof for Theorem B.2.* For $0 \leq j \leq k$, let $\hat{\boldsymbol{x}}_{j,k}$ be a random variable that is distributed as the probability distribution $\mathcal{P}_X(\boldsymbol{x}_j, j\eta_e, k\eta_e)$ conditioned on $\boldsymbol{x}_j$. By definition, $\Pr[\hat{\boldsymbol{x}}_{k,k} = \boldsymbol{x}_k] = 1$, $\hat{\boldsymbol{x}}_{0,k} \sim \boldsymbol{X}_{k\eta_e}$. Let $u(\boldsymbol{x}, s, t) := \mathbb{E}_{\boldsymbol{X}_t \sim \mathcal{P}_X(\boldsymbol{x},s,t)}[g(\boldsymbol{X}_t)]$. Then we can do the following decomposition for $\mathbb{E}[g(\boldsymbol{x}_k)] - \mathbb{E}[g(\boldsymbol{X}_{k\eta_e})]$:

$$|\mathbb{E}[g(\boldsymbol{x}_k)] - \mathbb{E}[g(\boldsymbol{X}_{k\eta_e})]| = \sum_{j=0}^{k-1} (\mathbb{E}[g(\hat{\boldsymbol{x}}_{j+1,k})] - \mathbb{E}[g(\hat{\boldsymbol{x}}_{j,k})])$$
$$= \sum_{j=0}^{k-1} (\mathbb{E}[u(\hat{\boldsymbol{x}}_{j+1,j+1}, (j+1)\eta_e, k\eta_e)] - \mathbb{E}[u(\hat{\boldsymbol{x}}_{j,j+1}, (j+1)\eta_e, k\eta_e)]).$$

Taking absolute values gives

$$|\mathbb{E}[g(\boldsymbol{x}_k)] - \mathbb{E}[g(\boldsymbol{X}_{k\eta_e})]| \leq \sum_{j=0}^{k-1} |\mathbb{E}[u(\hat{\boldsymbol{x}}_{j+1,j+1}, (j+1)\eta_e, k\eta_e)] - \mathbb{E}[u(\hat{\boldsymbol{x}}_{j,j+1}, (j+1)\eta_e, k\eta_e)]|.$$

Let $u_{j+1}(\boldsymbol{x}) := u(\boldsymbol{x}, (j+1)\eta_e, k\eta_e)$. Note that $\hat{\boldsymbol{x}}_{j+1,j+1} \sim \boldsymbol{x}_j + \boldsymbol{\Delta}(\boldsymbol{x}_j, j)$, $\hat{\boldsymbol{x}}_{j,j+1} \sim \boldsymbol{x}_j + \tilde{\boldsymbol{\Delta}}(\boldsymbol{x}_j, j)$. We can rewrite the above formula as

$$|\mathbb{E}[g(\boldsymbol{x}_k)] - \mathbb{E}[g(\boldsymbol{X}_{k\eta_e})]| \leq \sum_{j=0}^{k-1} \left|\mathbb{E}[u_{j+1}(\boldsymbol{x}_j + \boldsymbol{\Delta}(\boldsymbol{x}_j, j))] - \mathbb{E}[u_{j+1}(\boldsymbol{x}_j + \tilde{\boldsymbol{\Delta}}(\boldsymbol{x}_j, j))]\right|.$$

By Lemma B.4, $u \in G^4$ uniformly in $s, t$, so $u_1, \ldots, u_N \in G^4$ uniformly. Then by Lemma B.6, we know that there exists a function $K_1'(\boldsymbol{x}) = \kappa_1(1 + \|\boldsymbol{x}\|_2^{2m}) \in G$ such that

$$\left|\mathbb{E}[u_{j+1}(\boldsymbol{x}_j + \boldsymbol{\Delta}(\boldsymbol{x}_j, j))] - \mathbb{E}[u_{j+1}(\boldsymbol{x}_j + \tilde{\boldsymbol{\Delta}}(\boldsymbol{x}_j, j))]\right| \leq \mathbb{E}[K_1'(\boldsymbol{x}_j)\eta_e^2],$$

for all $0 \leq j < N$. Combining this with Lemma B.5, we can bound $|\mathbb{E}[g(\boldsymbol{x}_k)] - \mathbb{E}[g(\boldsymbol{X}_{k\eta_e})]|$ by

$$|\mathbb{E}[g(\boldsymbol{x}_k)] - \mathbb{E}[g(\boldsymbol{X}_{k\eta_e})]| \leq \sum_{j=0}^{k-1} \mathbb{E}[K_1'(\boldsymbol{x}_j)\eta_e^2] \leq \eta_e^2 \sum_{j=0}^{k-1} \mathbb{E}[\kappa_1(1 + \|\boldsymbol{x}_j\|_2^{2m})]$$
$$\leq \eta_e^2 \sum_{j=0}^{k-1} \kappa_1(1 + C_{2m}') \leq \kappa_1(1 + C_{2m}')T\eta_e.$$

We can complete the proof by noting that $\kappa_1, C_{2m}', T$ are independent of $\eta_e$. $\qquad\square$

## B.2 Proof for Lemma B.3

To prove Lemma B.3, we only need to verify the following lemma using Itô-Taylor expansion.

**Lemma B.7.** *Let $\psi : \mathbb{R}^D \to \mathbb{R}$ be a function in $G^4$. Define*

$$\mathcal{A}_t\psi(\boldsymbol{x}) := \sum_{i \in [D]} b_i(\boldsymbol{x},t)\partial_i\psi(\boldsymbol{x}) + \frac{1}{2}\sum_{i,j \in [D]}\left(\sum_{l \in [D]}\sigma_{i,l}(\boldsymbol{x},t)\sigma_{l,j}(\boldsymbol{x},t)\right)\partial^2_{i,j}\psi(\boldsymbol{x}).$$

*Then there exists a function $\hat{K} \in G$ such that*

$$\left|\mathbb{E}\left[\psi(\boldsymbol{x} + \tilde{\boldsymbol{\Delta}}(\boldsymbol{x},k))\right] - \psi(\boldsymbol{x}) - \eta_{\mathrm{e}}\mathcal{A}_{k\eta_{\mathrm{e}}}\psi(\boldsymbol{x})\right| \leq \hat{K}(\boldsymbol{x})\eta_{\mathrm{e}}^2. \tag{10}$$

*Proof for Lemma B.3.* We can prove the lemma by applying Lemma B.7 with $\psi(\boldsymbol{x})$ being $\psi(\tilde{\boldsymbol{x}}) = \prod_{r=1}^s(\tilde{x}_{i_r} - x_{i_r})$ for any tuple $(i_1,\ldots,i_s) \in [D]^s$ with $s \leq 3$ elements. $\qquad\square$

*Proof for Lemma B.7.* WLOG we prove the case of $k = 0$, then all the other cases can be proved by shifting the time. Let $\Lambda_t\psi(\boldsymbol{x}) := \boldsymbol{\sigma}(\boldsymbol{x},t)^\top \nabla\psi(\boldsymbol{x})$.

$$\psi(\boldsymbol{X}_\eta) = \psi(\boldsymbol{x}) + \int_0^{\eta_{\mathrm{e}}}\mathcal{A}_s\psi(\boldsymbol{X}_s)\mathrm{d}s + \int_0^{\eta_{\mathrm{e}}}\langle\Lambda_s\psi(\boldsymbol{X}_s),\mathrm{d}\boldsymbol{W}_s\rangle.$$

Now we further apply the above formula to $\mathcal{A}_s\psi(\boldsymbol{X}_s)$. Then we have

$$\psi(\boldsymbol{X}_\eta) = \psi(\boldsymbol{x}) + \int_0^{\eta_{\mathrm{e}}}\left(\mathcal{A}_s\psi(\boldsymbol{x}) + \int_0^s\mathcal{A}_r\mathcal{A}_s\psi(\boldsymbol{X}_r)\mathrm{d}r + \int_0^s\langle\Lambda_r\mathcal{A}_s\psi(\boldsymbol{X}_r),\mathrm{d}\boldsymbol{W}_r\rangle\right)\mathrm{d}s$$

$$+ \int_0^{\eta_{\mathrm{e}}}\langle\Lambda_s\psi(\boldsymbol{X}_s),\mathrm{d}\boldsymbol{W}_s\rangle$$

$$= \psi(\boldsymbol{x}) + \int_0^{\eta_{\mathrm{e}}}\mathcal{A}_s\psi(\boldsymbol{x})\mathrm{d}s + \int_0^{\eta_{\mathrm{e}}}\int_0^s\mathcal{A}_r\mathcal{A}_s\psi(\boldsymbol{X}_r)\mathrm{d}r\mathrm{d}s$$

$$+ \int_0^{\eta_{\mathrm{e}}}\int_0^s\langle\Lambda_r\mathcal{A}_s\psi(\boldsymbol{X}_r),\mathrm{d}\boldsymbol{W}_r\rangle\mathrm{d}s + \int_0^{\eta_{\mathrm{e}}}\langle\Lambda_s\psi(\boldsymbol{X}_s),\mathrm{d}\boldsymbol{W}_s\rangle.$$

Taking expectation, the last two integrals vanish. So we have

$$\mathbb{E}\psi(\boldsymbol{X}_\eta) = \psi(\boldsymbol{x}) + \int_0^{\eta_{\mathrm{e}}}\mathcal{A}_s\psi(\boldsymbol{x})\mathrm{d}s + \int_0^{\eta_{\mathrm{e}}}\int_0^s\mathbb{E}[\mathcal{A}_r\mathcal{A}_s\psi(\boldsymbol{X}_r)]\mathrm{d}r\mathrm{d}s.$$

By Lipschitzness of $\boldsymbol{b}$ and $\boldsymbol{\sigma}$, $\frac{1}{\eta_{\mathrm{e}}}(\mathcal{A}_s\psi(\boldsymbol{x}) - \mathcal{A}_0\psi(\boldsymbol{x}))$ is bounded by a function of $\boldsymbol{x}$ with polynomial growth. Also, $\mathcal{A}_r\mathcal{A}_s\psi(\cdot)$ is in $G$ uniformly, then Theorem 19 in (Li et al., 2019) implies that $\mathbb{E}[\mathcal{A}_r\mathcal{A}_s\psi(\boldsymbol{X}_r)]$ is also in $G$ uniformly. Then we know that there exists two functions $\hat{K}_1, \hat{K}_2 \in G$ such that

$$|\mathbb{E}[\psi(\boldsymbol{X}_{\eta_{\mathrm{e}}})] - \psi(\boldsymbol{x}) - \eta_{\mathrm{e}}\mathcal{A}_0\psi(\boldsymbol{x})| \leq \int_0^{\eta_{\mathrm{e}}}\eta_{\mathrm{e}}\hat{K}_1(\boldsymbol{x})\mathrm{d}s + \int_0^{\eta_{\mathrm{e}}}\int_0^s\hat{K}_2(\boldsymbol{x})\mathrm{d}r\mathrm{d}s$$

$$\leq \eta_{\mathrm{e}}^2(\hat{K}_1(\boldsymbol{x}) + \hat{K}_2(\boldsymbol{x})).$$

We can conclude the proof by setting $\hat{K}(\boldsymbol{x}) := \hat{K}_1(\boldsymbol{x}) + \hat{K}_2(\boldsymbol{x})$. $\qquad\square$

# C  RMSProp SDE Proof

In this section, we prove the theorem for the SDE approximation of RMSprop.

**Definition C.1** (SDE for RMSprop, matrix form). For constants $\sigma_0$, $\epsilon_0$, and $c_2$, define the SDE as $\mathrm{d}\boldsymbol{X}_t = \boldsymbol{b}(\boldsymbol{X}_t)\mathrm{d}t + \boldsymbol{\sigma}(\boldsymbol{X}_t)\mathrm{d}\boldsymbol{W}_t$, where $\boldsymbol{X}_t \in \mathbb{R}^d \times \mathbb{R}^d$, and $\boldsymbol{b}$ and $\boldsymbol{\sigma}$ are defined by

$$b_i(\boldsymbol{\theta},\boldsymbol{u}) := -\frac{1}{\sigma_0\sqrt{u_i}+\epsilon_0}\cdot\partial_i f(\boldsymbol{\theta}), \qquad b_{d+i}(\boldsymbol{\theta},\boldsymbol{u}) := c_2(\Sigma(\boldsymbol{\theta})_{i,i} - u_i).$$

$$\sigma_{i,j}(\boldsymbol{\theta},\boldsymbol{u}) := \frac{1}{\sqrt{u_i}+\epsilon_0/\sigma_0}\cdot\left(\boldsymbol{\Sigma}^{1/2}(\boldsymbol{\theta})\right)_{i,j}, \qquad \sigma_{i,d+j}(\boldsymbol{\theta},\boldsymbol{u}) := 0,$$

$$\sigma_{d+i,j}(\boldsymbol{\theta},\boldsymbol{u}) := 0, \qquad \sigma_{d+i,d+j}(\boldsymbol{\theta},\boldsymbol{u}) := 0,$$

for all $1 \leq i,j \leq d$.

**Theorem C.2.** *Fix constants $\sigma_0, c_2 > 0$, $\epsilon_0 \geq 0$. Let $T > 0$, $\eta^2 \in (0, 1 \wedge T \wedge \frac{1}{2c_2})$ and set $N = \lfloor T/\eta^2 \rfloor$. Let $\boldsymbol{u}_k \triangleq \boldsymbol{v}_k / \sigma^2$ and $\boldsymbol{x}_k \triangleq (\boldsymbol{\theta}_k, \boldsymbol{u}_k) \in \mathbb{R}^{2d}$. Let $\{\boldsymbol{x}_k : k \geq 0\}$ be the discrete RMSprop iterations defined in Definition 2.1, where $\sigma, \epsilon, \beta$ are set so that $\sigma_0 = \sigma\eta$, $\epsilon_0 = \epsilon\eta$ and $c_2 = (1 - \beta)/\eta^2$. For well-behaved NGOS that satisfies the bounded moments and low skewness conditions, the SDE as defined in Definition C.1 is an order-1 weak approximation (Definition 2.4) of discrete RMSprop, if they start with $\boldsymbol{X}_0 = \boldsymbol{x}_0$.*

The basic idea is to apply the general theorem (Theorem B.2) with $\eta_{\mathrm{e}} := \eta^2$. However, the SDE above does not satisfy Condition (1) in Theorem B.2 because the denominators such as $\sigma_0 \sqrt{u_i} + \epsilon_0$ can be unbounded. To solve this issue, the first step is to reduce Theorem C.2 to proving the order-1 weak approximation for the following auxiliary SDE:

**Definition C.3.** Define $\tau : \mathbb{R} \to \mathbb{R}$ to be the following smooth transition function:

$$\tau(z) = \begin{cases} 1 & \text{if } z \geq 1, \\ \frac{e^{-1/z}}{e^{-1/z} + e^{-1/(1-z)}} & \text{if } z \in (0, 1), \\ 0 & \text{if } z \leq 0. \end{cases} \tag{11}$$

**Definition C.4** (Auxiliary SDE for RMSprop, matrix form). For constants $\sigma_0, \epsilon_0, c_2$ and $u_{\min}$, define $\mu(u)$ as the following function

$$\mu(u) := \tfrac{1}{2} u_{\min} + \tau\left(\tfrac{2u}{u_{\min}} - 1\right) \cdot \left(u - \tfrac{1}{2} u_{\min}\right), \tag{12}$$

and define the SDE as $\mathrm{d}\boldsymbol{X}_t = \boldsymbol{b}(\boldsymbol{X}_t)\mathrm{d}t + \boldsymbol{\sigma}(\boldsymbol{X}_t)\mathrm{d}\boldsymbol{W}_t$, where $\boldsymbol{X}_t \in \mathbb{R}^d \times \mathbb{R}^d$, and $\boldsymbol{b}$ and $\boldsymbol{\sigma}$ are defined by

$$b_i(\boldsymbol{\theta}, \boldsymbol{u}) := -\frac{1}{\sigma_0 \sqrt{\mu(u_i)} + \epsilon_0} \cdot \partial_i f(\boldsymbol{\theta}), \qquad b_{d+i}(\boldsymbol{\theta}, \boldsymbol{u}) := c_2(\Sigma(\boldsymbol{\theta})_{i,i} - u_i).$$

$$\sigma_{i,j}(\boldsymbol{\theta}, \boldsymbol{u}) := \frac{1}{\sqrt{\mu(u_i)} + \epsilon_0/\sigma_0} \cdot \left(\Sigma(\boldsymbol{\theta})^{1/2}\right)_{i,j}, \qquad \sigma_{i,d+j}(\boldsymbol{\theta}, \boldsymbol{u}) := 0,$$

$$\sigma_{d+i,j}(\boldsymbol{\theta}, \boldsymbol{u}) := 0, \qquad \sigma_{d+i,d+j}(\boldsymbol{\theta}, \boldsymbol{u}) := 0.$$

for all $1 \leq i, j \leq d$.

**Theorem C.5.** *In the setting of Theorem C.2, let $u_{\min} = 2^{-c_2 T} \min_{i \in [d]} u_{0,i}$ then the SDE defined by Definition C.4 is an order-1 weak approximation (Definition 2.4) of discrete RMSprop (Definition 2.1), if they start with $\boldsymbol{X}_0 = \boldsymbol{x}_0$.*

*Proof for Theorem C.2.* Given Theorem C.5, we only need to show that $\boldsymbol{X}_t$ has the same distribution in the original and auxiliary SDEs for all $t \in [0, T]$, when $\boldsymbol{X}_0 = (\boldsymbol{\theta}_0, \boldsymbol{u}_0)$. To see this, we only need to note that in the original SDE

$$\frac{\mathrm{d}u_{t,i}}{\mathrm{d}t} = c_2(\Sigma(\boldsymbol{\theta}_t)_{i,i} - u_{t,i}) \geq -c_2 u_{t,i}.$$

Thus, $u_{t,i} \geq \exp(-c_2 t) u_{0,i} \geq u_{\min}$, which means $\Pr[u_{t,i} = \mu(u_{t,i})] = 1$ for all $t \in [0, T]$. $\qquad \square$

It remains to prove Theorem C.5 by applying Theorem B.2. In the rest of this section, we verify the three conditions in Theorem B.2 respectively.

Below we use the notations $\boldsymbol{x}_k, \boldsymbol{X}_t, \boldsymbol{b}, \boldsymbol{\sigma}$ defined as in Theorem C.5. Let $D = 2d$. Every $\boldsymbol{x}_k \in \mathbb{R}^D$ is a concatenation of two $\mathbb{R}^d$-vectors $\boldsymbol{\theta}_k$ and $\boldsymbol{u}_k$. According to the update rule of RMSprop, $\boldsymbol{x}_k$ can be seen as SGA $\boldsymbol{x}_{k+1} = \boldsymbol{x}_k - \eta_{\mathrm{e}} \boldsymbol{h}_k(\boldsymbol{x}_k, \boldsymbol{z}_k, \eta_{\mathrm{e}})$, where $\eta_{\mathrm{e}} = \eta^2$, $\boldsymbol{z}_k \sim \mathcal{Z}_\sigma(\boldsymbol{\theta}_k)$, and $\boldsymbol{h}_k$ is defined below:

$$\boldsymbol{h}_k(\boldsymbol{\theta}, \boldsymbol{u}, \boldsymbol{z}, \eta_{\mathrm{e}}) := \begin{bmatrix} -(\nabla f(\boldsymbol{\theta}) + \sigma\boldsymbol{z}) \odot (\sigma_0 \sqrt{\boldsymbol{u}} + \epsilon_0)^{-1} \\ c_2\left((\nabla f(\boldsymbol{\theta})/\sigma + \boldsymbol{z})^2 - \boldsymbol{u}\right) \end{bmatrix}.$$

We define $\boldsymbol{\Delta}$ and $\tilde{\boldsymbol{\Delta}}$ as in (6) and (7). Fix $\boldsymbol{x}_0 = (\boldsymbol{\theta}_0, \boldsymbol{u}_0)$ with $u_{0,j} > 0$ for all $j \in [d]$. Define $u_{\min}$ as in Theorem C.5. Let $\mathcal{X}_k$ be the support of the random variable $\boldsymbol{x}_k$ given $\boldsymbol{x}_0$, then it is easy to show that $\mathcal{X}_k$ is a subset of $\{(\boldsymbol{\theta}, \boldsymbol{u}) : u_j \geq u_{\min} \text{ for all } j \in [d]\}$.

## C.1 Verifying Condition (1)

**Lemma C.6.** *The drift function $\boldsymbol{b}$ and diffusion function $\boldsymbol{\sigma}$ are Lipschitz and belong to $G^4$.*

*Proof.* $\boldsymbol{\Sigma}^{1/2}(\boldsymbol{\theta})$ is bounded and Lipschitz, so $\boldsymbol{\Sigma}(\boldsymbol{\theta})$ is Lipschitz. Note that the denominators in the fractions in the formulas of $\boldsymbol{b}$ and $\boldsymbol{\sigma}$ are always lower bounded by a constant. Then the Lipschitz property of $\boldsymbol{b}$ and $\boldsymbol{\sigma}$ can be implied by the Lipschitz property of $\nabla f(\boldsymbol{\theta})$, $\boldsymbol{\Sigma}(\boldsymbol{\theta})$, $\boldsymbol{\Sigma}^{1/2}(\boldsymbol{\theta})$, and $\boldsymbol{b}, \boldsymbol{\sigma} \in G^4$ can be implied by $\nabla f, \boldsymbol{\Sigma}^{1/2}, \mu \in G^4$. $\square$

## C.2 Verifying Condition (2)

To verify Condition (2), we only need to compute the moments of $\boldsymbol{\Delta}$ and $\tilde{\boldsymbol{\Delta}}$ for the discrete RMSprop and the auxiliary SDE, and show that they are close to each other. We compute them by the following two lemmas.

**Lemma C.7.** *For $\boldsymbol{x} = (\boldsymbol{\theta}, \boldsymbol{u}) \in \mathcal{X}_k$, if the NGOS is well-behaved and $\mathcal{Z}_\sigma$ satisfies the bounded moments and low skewness condition, then the moments of $\boldsymbol{\Delta} := \boldsymbol{\Delta}(\boldsymbol{x}, k) - \boldsymbol{x}$ can be written as below.*

1. *For $1 \leq i \leq d$, the following holds for the first moments:*

$$\mathbb{E}[\Delta_i] = -\frac{\eta^2}{\sigma_0 \sqrt{u_i} + \epsilon_0} \partial_i f(\boldsymbol{\theta}), \qquad \mathbb{E}[\Delta_{d+i}] = \eta^2 c_2 \left(\Sigma_{ii}(\boldsymbol{\theta}) - u_i\right) + \frac{\eta^4 c_2}{\sigma_0^2}(\partial_i f(\boldsymbol{\theta}))^2$$

$$= \eta^2 c_2 \left(\Sigma_{ii}(\boldsymbol{\theta}) - u_i\right) + \mathcal{O}(\eta^4).$$

2. *For $1 \leq i, j \leq d$, the following holds for the second moments:*

$$\mathbb{E}[\Delta_i \Delta_j] = \frac{\eta^2 \Sigma_{ij}(\boldsymbol{\theta})}{(\sqrt{u_i} + \epsilon_0/\sigma_0)(\sqrt{u_j} + \epsilon_0/\sigma_0)} + \mathcal{O}(\eta^4) \qquad \mathbb{E}[\Delta_i \Delta_{d+j}] = \mathcal{O}(\eta^4)$$

$$\mathbb{E}[\Delta_{d+i} \Delta_j] = \mathcal{O}(\eta^4) \qquad\qquad\qquad \mathbb{E}[\Delta_{d+i} \Delta_{d+j}] = \mathcal{O}(\eta^4).$$

   *for all $i, j \in [d]$.*

3. *The third moments are bounded by $\mathbb{E}[\boldsymbol{\Delta}^{\otimes 3}] = \mathcal{O}(\eta^4)$.*

*Here the big-O notation $\mathcal{O}(\,\cdot\,)$ is used in a way that $\mathcal{O}(1)$ hides constants (independent of $\eta$ and $\boldsymbol{x}$) and values that are bounded by a function of $\boldsymbol{x}$ with polynomial growth.*

*Proof.* We note that

$$\Delta_i = -\frac{\eta^2}{\sigma_0 \sqrt{u_i} + \epsilon_0}\left(\partial_i f(\boldsymbol{\theta}) + \sigma z_i\right), \qquad \Delta_{d+i} = (1 - \beta)\left((\partial_i f(\boldsymbol{\theta})/\sigma + z_i)^2 - u_i\right).$$

Let $\nu_i := \frac{1}{\sigma_0 \sqrt{u_i} + \epsilon_0}$. Since $\boldsymbol{x} \in \mathcal{X}_k$, $\nu_i \leq \frac{1}{\sigma_0 \sqrt{u_{\min}} + \epsilon_0} = \mathcal{O}(1)$. Writing $1 - \beta$ as $c_2 \eta^2$, we have

$$\Delta_i = -\nu_i \eta^2 \left(\partial_i f(\boldsymbol{\theta}) + \sigma z_i\right), \qquad \Delta_{d+i} = c_2 \eta^2 \left((\partial_i f(\boldsymbol{\theta})/\sigma + z_i)^2 - u_i\right). \qquad (13)$$

We can now compute the first moments:

$$\mathbb{E}[\Delta_i] = -\nu_i \eta^2 \partial_i f(\boldsymbol{\theta}) \qquad\qquad (\mathbb{E}[z_i] = 0)$$

$$\mathbb{E}[\Delta_{d+i}] = c_2 \eta^2 \left(\mathbb{E}[(\partial_i f(\boldsymbol{\theta})/\sigma)^2 + \mathbb{E}[z_i^2] - \mathbb{E}[u_i]\right)$$

$$= c_2 \eta^2 \left((\partial_i f(\boldsymbol{\theta})/\sigma)^2 + \Sigma_{ii} - u_i\right).$$

Now we observe that $1/\sigma = \mathcal{O}(\eta)$, so we can write

$$\mathbb{E}[\Delta_{d+i}] = \eta^2 c_2 (\Sigma_{ii} - u_i) + \mathcal{O}(\eta^4).$$

Let $\boldsymbol{\delta} := \boldsymbol{\Delta} - \mathbb{E}[\boldsymbol{\Delta}]$. That is,

$$\delta_i = -\nu_i \eta^2 \sigma z_i \qquad\qquad \delta_{d+i} = c_2 \eta^2 (z_i^2 - \Sigma_{ii} + 2z_i \partial_i f(\boldsymbol{\theta})/\sigma)$$

$$= -\nu_i \sigma_0 \eta z_i. \qquad\qquad = c_2 \eta^2 (z_i^2 - \Sigma_{ii}) + 2c_2(\partial_i f(\boldsymbol{\theta})/\sigma_0)\eta^3 z_i.$$

For convenience we also define $w_i = (z_i^2 - \Sigma_{ii}) + 2(\partial_i f(\boldsymbol{\theta})/\sigma_0)\eta z_i$ and write $\delta_{d+i} = c_2\eta^2 w_i$.

For the second moments we have

$$\mathbb{E}[\Delta_p\Delta_q] = \mathbb{E}[\delta_p\delta_q] + \mathbb{E}[\Delta_p]\mathbb{E}[\Delta_q] = \mathbb{E}[\delta_p\delta_q] + \mathcal{O}(\eta^4) \qquad \text{for all } 1 \leq p, q \leq 2d.$$

Then it suffices to compute the second moments for $\delta$. For $\mathbb{E}[\delta_i\delta_j]$ we have

$$\mathbb{E}[\delta_i\delta_j] = \nu_i\nu_j\eta^4\sigma^2\mathbb{E}[z_iz_j] = \nu_i\nu_j\sigma_0^2\Sigma_{ij}.$$

For $\mathbb{E}[\delta_i\delta_{d+j}]$ we have

$$\begin{aligned}
\mathbb{E}[\delta_i\delta_{d+j}] &= -c_2\nu_i\sigma_0\eta^3\mathbb{E}\left[z_iw_j\right] \\
&= -c_2\nu_i\sigma_0\eta^3\mathbb{E}[z_iz_j^2] + \mathcal{O}(\eta^4) \\
&= \mathbb{E}[z_iz_j^2]\cdot\mathcal{O}(\eta^3) + \mathcal{O}(\eta^4).
\end{aligned}$$

Similarly, we have $\mathbb{E}[\Delta_{d+i}\Delta_j] = \mathbb{E}[z_i^2z_j]\cdot\mathcal{O}(\eta^3) + \mathcal{O}(\eta^4)$.

For $\mathbb{E}[\delta_{d+i}\delta_{d+j}]$, we note that $\boldsymbol{z} \sim \mathcal{Z}_\sigma(\boldsymbol{\theta})$ has bounded 4th-order moments, so $\mathbb{E}[g(\boldsymbol{z})] = \mathcal{O}(1)$ for any polynomial $g$ of degree at most 4, if the coefficients of $g$ are bounded by $\mathcal{O}(1)$. Then we have

$$\mathbb{E}[\delta_{d+i}\delta_{d+j}] = c_2^2\eta^4\mathbb{E}[w_iw_j] = \mathcal{O}(\eta^4).$$

Now we can check the third moments.

$$\begin{aligned}
\mathbb{E}[\Delta_p\Delta_q\Delta_r] &= \mathbb{E}[\delta_p\delta_q\delta_r] + (\mathbb{E}[\delta_p\delta_q]\mathbb{E}[\Delta_r] + \mathbb{E}[\delta_p\delta_r]\mathbb{E}[\Delta_q] + \mathbb{E}[\delta_q\delta_r]\mathbb{E}[\Delta_p]) + \mathbb{E}[\Delta_p]\mathbb{E}[\Delta_q]\mathbb{E}[\Delta_r] \\
&= \mathbb{E}[\delta_p\delta_q\delta_r] + \mathcal{O}(\eta^2)\cdot\mathcal{O}(\eta^2) + \mathcal{O}(\eta^6) \\
&= \mathbb{E}[\delta_p\delta_q\delta_r] + \mathcal{O}(\eta^4).
\end{aligned}$$

Note that $\delta_p\delta_q\delta_r$ is a polynomial of $\boldsymbol{z}$. For $p = i, q = j, r = k$, by the low skewness condition for $\mathcal{Z}_\sigma$ we have

$$\mathbb{E}[\delta_i\delta_j\delta_k] = -\nu_i^3\sigma_0^3\eta^3\mathbb{E}[z_iz_jz_k] = K_3(\boldsymbol{\theta})/\sigma \cdot \mathcal{O}(\eta^3) = \mathcal{O}(\eta^4).$$

Except the above case, it can be shown that $\delta_p\delta_q\delta_r$ is a polynomial with coefficients bounded by $\mathcal{O}(\eta^4)$. Combining this with the fact that $\boldsymbol{z} \sim \mathcal{Z}_\sigma(\boldsymbol{\theta})$ has bounded moments of any order, we have $\mathbb{E}[\delta_p\delta_q\delta_r] = \mathcal{O}(\eta^4)$. $\qquad\square$

**Lemma C.8.** *For $\boldsymbol{x} = (\boldsymbol{\theta}, \boldsymbol{u}) \in \mathcal{X}_k$, if the NGOS is well-behaved, then the moments of $\tilde{\boldsymbol{\Delta}} := \tilde{\boldsymbol{\Delta}}(x, k)$ can be written as below.*

1. *The first moments are given by*

$$\mathbb{E}[\tilde{\Delta}_i] = -\frac{\eta^2}{\sigma_0\sqrt{\mu(u_i)} + \epsilon_0}\partial_i f(\boldsymbol{\theta}), \qquad \mathbb{E}[\tilde{\Delta}_{d+i}] = \eta^2 c_2\left(\Sigma_{ii}(\boldsymbol{\theta}) - u_i\right) + \mathcal{O}(\eta^4).$$

   *for all $i \in [d]$.*

2. *The second moments are given by*

$$\mathbb{E}[\tilde{\Delta}_i\tilde{\Delta}_j] = \frac{\eta^2\Sigma_{ij}(\boldsymbol{\theta})}{(\sqrt{\mu(u_i)} + \epsilon_0/\sigma_0)(\sqrt{\mu(u_j)} + \epsilon_0/\sigma_0)} + \mathcal{O}(\eta^4) \qquad \mathbb{E}[\tilde{\Delta}_i\tilde{\Delta}_{d+j}] = \mathcal{O}(\eta^4)$$

$$\mathbb{E}[\tilde{\Delta}_{d+i}\tilde{\Delta}_j] = \mathcal{O}(\eta^4) \qquad\qquad\qquad\qquad\qquad \mathbb{E}[\tilde{\Delta}_{d+i}\tilde{\Delta}_{d+j}] = \mathcal{O}(\eta^4).$$

   *for all $i, j \in [d]$.*

3. *The third moments are bounded by $\mathbb{E}[\tilde{\boldsymbol{\Delta}}^{\otimes 3}] = \mathcal{O}(\eta^4)$.*

*Here the big-O notation $\mathcal{O}(\cdot)$ is used in a way that $\mathcal{O}(1)$ hides constants (independent of $\eta$ and $\boldsymbol{x}$) and values that are bounded by a function of $\boldsymbol{x}$ with polynomial growth.*

*Proof.* Applying Lemma B.3 gives

$$\mathbb{E}[\tilde{\boldsymbol{\Delta}}] = \eta^2\boldsymbol{b}(\boldsymbol{x}) + \mathcal{O}(\eta^4), \qquad \mathbb{E}[\tilde{\boldsymbol{\Delta}}\tilde{\boldsymbol{\Delta}}^\top] = \eta^2\boldsymbol{\sigma}(\boldsymbol{x})\boldsymbol{\sigma}(\boldsymbol{x})^\top + \mathcal{O}(\eta^4), \qquad \mathbb{E}[\tilde{\boldsymbol{\Delta}}^{\otimes 3}] = \mathcal{O}(\eta^4).$$

Splitting up the formula by indices proves the claim. $\qquad\square$

## C.3 Verifying Condition (3)

**Lemma C.9.** *Let $P := \{1, 2, \ldots, d\}$. Then*

*1. There is a constant $C_1 > 0$ (independent of $\eta_e$) so that for all $k \leq N$ and $x \in \mathcal{X}_k$,*

$$\|\mathbb{E}\boldsymbol{\Delta}(\boldsymbol{x}, k)\|_{\mathrm{P}} \leq C_1\eta_{\mathrm{e}}(1 + \|\boldsymbol{x}\|_{\mathrm{P}}),$$
$$\|\mathbb{E}\boldsymbol{\Delta}(\boldsymbol{x}, k)\|_{\mathrm{R}} \leq C_1\eta_{\mathrm{e}}(1 + \|\boldsymbol{x}\|_{\mathrm{P}}^2)(1 + \|\boldsymbol{x}\|_{\mathrm{R}}),$$

*2. For all $m \geq 1$, there is a constant $C_{2m}$ (independent of $\eta_e$) so that for all $k \leq N$ and $x \in \mathcal{X}_k$,*

$$\mathbb{E}\|\boldsymbol{\Delta}(\boldsymbol{x}, k)\|_{\mathrm{P}}^{2m} \leq C_{2m}\eta_{\mathrm{e}}^m(1 + \|\boldsymbol{x}\|_{\mathrm{P}}^{2m}),$$
$$\mathbb{E}\|\boldsymbol{\Delta}(\boldsymbol{x}, k)\|_{\mathrm{R}}^{2m} \leq C_{2m}\eta_{\mathrm{e}}^m(1 + \|\boldsymbol{x}\|_{\mathrm{P}}^{4m})(1 + \|\boldsymbol{x}\|_{\mathrm{R}}^{2m}),$$

*Proof.* By Lemma C.7, for all $i \in [d]$,

$$\mathbb{E}[\Delta_i] = -\frac{\eta^2}{\sigma_0\sqrt{u_i} + \epsilon_0}\partial_i f(\boldsymbol{\theta}), \qquad \mathbb{E}[\Delta_{d+i}] = \eta^2 c_2\left(\Sigma_{ii}(\boldsymbol{\theta}) - u_i\right) + \frac{\eta^4 c_2}{\sigma_0^2}(\partial_i f(\boldsymbol{\theta}))^2.$$

Combining this with the Lipschitzness of $\nabla f(\boldsymbol{\theta})$ and the boundedness of $\boldsymbol{\Sigma}(\boldsymbol{\theta})$ proves Item 1.

By (13), for all $i \in [d]$,

$$|\Delta_i| = \nu_i\eta^2 |\partial_i f(\boldsymbol{\theta}) + \sigma z_i| \leq \eta^2\nu_i(1 + |\partial_i f(\boldsymbol{\theta})|)(1 + \sigma|z_i|) \leq \eta\nu_i(1 + |\partial_i f(\boldsymbol{\theta})|)(1 + \sigma_0|z_i|)$$
$$|\Delta_{d+i}| = c_2\eta^2 \left|(\partial_i f(\boldsymbol{\theta})/\sigma + z_i)^2 - u_i\right| \leq c_2\eta^2(1 + (\partial_i f(\boldsymbol{\theta})/\sigma + z_i)^2)(1 + u_i)$$

By the Lipschitzness of $\nabla f(\boldsymbol{\theta})$ and the bounded moments condition for $\mathcal{Z}_\sigma$, we can prove Item 2 by taking powers and expectations on both sides of the above inequalities. $\qquad\square$

# D  Adam SDE Proof

In this section, we prove the theorem for the SDE approximation of Adam.

**Definition D.1** (SDE for Adam, matrix form)**.** For constants $\sigma_0, \epsilon_0, c_1$ and $c_2$, define $\gamma_1(t) := 1 - e^{-c_1 t}, \gamma_2(t) := 1 - e^{-c_2 t}$ and define the SDE as $d\boldsymbol{X}_t = \boldsymbol{b}(\boldsymbol{X}_t)dt + \boldsymbol{\sigma}(\boldsymbol{X}_t)d\boldsymbol{W}_t$, where $\boldsymbol{X}_t \in \mathbb{R}^d \times \mathbb{R}^d \times \mathbb{R}^d$, $\boldsymbol{b}$ is defined by

$$b_i(\boldsymbol{x}, t) := -\frac{\sqrt{\gamma_2(t)}}{\gamma_1(t)} \cdot \frac{m_i}{\sigma_0\sqrt{u_i} + \epsilon_0\sqrt{\gamma_2(t)}},$$
$$b_{d+i}(\boldsymbol{x}, t) := c_1(\partial_i f(\boldsymbol{\theta}) - m_i),$$
$$b_{2d+i}(\boldsymbol{x}, t) := c_2(\Sigma(\boldsymbol{\theta})_{i,i} - u_i),$$

for all $1 \leq i \leq d$, and $\sigma_{d+i, d+j}$ for all $1 \leq i, j \leq d$ is given by

$$\sigma_{d+i, d+j}(\boldsymbol{x}, t) := \sigma_0 c_1\left(\boldsymbol{\Sigma}^{1/2}(\boldsymbol{\theta})\right)_{i,j},$$

and all the other entries of $\boldsymbol{\sigma}$ are zero.

**Theorem D.2.** *Fix $\sigma_0, c_1, c_2 > 0$, $\epsilon_0 \geq 0$. Let $T > 0$, $\eta^2 \in (0, 1 \wedge T \wedge \frac{1}{2c_2})$ and set $N = \lfloor T/\eta^2 \rfloor$. Let $\boldsymbol{u}_k \triangleq \boldsymbol{v}_k/\sigma^2$ and $\boldsymbol{x}_k \triangleq (\boldsymbol{\theta}_k, \boldsymbol{m}_k, \boldsymbol{u}_k) \in \mathbb{R}^{3d}$. Let $\{\boldsymbol{x}_k : k \geq 0\}$ be the discrete Adam iterations defined in Definition 2.2. Set $\sigma, \epsilon, \beta_1, \beta_2$ so that $\sigma_0 = \sigma\eta$, $\epsilon_0 = \epsilon\eta$, $c_1 = (1 - \beta_1)/\eta^2$ and $c_2 = (1 - \beta_2)/\eta^2$. For well-behaved NGOS that satisfies the bounded moments and low skewness conditions, for any constant $t_0 > 0$, the solution $\boldsymbol{X}_t$ ($t \in [t_0, T]$) of the SDE defined as in Definition D.1 is an order-1 weak approximation (Definition 2.4) of the sequence of Adam iterates $\boldsymbol{x}_k$ starting from $k_0 = \lceil t_0/\eta^2 \rceil$, if the initial condition of the SDE is set to $\boldsymbol{X}_{t_0} = \boldsymbol{x}_{k_0}$.*

The proof strategy is essentially the same as that for RMSprop. Similar to what we have done for RMSprop, we turn to prove the approximation order for the following auxiliary SDE.

**Definition D.3** (Auxiliary SDE for Adam, matrix form). For constants $\sigma_0, \epsilon_0, c_1, c_2$ and $u_{\min}$, define $\gamma_1(t) := 1 - e^{-c_1 t}, \gamma_2(t) := 1 - e^{-c_2 t}, \mu(u) := \frac{1}{2}u_{\min} + \tau(\frac{2u}{u_{\min}} - 1) \cdot (u - \frac{1}{2}u_{\min})$, and define the SDE as $\mathrm{d}\boldsymbol{X}_t = \boldsymbol{b}(\boldsymbol{X}_t)\mathrm{d}t + \boldsymbol{\sigma}(\boldsymbol{X}_t)\mathrm{d}\boldsymbol{W}_t$, where $\boldsymbol{X}_t \in \mathbb{R}^d \times \mathbb{R}^d \times \mathbb{R}^d$, $\boldsymbol{b}$ is defined by

$$b_i(\boldsymbol{x}, t) := -\frac{\sqrt{\gamma_2(t)}}{\gamma_1(t)} \cdot \frac{m_i}{\sigma_0\sqrt{\mu(u_i)} + \epsilon_0\sqrt{\gamma_2(t)}},$$
$$b_{d+i}(\boldsymbol{x}, t) := c_1(\partial_i f(\boldsymbol{\theta}) - m_i),$$
$$b_{2d+i}(\boldsymbol{x}, t) := c_2(\Sigma(\boldsymbol{\theta})_{i,i} - u_i),$$

for all $1 \leq i \leq d$, and $\sigma_{d+i,d+j}$ for all $1 \leq i, j \leq d$ is given by

$$\sigma_{d+i,d+j}(\boldsymbol{x}, t) := \sigma_0 c_1 \left(\boldsymbol{\Sigma}^{1/2}(\boldsymbol{\theta})\right)_{i,j},$$

and all the other entries of $\boldsymbol{\sigma}$ are zero.

**Theorem D.4.** *In the setting of Theorem D.2, let $u_{\min} = 2^{-(T-t_0)}\min_{i\in[d]} u_{k_0,i}$, For any constant $t_0 > 0$, the solution $\boldsymbol{X}_t$ ($t \in [t_0, T]$) of the SDE defined as in Definition D.3 is an order-1 weak approximation (Definition 2.4) of the sequence of Adam iterates $\boldsymbol{x}_k$ starting from $k_0 = \lceil t_0/\eta^2 \rceil$, if the initial condition of the SDE is set to $\boldsymbol{X}_{t_0} = \boldsymbol{x}_{k_0}$.*

*Proof for Theorem D.2.* Given Theorem D.4, we only need to show that $\boldsymbol{X}_t$ has the same distribution in the original and auxiliary SDEs for all $t \in [0, T]$, when $\boldsymbol{X}_0 = (\boldsymbol{\theta}_0, \boldsymbol{m}_0, \boldsymbol{u}_0)$. To see this, we only need to note that in the original SDE

$$\frac{\mathrm{d}u_{t,i}}{\mathrm{d}t} = c_2(\Sigma(\boldsymbol{\theta}_t)_{i,i} - u_{t,i}) \geq -c_2 u_{t,i}.$$

Thus, $u_{t,i} \geq \exp(-c_2 t)u_{0,i} \geq u_{\min}$, which means $\Pr[u_{t,i} = \mu(u_{t,i})] = 1$ for all $t \in [0, T]$. $\square$

It remains to prove Theorem D.4 by applying Theorem B.2. In the rest of this section, we verify the three conditions in Theorem B.2 respectively.

Below we use the notations $\boldsymbol{x}_k, \boldsymbol{X}_t, \boldsymbol{b}, \boldsymbol{\sigma}$ defined as in Theorem D.4. Let $D = 3d$. Every $\boldsymbol{x}_k \in \mathbb{R}^D$ is a concatenation of three $\mathbb{R}^d$-vectors $\boldsymbol{\theta}_k, \boldsymbol{m}_k$ and $\boldsymbol{u}_k$. According to the update rule of Adam, $\boldsymbol{x}_k$ can be seen as SGA $\boldsymbol{x}_{k+1} = \boldsymbol{x}_k - \eta_e \boldsymbol{h}_k(\boldsymbol{x}_k, \boldsymbol{z}_k, \eta_e)$, where $\eta_e = \eta^2$, $\boldsymbol{z}_k \sim \mathcal{Z}_\sigma(\boldsymbol{\theta}_k)$, and $\boldsymbol{h}_k$ is defined below:

$$\boldsymbol{h}_k(\boldsymbol{\theta}, \boldsymbol{u}, \boldsymbol{z}, \eta_e) := \begin{bmatrix} -\frac{\sqrt{1-\beta_2^k}}{1-\beta_1^{k+1}}\boldsymbol{m} \odot \left(\sigma_0\sqrt{\boldsymbol{u}} + \epsilon_0\sqrt{1-\beta_2^k}\right)^{-1} \\ c_1(\nabla f(\boldsymbol{\theta}) + \sigma\boldsymbol{z} - \boldsymbol{m}) \\ c_2\left((\nabla f(\boldsymbol{\theta})/\sigma + \boldsymbol{z})^2 - \boldsymbol{u}\right) \end{bmatrix}.$$

We define $\boldsymbol{\Delta}$ and $\tilde{\boldsymbol{\Delta}}$ as in (6) and (7).

Fix $\boldsymbol{x}_0 = (\boldsymbol{\theta}_0, \boldsymbol{m}_0, \boldsymbol{u}_0)$ with $u_{0,j} > 0$ for all $j \in [d]$. Define $u_{\min}$ as in Theorem D.4. Let $\mathcal{X}_k$ be the support of the random variable $\boldsymbol{x}_k$ given $\boldsymbol{x}_0$, then it is easy to show that $\mathcal{X}_k$ is a subset of $\{(\boldsymbol{\theta}, \boldsymbol{m}, \boldsymbol{u}) : u_j \geq u_{\min} \text{ for all } j \in [d]\}$.

## D.1 Verifying Condition (1)

**Lemma D.5.** *The drift function $\boldsymbol{b}$ and diffusion function $\boldsymbol{\sigma}$ are Lipschitz and belong to $G^4$.*

*Proof.* Same argument as for Lemma C.6. $\square$

## D.2 Verifying Condition (2)

Let $\boldsymbol{x} = (\boldsymbol{\theta}, \boldsymbol{m}, \boldsymbol{u}) \in \mathbb{R}^{3d}$. For ease of notation, let $\hat{\gamma}_1 = 1 - \beta_1^{k+1}$ and $\hat{\gamma}_2 = 1 - \beta_2^k$. These are not constants across time steps like the other constants, but they are deterministic and upper bounded by constants for $k \geq t_0/\eta^2$.

To verify Condition 2, we only need to compute the moments of $\boldsymbol{\Delta}$ and $\tilde{\boldsymbol{\Delta}}$ for the discrete Adam and the auxiliary SDE, and show that they are close to each other. We compute them by the following two lemmas.

**Lemma D.6.** *For $x = (\theta, u) \in \mathcal{X}_k$, if the NGOS is well-behaved and $\mathcal{Z}_\sigma$ satisfies the bounded moments and low skewness condition, then the moments of $\Delta := \Delta(x, k)$ can be written as below.*

1. *The first moments are given by*

$$\mathbb{E}[\Delta_i] = -\frac{\sqrt{\hat{\gamma}_2}}{\hat{\gamma}_1} \cdot \frac{\eta^2}{\sigma_0 \sqrt{u_i} + \epsilon_0 \sqrt{\hat{\gamma}_2}}(m_i + c_1 \eta^2 (\partial_i f(\theta) - m_i))$$

$$= -\frac{\sqrt{\hat{\gamma}_2}}{\hat{\gamma}_1} \cdot \frac{\eta^2 m_i}{\sigma_0 \sqrt{u_i} + \epsilon_0 \sqrt{\hat{\gamma}_2}} + \mathcal{O}(\eta^4)$$

$$\mathbb{E}[\Delta_{d+i}] = c_1 \eta^2 (\partial_i f(\theta) - m_i)$$

$$\mathbb{E}[\Delta_{2d+i}] = c_2 \eta^2 ((\partial_i f(\theta)/\sigma)^2 + \Sigma_{ii} - u_i)$$

$$= c_2 \eta^2 (\Sigma_{ii} - u_i) + \mathcal{O}(\eta^4).$$

   *for all $i \in [d]$.*

2. *The second moments are given by*

$$\mathbb{E}[\Delta_p \Delta_q] = \begin{cases} c_1^2 \sigma_0^2 \eta^2 \Sigma_{ij} + \mathcal{O}(\eta^4) & \text{if} \quad p = d+i, q = d+j \quad \text{for some} \quad i, j \in [d] \\ \mathcal{O}(\eta^4) & \text{otherwise.} \end{cases}$$

   *for all $p, q \in [3d]$.*

3. *The third moments are bounded by $\mathbb{E}[\Delta^{\otimes 3}] = \mathcal{O}(\eta^4)$.*

*Here the big-O notation $\mathcal{O}(\cdot)$ is used in a way that $\mathcal{O}(1)$ hides constants (independent of $\eta$ and $x$) and values that are bounded by a function of $x$ with polynomial growth.*

*Proof.* For notational convenience, we write $\Delta, \Sigma$ instead of $\Delta(x), \Sigma(\theta)$. We note that

$$\Delta_i = -\frac{\sqrt{\hat{\gamma}_2}}{\hat{\gamma}_1} \cdot \frac{\eta^2 m_i + \eta^2 (1 - \beta_1)(\partial_i f(\theta) + \sigma z_i - m_i)}{\sigma_0 \sqrt{u_i} + \epsilon_0 \sqrt{\hat{\gamma}_2}}$$

$$\Delta_{d+i} = (1 - \beta_1)(\partial_i f(\theta) + \sigma z_i - m_i)$$

$$\Delta_{2d+i} = (1 - \beta_2)((\partial_i f(\theta)/\sigma + z_i)^2 - u_i)$$

Let $\nu_i := \frac{\sqrt{\hat{\gamma}_2}}{\hat{\gamma}_1} \cdot \frac{1}{\sigma_0 \sqrt{u_i} + \epsilon_0 \sqrt{\hat{\gamma}_2}}$. And we write $1 - \beta = c_2 \eta^2$. Then we have

$$\Delta_i = -\nu_i \eta^2 (m_i + c_1 \eta^2 (\partial_i f(\theta) + \sigma z_i - m_i))$$

$$\Delta_{d+i} = c_1 \eta^2 (\partial_i f(\theta) + \sigma z_i - m_i) \tag{14}$$

$$\Delta_{2d+i} = c_2 \eta^2 ((\partial_i f(\theta)/\sigma + z_i)^2 - u_i)$$

The first moments follow directly:

$$\mathbb{E}[\Delta_i] = -\nu_i \eta^2 (m_i + c_1 \eta^2 (\partial_i f(\theta) - m_i)) = -\nu_i \eta^2 m_i + \mathcal{O}(\eta^4)$$

$$\mathbb{E}[\Delta_{d+i}] = c_1 \eta^2 (\partial_i f(\theta) - m_i)$$

$$\mathbb{E}[\Delta_{2d+i}] = c_2 \eta^2 ((\partial_i f(\theta)/\sigma)^2 + \Sigma_{ii} - u_i) = c_2 \eta^2 (\Sigma_{ii} - u_i) + \mathcal{O}(\eta^4).$$

Let $\delta := \Delta - \mathbb{E}[\Delta]$. That is,

$$\delta_i = -\nu_i c_1 \eta^4 \sigma z_i = -\nu_i c_1 \sigma_0 \eta^3 z_i$$

$$\delta_{d+i} = c_1 \eta^2 \sigma z_i = c_1 \sigma_0 \eta z_i$$

$$\delta_{2d+i} = c_2 \eta^2 (z_i^2 - \Sigma_{ii} + 2z_i \partial_i f(\theta)/\sigma)$$

$$= c_2 \eta^2 (z_i^2 - \Sigma_{ii}) + 2c_2 (\partial_i f(\theta)/\sigma_0) \eta^3 z_i.$$

For convenience we also define $w_i = (z_i^2 - \Sigma_{ii}) + 2(\partial_i f(\theta)/\sigma_0) \eta z_i$ and write $\delta_{d+i} = c_2 \eta^2 w_i$.

Similar as the proof for Lemma C.7, for the second moments we have

$$\mathbb{E}[\Delta_p \Delta_q] = \mathbb{E}[\delta_p \delta_q] + \mathbb{E}[\Delta_p]\mathbb{E}[\Delta_q] = \mathbb{E}[\delta_p \delta_q] + \mathcal{O}(\eta^4) \qquad \text{for all } 1 \le p, q \le 2d.$$

Then it suffices to compute the second order moments for $\delta$. A direct computation gives the following:

$$\mathbb{E}[\delta_i \delta_j] = \nu_i \nu_j c_1^2 \sigma_0^2 \eta^6 \mathbb{E}[z_i z_j] = \mathcal{O}(\eta^6).$$
$$\mathbb{E}[\delta_i \delta_{d+j}] = -\nu_i c_1^2 \sigma_0^2 \eta^4 \mathbb{E}[z_i z_j] = \mathcal{O}(\eta^4).$$
$$\mathbb{E}[\delta_i \delta_{2d+j}] = -\nu_i c_1 c_2 \sigma_0 \eta^5 \mathbb{E}[z_i w_j] = \mathcal{O}(\eta^5).$$
$$\mathbb{E}[\delta_{d+i} \delta_{d+j}] = c_1^2 \sigma_0^2 \eta^2 \mathbb{E}[z_i z_j] = c_1^2 \sigma_0^2 \eta^2 \Sigma_{ij}.$$
$$\mathbb{E}[\delta_{d+i} \delta_{2d+j}] = c_1 c_2 \sigma_0 \eta^3 \mathbb{E}[z_i w_j] = c_1 c_2 \sigma_0 \eta^3 \mathbb{E}[z_i z_j^2] + \mathcal{O}(\eta^4).$$
$$\mathbb{E}[\delta_{2d+i} \delta_{2d+j}] = c_2^2 \eta^4 \mathbb{E}[w_i w_j] = \mathcal{O}(\eta^4).$$

Now we check the third moments. Similar as the proof for Lemma C.7, $\mathbb{E}[\Delta_p \Delta_q \Delta_r] = \mathbb{E}[\delta_p \delta_q \delta_r] + \mathcal{O}(\eta^4)$. Note that $\delta_p \delta_q \delta_r$ is a polynomial of $z$. For $p = d+i, q = d+j, r = d+k$, by the low skewness condition for $\mathcal{Z}_\sigma$ we have

$$\mathbb{E}[\delta_{d+i} \delta_{d+j} \delta_{d+k}] = c_1^3 \sigma_0^3 \eta^3 \mathbb{E}[z_i z_j z_k] = K_3(\boldsymbol{\theta})/\sigma \cdot \mathcal{O}(\eta^3) = \mathcal{O}(\eta^4).$$

Except the above case, it can be shown that $\delta_p \delta_q \delta_r$ is a polynomial with coefficients bounded by $\mathcal{O}(\eta^4)$. Combining this with the fact that $z \sim \mathcal{Z}_\sigma(\boldsymbol{\theta})$ has bounded moments of any order, we have $\mathbb{E}[\delta_p \delta_q \delta_r] = \mathcal{O}(\eta^4)$. $\qquad\square$

**Lemma D.7.** *For $\boldsymbol{x} = (\boldsymbol{\theta}, \boldsymbol{u}) \in \mathcal{X}_k$, if the NGOS is well-behaved, then the moments of $\tilde{\boldsymbol{\Delta}} := \tilde{\boldsymbol{\Delta}}(\boldsymbol{x}, k)$ can be written as below.*

1. *The first moments are given by*

$$\mathbb{E}[\Delta_i] = -\frac{\sqrt{\hat{\gamma}_2}}{\hat{\gamma}_1} \cdot \frac{\eta^2 m_i}{\sigma_0 \sqrt{u_i} + \epsilon_0 \sqrt{\hat{\gamma}_2}} + \mathcal{O}(\eta^4)$$
$$\mathbb{E}[\Delta_{d+i}] = c_1 \eta^2 (\partial_i f(\boldsymbol{\theta}) - m_i) + \mathcal{O}(\eta^4)$$
$$\mathbb{E}[\Delta_{2d+i}] = c_2 \eta^2 (\Sigma_{ii} - u_i) + \mathcal{O}(\eta^4).$$

   *for all $i \in [d]$.*

2. *The second moments are given by*

$$\mathbb{E}[\Delta_p \Delta_q] = \begin{cases} c_1^2 \sigma_0^2 \eta^2 \Sigma_{ij} + \mathcal{O}(\eta^4) & \text{if} \quad p = d+i, q = d+j \quad \text{for some} \quad i, j \in [d] \\ \mathcal{O}(\eta^4) & \text{otherwise.} \end{cases}$$

   *for all $p, q \in [3d]$.*

3. *The third moments are bounded by $\mathbb{E}[\boldsymbol{\Delta}^{\otimes 3}] = \mathcal{O}(\eta^4)$.*

*Here the big-O notation $\mathcal{O}(\cdot)$ is used in a way that $\mathcal{O}(1)$ hides constants (independent of $\eta$ and $\boldsymbol{x}$) and values that are bounded by a function of $\boldsymbol{x}$ with polynomial growth.*

*Proof.* Applying Lemma B.3 gives

$$\mathbb{E}[\tilde{\boldsymbol{\Delta}}] = \eta^2 \boldsymbol{b}(\boldsymbol{x}) + \mathcal{O}(\eta^4), \qquad \mathbb{E}[\tilde{\boldsymbol{\Delta}} \tilde{\boldsymbol{\Delta}}^\top] = \eta^2 \boldsymbol{\sigma}(\boldsymbol{x}) \boldsymbol{\sigma}(\boldsymbol{x})^\top + \mathcal{O}(\eta^4), \qquad \mathbb{E}[\tilde{\boldsymbol{\Delta}}^{\otimes 3}] = \mathcal{O}(\eta^4).$$

Noting that $\gamma_1(k\eta_e) = \hat{\gamma}_1 + \mathcal{O}(\eta^2)$ and $\gamma_2(k\eta_e) = \hat{\gamma}_2 + \mathcal{O}(\eta^2)$, we can prove the claim. $\qquad\square$

### D.3 Verifying Condition (3)

**Lemma D.8.** *Let $P := \{1, 2, \ldots, 2d\}$. Then*

1. *There is a constant $C_1 > 0$ (independent of $\eta_e$) so that for all $k \leq N$ and $\boldsymbol{x} \in \mathcal{X}_k$,*

$$\|\mathbb{E}\boldsymbol{\Delta}(\boldsymbol{x}, k)\|_P \leq C_1 \eta_e (1 + \|\boldsymbol{x}\|_P),$$
$$\|\mathbb{E}\boldsymbol{\Delta}(\boldsymbol{x}, k)\|_R \leq C_1 \eta_e (1 + \|\boldsymbol{x}\|_P^2)(1 + \|\boldsymbol{x}\|_R).$$

2. *For all $m \geq 1$, there is a constant $C_{2m}$ (independent of $\eta_e$) so that for all $k \leq N$ and $\boldsymbol{x} \in \mathcal{X}_k$,*

$$\mathbb{E}\|\boldsymbol{\Delta}(\boldsymbol{x}, k)\|_P^{2m} \leq C_{2m} \eta_e^m (1 + \|\boldsymbol{x}\|_P^{2m}),$$
$$\mathbb{E}\|\boldsymbol{\Delta}(\boldsymbol{x}, k)\|_R^{2m} \leq C_{2m} \eta_e^m (1 + \|\boldsymbol{x}\|_P^{4m})(1 + \|\boldsymbol{x}\|_R^{2m}).$$

*Proof.* By Lemma D.6, for all $i \in [d]$,

$$\mathbb{E}[\Delta_i] = -\nu_i \eta^2 (m_i + c_1 \eta^2 (\partial_i f(\boldsymbol{\theta}) - m_i))$$
$$\mathbb{E}[\Delta_{d+i}] = c_1 \eta^2 (\partial_i f(\boldsymbol{\theta}) - m_i)$$
$$\mathbb{E}[\Delta_{2d+i}] = c_2 \eta^2 ((\partial_i f(\boldsymbol{\theta})/\sigma)^2 + \Sigma_{ii} - u_i).$$

Combining this with the Lipschitzness of $\nabla f(\boldsymbol{\theta})$ and the boundedness of $\Sigma(\boldsymbol{\theta})$ proves Item 1.

By (14), for all $i \in [d]$, one can show that there exists a constant $\hat{C}$ such that

$$|\Delta_i| \leq \eta^2 \hat{C}(1 + |m_i| + |\partial_i f(\boldsymbol{\theta})|)(1 + |z_i|)$$
$$|\Delta_{d+i}| \leq \eta^2 \hat{C}(1 + |m_i| + |\partial_i f(\boldsymbol{\theta})|)(1 + |z_i|)$$
$$|\Delta_{2d+i}| \leq \eta^2 \hat{C}(1 + |\partial_i f(\boldsymbol{\theta})|^2 + z_i^2)(1 + u_i)$$

By the Lipschitzness of $\nabla f(\boldsymbol{\theta})$ and the bounded moments condition for $\mathcal{Z}_\sigma$, we can prove Item 2 by taking powers and expectations on both sides of the above inequalities. □

# E    Analysis of SVAG Operator

**Lemma E.1.** *Let $\mathcal{G}_\sigma = (f, \Sigma, \mathcal{Z}_\sigma)$ be a NGOS and $\widehat{\mathcal{G}}_{\ell\sigma} = (f, \Sigma, \widehat{\mathcal{Z}}_{\ell\sigma})$ be the NGOS after applying the SVAG operator with hyperparameter $\ell > 0$. Then $\widehat{\mathcal{G}}_{\ell\sigma}$ is indeed an NGOS. That is, $\widehat{\mathcal{Z}}_{\ell\sigma}(\boldsymbol{\theta})$ is well-defined and has mean zero, covariance $\Sigma(\boldsymbol{\theta})$.*

*Proof.* Let $\widehat{\mathcal{Z}}_{\ell\sigma}(\boldsymbol{\theta})$ be the distribution of $\hat{z} := \frac{1}{\ell}(r_1(\ell)z_1 + r_2(\ell)z_2)$ when $z_1, z_2 \sim \mathcal{Z}_\sigma(\boldsymbol{\theta})$. Then it is easy to check that $\hat{g}$ has the same distribution as $\nabla f(\boldsymbol{\theta}) + \ell\sigma\hat{z}$, since

$$\hat{g} = r_1(\ell)g_1 + r_2(\ell)g_2 \sim r_1(\ell)(\nabla f(\boldsymbol{\theta}) + \sigma z_1) + r_2(\ell)(\nabla f(\boldsymbol{\theta}) + \sigma z_2)$$
$$= (r_1(\ell) + r_2(\ell))\nabla f(\boldsymbol{\theta}) + \sigma(r_1(\ell)z_1 + r_2(\ell)z_2)$$
$$= \nabla f(\boldsymbol{\theta}) + \ell\sigma\hat{z},$$

where the last equality uses the fact that $r_1(\ell) + r_2(\ell) = 1$. Hence $\widehat{\mathcal{Z}}_{\ell\sigma}(\boldsymbol{\theta})$ is well-defined.

Now we check the mean and covariance of $\hat{z} \sim \widehat{\mathcal{Z}}_{\ell\sigma}(\boldsymbol{\theta})$. By linearity of expectation and linearity of variance (for independent variables), we have

$$\mathbb{E}[\hat{z}] = \frac{1}{\ell}(r_1(\ell)\mathbb{E}[z_1] + r_2(\ell)\mathbb{E}[z_2]) = \frac{1}{\ell}(\mathbf{0} + \mathbf{0}) = \mathbf{0},$$
$$\text{Cov}(\hat{z}) = \frac{1}{\ell^2}(r_1^2(\ell)\text{Cov}(z_1) + r_2^2(\ell)\text{Cov}(z_2)) = \frac{1}{\ell^2}(r_1^2(\ell) + r_2^2(\ell))\Sigma(\boldsymbol{\theta}) = \Sigma(\boldsymbol{\theta}),$$

where the last equality is due to $r_1^2(\ell) + r_2^2(\ell) = \left(\frac{1+\sqrt{2\ell^2-1}}{2}\right)^2 + \left(\frac{1-\sqrt{2\ell^2-1}}{2}\right)^2 = \frac{1+2\ell^2-1}{2} = \ell^2$. □

**Lemma E.2.** *Let $\mathcal{G}_\sigma = (f, \Sigma, \mathcal{Z}_\sigma)$ be a NGOS with scale $\sigma$. Applying the SVAG operator with hyperparameter $\ell \geq 1$, we obtain $\widehat{\mathcal{G}}_{\hat{\sigma}} = (f, \Sigma, \widehat{\mathcal{Z}}_{\hat{\sigma}})$ with scale $\hat{\sigma} = \ell\sigma$. Fixing $\sigma$ and changing $\ell$ produces $\widehat{\mathcal{G}}_{\hat{\sigma}}$ for all scales $\hat{\sigma} \geq \sigma$. If $\mathcal{G}_\sigma$ is well-behaved and satisfies the bounded moments condition, then $\widehat{\mathcal{G}}_{\hat{\sigma}}$ is also well-behaved and satisfies the bounded moments condition. Furthermore, $\widehat{\mathcal{G}}_{\hat{\sigma}}$ satisfies the low-skewness condition.*

*Proof.* The loss function $f$ and covariance function $\Sigma$ are not changed after applying the SVAG operator, so $\widehat{\mathcal{G}}_{\hat{\sigma}}$ is well-behaved.

Now we verify the bounded moments condition. Let $\hat{z} = \frac{1}{\ell}(r_1(\ell)z_1 + r_2(\ell)z_2)$, where $z_1, z_2 \sim \mathcal{Z}_\sigma(\boldsymbol{\theta})$. Then we have

$$\mathbb{E}[\|\hat{z}\|_2^{2m}]^{\frac{1}{2m}} \leq \frac{1}{\ell}\left(|r_1(\ell)|\mathbb{E}[\|z_1\|_2^{2m}]^{\frac{1}{2m}} + |r_2(\ell)|\mathbb{E}[\|z_2\|_2^{2m}]^{\frac{1}{2m}}\right)$$
$$\leq \frac{1}{\ell}(|r_1(\ell)| + |r_2(\ell)|) \cdot \mathbb{E}[\|z_1\|_2^{2m}]^{\frac{1}{2m}}$$
$$\leq (1 + \sqrt{2})\mathbb{E}[\|z_1\|_2^{2m}]^{\frac{1}{2m}}.$$

By the bounded moments condition for $\mathcal{G}_\sigma$, there exists a constant $C_{2m}$ such that $\mathbb{E}_{\boldsymbol{z}\sim\mathcal{Z}_\sigma(\boldsymbol{\theta})}[\|\boldsymbol{z}\|_2^{2m}]^{\frac{1}{2m}} \leq C_{2m}(1+\|\boldsymbol{\theta}\|_2)$ for all $\boldsymbol{\theta}\in\mathbb{R}^d$. So $\mathbb{E}[\|\hat{\boldsymbol{z}}\|_2^{2m}]^{\frac{1}{2m}} \leq (1+\sqrt{2})C_{2m}(1+\|\boldsymbol{\theta}\|_2)$ for all $\boldsymbol{\theta}\in\mathbb{R}^d$.

We now verify the low skewness condition by showing that third moment of $\hat{\boldsymbol{z}}$ is $\mathcal{O}(1/\ell)$. By the bounded moments condition with $m=2$ and Jensen's inequality,

$$\left|\mathbb{E}[\boldsymbol{z}_1^{\otimes 3}]\right| \leq \mathbb{E}[\|\boldsymbol{z}_1\|_2^3] \leq \mathbb{E}[\|\boldsymbol{z}_1\|_2^4]^{3/4} \leq (C_4(1+\|\boldsymbol{\theta}\|_2))^{3/4}.$$

So $\left|\mathbb{E}[\boldsymbol{z}_1^{\otimes 3}]\right|$ is bounded by $\tilde{K}_3(\boldsymbol{\theta}) := (C_4(1+\|\boldsymbol{\theta}\|_2))^{3/4}$ of polynomial growth.

Let $r = \sqrt{2\ell^2-1}$. Then $r_1(\ell) = \frac{1}{2}(1+r), r_2(\ell) = \frac{1}{2}(1-r)$. Since the third moments of two independent random vectors are additive,

$$\begin{aligned}
\mathbb{E}[\hat{\boldsymbol{z}}^{\otimes 3}] &= \frac{1}{8\ell^3}\mathbb{E}\left[((1+r)\boldsymbol{z}_1 + (1-r)\boldsymbol{z}_2)^{\otimes 3}\right] \\
&= \frac{1}{8\ell^3}\left(\mathbb{E}\left[(1+r)^3\boldsymbol{z}_1^{\otimes 3}\right] + \left[(1-r)^3\boldsymbol{z}_2^{\otimes 3}\right]\right) \\
&= \frac{1}{8\ell^3}\left((1+r)^3 + (1-r)^3\right)\mathbb{E}\left[\boldsymbol{z}_1^{\otimes 3}\right] \\
&= \frac{1}{8\ell^3}(2+6r^2)\mathbb{E}[\boldsymbol{z}_1^{\otimes 3}] \\
&= \frac{1}{8\ell^3}(12\ell^2-4)\mathbb{E}[\boldsymbol{z}_1^{\otimes 3}] \leq \frac{1}{\ell}\tilde{K}_3(\boldsymbol{\theta}),
\end{aligned}$$

where the 4th equality is due to $(1+r)^3+(1-r)^3 = (1+3r+3r^2+r^3)+(1-3r+3r^2-r^3) = 2+6r^2$. Therefore, the low skewness condition is verified. $\square$

# F    Miscellaneous Theoretical Arguments

## F.1    How does the noise scale change with batch size?

We discuss how the noise scale $\sigma$ in the NGOS (Definition 2.3) changes when the batch size changes: in particular, $\sigma \sim 1/\sqrt{B}$. The argument follows from the linearity of covariance and is an already well-known result - we reproduce it here for clarity. We first fix a parameter vector $\boldsymbol{\theta}$. Let $\boldsymbol{g}^{(1)}, \ldots, \boldsymbol{g}^{(B)}$ be the gradients evaluated at data points from a batch of size $B$, where $\boldsymbol{g}^{(b)} = \nabla f(\boldsymbol{\theta}) + \boldsymbol{z}^{(b)}$, and every $\boldsymbol{z}^{(b)}$ is a gradient noise vector drawn i.i.d. from a distribution with mean $\boldsymbol{0}$ and covariance $\boldsymbol{\Sigma}(\boldsymbol{\theta})$. For sampling with replacement on a finite dataset of size $n$, where $f_1(\boldsymbol{\theta}), \ldots, f_n(\boldsymbol{\theta})$ are the loss functions for the $n$ data points (and the average of these $n$ functions is $f(\boldsymbol{\theta})$), this covariance matrix can be explicitly written as:

$$\boldsymbol{\Sigma}(\boldsymbol{\theta}) = \frac{1}{n}\sum_{i=1}^n (\nabla f_i(\boldsymbol{\theta}) - \nabla f(\boldsymbol{\theta}))(\nabla f_i(\boldsymbol{\theta}) - \nabla f(\boldsymbol{\theta}))^\top.$$

The average gradient over the batch is $\boldsymbol{g} := \frac{1}{B}\sum_{b=1}^B \boldsymbol{g}^{(b)} = \nabla f(\boldsymbol{\theta}) + \frac{1}{B}\sum_{b=1}^B \boldsymbol{z}^{(b)}$. As $\boldsymbol{z}^{(1)}, \ldots, \boldsymbol{z}^{(B)}$ are sampled i.i.d., their average $\frac{1}{B}\sum_{b=1}^B \boldsymbol{z}^{(b)}$ has mean $\boldsymbol{0}$ and covariance $\boldsymbol{\Sigma}(\boldsymbol{\theta})$ by linearity of expectation and covariance. We can set $\mathcal{Z}_\sigma(\boldsymbol{\theta})$ to be the distribution of the random variable $\frac{1}{\sqrt{B}}\sum_{b=1}^B \boldsymbol{z}_b$, where $\sigma = \frac{1}{\sqrt{B}}$, then $\boldsymbol{g}$ has the same distribution as the stochastic gradient produced by the NGOS $\mathcal{G}_\sigma = (f, \boldsymbol{\Sigma}, \mathcal{Z}_\sigma)$.

## F.2    What happens when the noise does not dominate the gradient?

We discuss the linear warm-up setting described in Section 4.1. Recall that when ignoring the effect of $\epsilon$, the RMSprop update can be written as

$$\boldsymbol{\theta}_{k+1} \approx \boldsymbol{\theta}_k - \eta\boldsymbol{g}_k \odot (\bar{\boldsymbol{g}}^2 + \sigma^2\mathbf{1})^{-1/2}.$$

From the above equation, it is clear that the dynamics of $\boldsymbol{\theta}$ depends on the relationship between the noise scale $\sigma$ and the gradient $\|\bar{\boldsymbol{g}}\|$. In Section 4.1, we discuss the case where $\sigma \gg \|\bar{\boldsymbol{g}}\|$, which is the regime where the SDE approximation can exist.

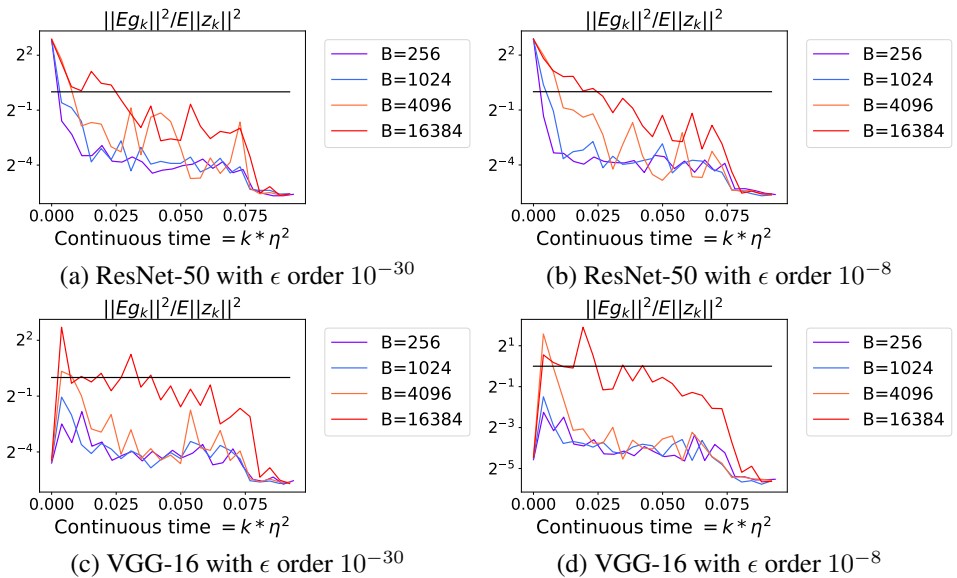

(a) ResNet-50 with $\epsilon$ order $10^{-30}$

(b) ResNet-50 with $\epsilon$ order $10^{-8}$

(c) VGG-16 with $\epsilon$ order $10^{-30}$

(d) VGG-16 with $\epsilon$ order $10^{-8}$

Figure 4: We compare the norm of the average gradient with the noise scale for different batch sizes during training of ResNet-50 and VGG-16 model with RMSprop on the CIFAR-10 dataset. Here, $(\eta, \beta) = (10^{-3}, 0.999)$ for batch size 256 and scaled with our proposed square root scaling rule (Definition 5.1) for the other batch sizes. We show the results for $\epsilon$ at both small (of order $10^{-30}$) and large scale (of order $10^{-8}$). We observe that for small batches, the noise in the gradient dominates the signal in the gradient, supporting our hypothesis. For larger batches, the hypothesis seems to hold true towards the end of training.

Here, we argue that when $\sigma \ll \|\bar{g}\|$, no SDE approximation can exist for the discrete trajectory. In this case, the RMSprop update would instead be $\theta_{k+1} \approx \theta_k - \eta U^{-1} g_k$, where $U = \text{diag}(\sqrt{\bar{g}^2})$. Combining this with $g_k \sim \mathcal{N}(\bar{g}, \sigma I)$ yields that $\theta_{k+1} - \theta_k \sim \mathcal{N}(\eta U^{-1}\bar{g}, \eta^2 \sigma^2 U^{-2})$ approximately. We can again take a telescoping sum to obtain the marginal distribution of $\theta_k$: $\theta_k \sim \mathcal{N}(k\eta U^{-1}\bar{g}, k\eta^2 \sigma^2 U^{-2})$ approximately.

However, it is impossible to make the above distribution fixed even as $\sigma$ changes, so no SDE approximation exists. In particular, we need to make both $k\eta$ and $k\eta^2\sigma^2$ fixed, so $\eta\sigma$ must be a constant. If $\sigma \ll \|\bar{g}\|$, then $\frac{1}{\eta} \ll \|\bar{g}\|$ too. This requirement on $\eta$ implies that no SDE approximation exists when $\sigma \ll \|\bar{g}\|$, and hence motivates us to study the case of $\sigma \gg \|\bar{g}\|$.

# G  Experimental Verification of Assumptions

In this section, we take measurements and perform experiments to verify that the various assumptions made in our theory do not harm the applicability of our findings to realistic settings.

## G.1  Noise Dominates the Gradient

Our analysis in Section 4.1 suggests that an SDE approximation cannot exist when the the gradient $\bar{g}$ dominates the noise scale $\sigma$. Note that Section 4.1 performs a rough analysis under the assumption of a linear loss function (i.e., fixed gradient throughout training), which is far from practice. In the more general setting, we require that for every step $k$, $\mathbb{E}\|z_k\|^2$ (i.e., the gradient variance) dominates $\|\mathbb{E}g_k\|$ (i.e., the norm of the average gradient), where the expectations are taken over sampling seeds, in order for the SDE approximation to exist. Figure 4 shows that our assumption holds for small batches and for large batches near the end of training.

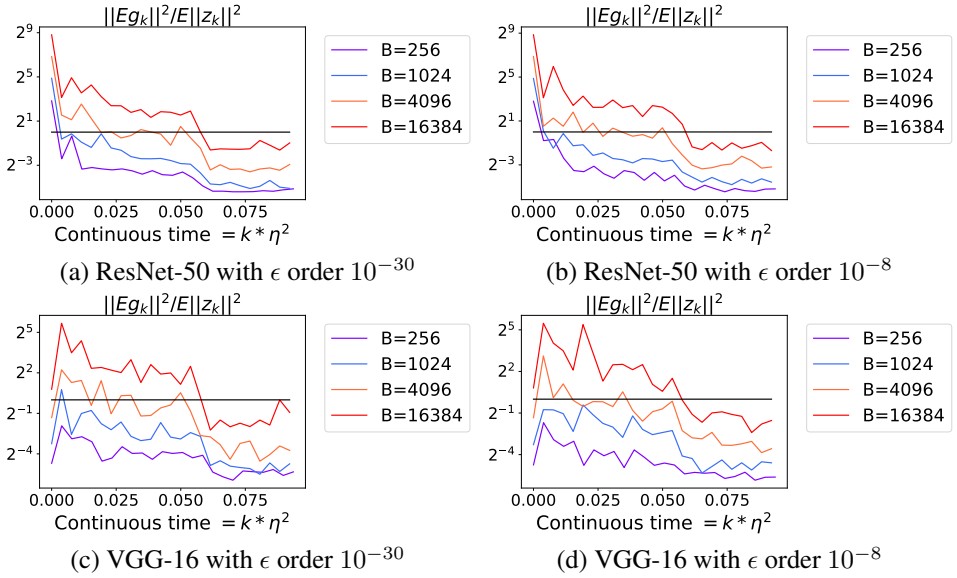

(a) ResNet-50 with $\epsilon$ order $10^{-30}$

(b) ResNet-50 with $\epsilon$ order $10^{-8}$

(c) VGG-16 with $\epsilon$ order $10^{-30}$

(d) VGG-16 with $\epsilon$ order $10^{-8}$

Figure 5: We compare the norm of the average gradient with the noise scale for different batch sizes during training of ResNet-50 model with Adam on the CIFAR-10 dataset. Here, $(\eta, \beta_1, \beta_2) = (10^{-3}, 0.999, 0.999)$ for batch size 256 and scaled with our proposed square root scaling rule (Definition 5.1) for the other batch sizes. We show the results for $\epsilon$ at both small (of order $10^{-30}$) and large scale (of order $10^{-8}$). We observe that for small batches, the noise in the gradient dominates the signal in the gradient, supporting our hypothesis. For larger batches, the hypothesis seems to hold true towards the end of training.

### G.2   Using $v_k$ instead of $v_{k+1}$ in the update rule

In Definitions 2.1 and 2.2, we slightly modify the standard implementation of RMSprop and Adam by using $v_k$ in the update rule instead of $v_{k+1}$. Here, we verify for Adam that this modification of the optimization algorithms does not significantly harm performance. Figure 6 shows the behavior of ResNet-50 and VGG-16 trained with Adam on CIFAR-10 with the above modification of the optimization algorithm. We observe a small drop ($\approx 1\%$) in test accuracies. However, the behavior of the trajectories across different batch sizes for the proposed scaling rule stays the same, i.e. we observe a maximum of $3\%$ test accuracy gap between training batch size 256 and 8192. Moreover, the behavior of the test functions match across the trajectories of different batch sizes.

## H   SVAG Experiments

Recall that the SVAG algorithm (Definition 6.2) is a computationally efficient simulation of the SDEs corresponding to RMSprop and Adam. The SVAG algorithm requires a hyperparameter $\ell$, and the resulting parameters after $k\ell^2$ steps should match the parameters on the corresponding discrete optimization trajectory after $k$ steps. In particular, Theorem 6.3 shows that the SVAG algorithm is an order-1 weak approximation (Definition 2.4) of the SDE, and the approximation error scales as $1/\ell$. One may be initially concerned that realistic deep learning settings require $1/\ell$ to be very small, which would make $\ell$ large and hence computationally intractable. Li et al. (2021) showed that the SVAG trajectories appear to converge to the SDE trajectory for computationally tractable small values of $\ell$. We similarly find that our proposed SVAG-like algorithms in Definition 6.2 appear to converge for small $\ell$ in various settings.

**CIFAR-10.** Figures 7 and 8 show that SVAG converges and closely tracks RMSprop at smaller $\epsilon$ (=$10^{-30}$), and at larger $\epsilon$ (=$10^{-8}$) respectively. Figures 3 and 9 show that SVAG converges and closely tracks Adam at smaller $\epsilon$ (=$10^{-30}$), and at larger $\epsilon$ (=$10^{-8}$) respectively. All experiments follow the setting in Appendix J.2 for batch size 256.

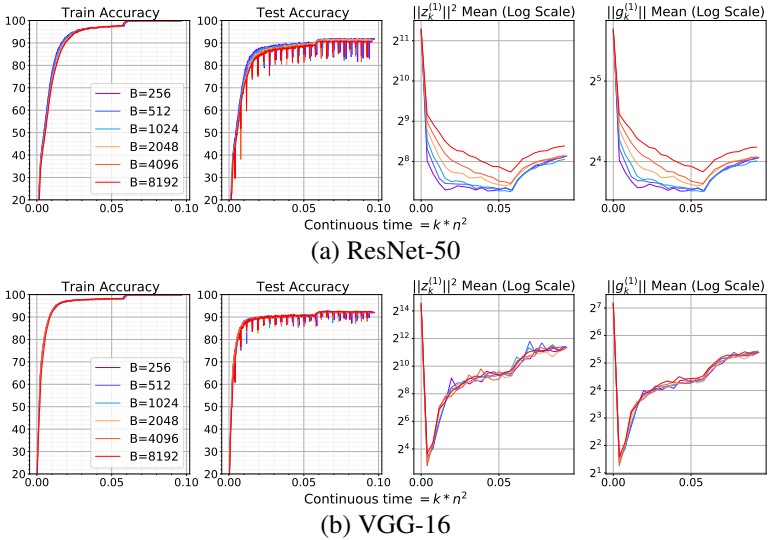

(a) ResNet-50

(b) VGG-16

Figure 6: We repeat the Square Root Scaling experiments with Adam for ResNet-50 and VGG-16 on CIFAR-10 dataset, where we slightly modify the standard implementation of Adam by using $v_k$ in the update rule instead of $v_{k+1}$. For batch size 256, $(\eta, \epsilon, \beta_1, \beta_2) = (10^{-3}, 10^{-8}, 0.999, 0.999)$ and the hyperparameters are scaled according to the square root scaling rule for other batch sizes. We observe a small drop ($\approx 1\%$) in test accuracies. However, the behavior of the trajectories across different batch sizes stays the same.

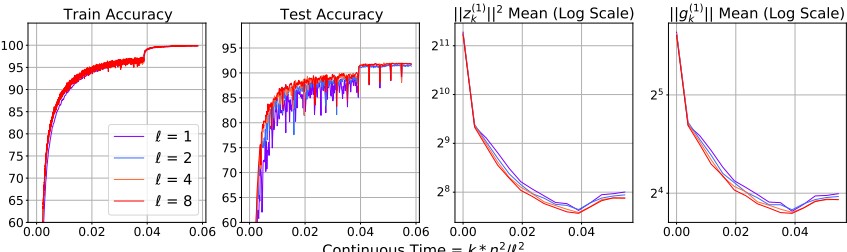

Figure 7: SVAG experiments on ResNet-50 trained on CIFAR-10 with RMSprop. We use batch size 256 and the hyperparameters $\eta = 10^{-3}, \beta = 0.999, \epsilon = 10^{-30}$ and a weight decay of $10^{-4}$. Since SVAG takes $\ell$ smaller steps to simulate the continuous dynamics in $\eta$ time, we plot accuracy against continuous time defined as $k \times \eta^2/\ell^2$.

**Wikipedia + Books (Academic BERT).** Figure 10 shows that SVAG converges and closely tracks Adam. We use the experimental setting for batch size 1024 in Appendix J.4 except the hyperparameters $\beta_1$ and $\beta_2$ are fixed at 0.9 and 0.98 respectively.

**WikiText-103 (GPT).** Figure 23 shows that SVAG converges and closely tracks Adam. We use the experimental setting for batch size 1024 in Appendix J.4 except the hyperparameters $\beta_1$ and $\beta_2$ are fixed at 0.9 and 0.98 respectively. Additionally, for computational reasons, we pretrain on sequences of length 64.

## I  Square Root Scaling Experiments

We experimentally evaluate the scaling rules proposed in Definitions 5.1 and 5.2 by training models with different batch sizes and modifying the optimization hyperparameters accordingly. The number of gradient steps were modified to keep the total amount of continuous time same across all the

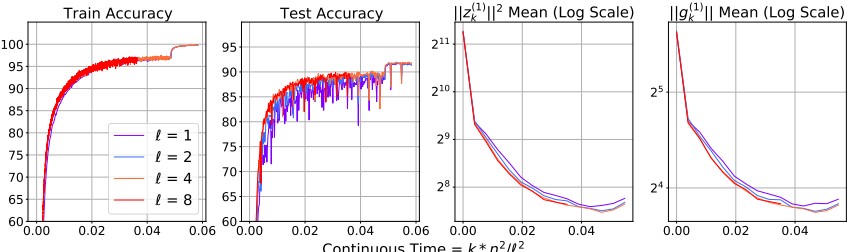

Figure 8: SVAG experiments on ResNet-50 trained on CIFAR-10 with RMSprop. We use batch size 256 and the hyperparameters $\eta = 10^{-3}, \beta = 0.999, \epsilon = 10^{-8}$, and a weight decay of $10^{-4}$. Since SVAG takes $\ell$ smaller steps to simulate the continuous dynamics in $\eta$ time, we plot accuracy against continuous time defined as $k \times \eta^2/\ell^2$.

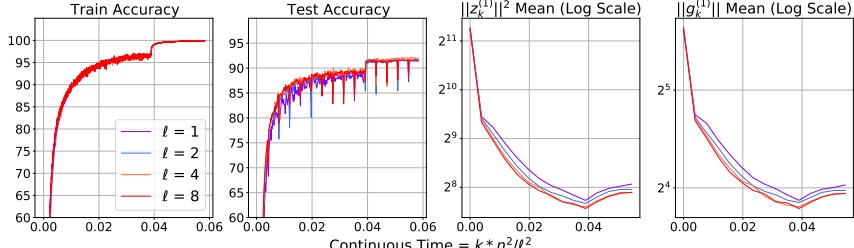

Figure 9: SVAG experiments on ResNet-50 trained on CIFAR-10 with Adam. We use batch size 256 and the hyperparameters $\eta = 10^{-3}, \beta_1 = 0.9, \beta_2 = 0.999, \epsilon = 10^{-30}$, and a weight decay of $10^{-4}$. Since SVAG takes $\ell$ smaller steps to simulate the continuous dynamics in $\eta$ time, we plot accuracy against continuous time defined as $k \times \eta^2/\ell^2$.

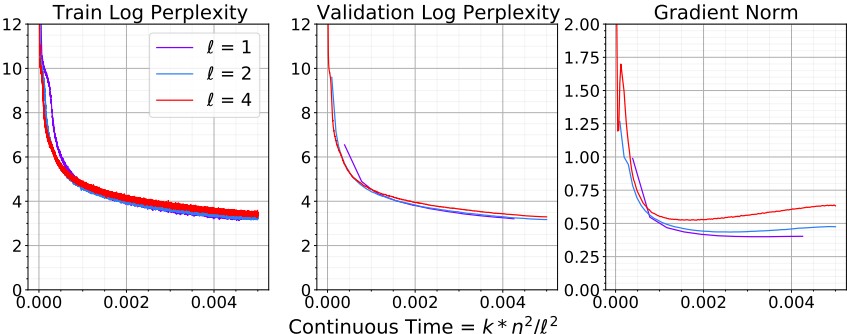

Figure 10: SVAG experiments on RoBERTa pretrained on Bookcorpus + Wikipedia dataset with Adam. We use batch size 1024 and the hyperparameters $\eta = 10^{-3}, \beta_1 = 0.9, \beta_2 = 0.98, \epsilon = 2 \times 10^{-6}$. Since SVAG takes $\ell$ smaller steps to simulate the continuous dynamics in $\eta$ time, we plot accuracy against continuous time defined as $k \times \eta^2/\ell^2$.

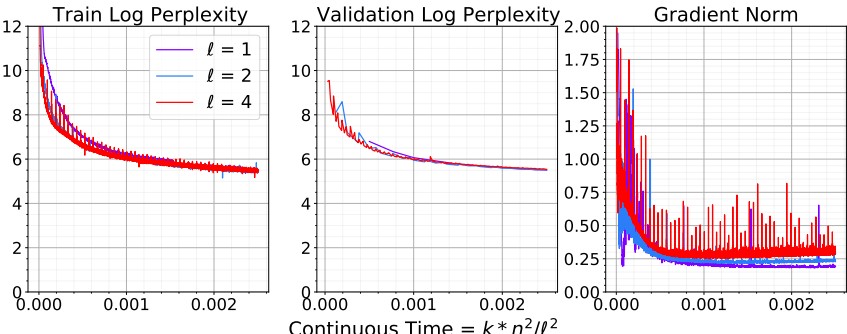

Figure 11: SVAG experiments on GPT pretrained on WikiText-103 with Adam. We use batch size 1024 and the hyperparameters $\eta = 10^{-3}, \beta_1 = 0.9, \beta_2 = 0.98, \epsilon = 2 \times 10^{-8}$. Since SVAG takes $\ell$ smaller steps to simulate the continuous dynamics in $\eta$ time, we plot accuracy against continuous time defined as $k \times \eta^2/\ell^2$. For computational constraints, we train on sequences of length 64.

batches. The warmup schedule and the learning rate decay schedule were kept the same. See Appendix J for details about the baseline runs for each dataset.

We perform two types of ablation studies. In Appendix I.1, we compare our proposed scaling rule to variants that only scale some subset of the optimization hyperparameters. We find that our proposed scaling rule is the best at preserving the validation accuracy and the other test functions across different batch sizes. In Appendix I.2, we compare the proposed square root scaling rule to a linear one and find that they perform comparably on CIFAR-10, though the square root scaling significantly outperforms linear scaling on the ImageNet dataset. We hypothesize that for simpler datasets, like CIFAR-10, the differences between the two scaling rules is not reflected clearly in the validation accuracies because the task is too easy to learn.

**CIFAR-10.** Figures 12 and 14 show the performance of VGG-16 when trained with different batch sizes with RMSprop at smaller $\epsilon$ (=$10^{-30}$), and at larger $\epsilon$ (=$10^{-8}$) respectively. Figures 13 and 15 show the performance of VGG-16 when trained with different batch sizes with Adam at smaller $\epsilon$ (=$10^{-30}$), and at larger $\epsilon$ (=$10^{-8}$) respectively. For the corresponding experiments on ResNet-50, please refer to Figures 16 to 19. For the details on the experimental setting, please refer to Appendix J.2.

We observe that in all settings the test accuracies vary by at most $3\%$ across batch sizes when using the proposed square root scaling rule. Moreover, the test functions stay close across multiple batch sizes, signifying that the trajectories stay close across the batch sizes using the Square root scaling rule.

**ImageNet.** Figures 20 and 21 show the performance of ResNet-50 when trained with different batch sizes with RMSprop at smaller $\epsilon$ (=$10^{-30}$), and at larger $\epsilon$ (=$10^{-8}$) respectively. For the details on the experimental setting, please refer to Appendix J.3.

We observe that on the validation set, the loss and accuracy behavior for the model is very similar, when trained with different batch sizes. Moreover, the difference between the validation accuracies is atmost $3\%$ between the batch sizes 256, 1024, 4096, and 16384 for smaller $\epsilon$. The difference between the validation accuracies is atmost $1.5\%$ between the batch sizes 256, 1024, 4096, 16384, and 32768 for larger $\epsilon$.

**Wikipedia + Books (Academic BERT).** Figure 22 shows the performance of a RoBERTa model when pretrained with different batch sizes with ADAM. The scaling rule is applied to modify the peak values of the optimization hyperparameters. We also evaluate the pretrained models on several downstream tasks, and show the results in Table 1. For the details on the setting for both pretraining and downstream experiments, please refer to Appendix J.4.

| Pretrain batch size $B$ | CoLA | SST-2 | MRPC | STS-B | QQP | MNLI | QNLI | RTE |
|---|---|---|---|---|---|---|---|---|
| 1024 | 0.585 | 0.92 | 0.73 | 0.866 | 0.873 | 0.836 | 0.906 | 0.682 |
| 2048 | 0.563 | 0.928 | 0.803 | 0.869 | 0.875 | 0.826 | 0.897 | 0.653 |
| 4096 | 0.581 | 0.921 | 0.778 | 0.869 | 0.875 | 0.839 | 0.892 | 0.675 |
| 8192 | 0.626 | 0.929 | 0.778 | 0.884 | 0.877 | - | 0.9 | 0.675 |

Table 1: Performance of the pretrained RoBERTa models when finetuned on different downstream tasks in GLUE Wang et al. (2019). F1 scores are reported for QQP and MRPC, Spearman correlations are reported for STS-B, Matthews correlations for CoLA, and accuracy scores are reported for the other tasks. Here, $B$ denotes the batch size used for pretraining. We run an extensive grid search (Table 4) to find the best performance of each pretrained model on each of the downstream tasks.

We observe that the log perplexity on the training and validation datasets matches across different batch sizes throughout pretraining. Moreover, we also observe that the gradient norms across different batch sizes remain close throughout the pretraining. Furthermore, the models pretrained across different batch sizes can achieve very similar performance in the downstream tasks.

**WikiText-103 (Academic GPT).** Figure 23 shows the log perplexity behavior of GPT on training and validation datasets, across different batch sizes of pretraining . We observe that the log perplexity matches across different batch sizes. Moreover, we also observe that the gradient norms across different batch sizes remain close throughout the training.

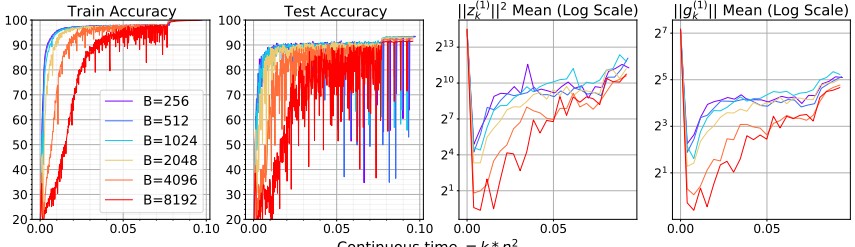

Figure 12: VGG-16 trained on CIFAR-10 using RMSprop are close for different batch sizes when the optimization hyperparameters are varied according to the proposed scaling rule for RMSprop (Definition 5.2). We use a baseline setting of $\eta = 10^{-3}$, $\epsilon = 10^{-8}$, and $\beta = 0.999$ for batch size 256. We use a weight decay factor of $10^{-4}$. We observe a gap of at most 3% among the different batch sizes under consideration.

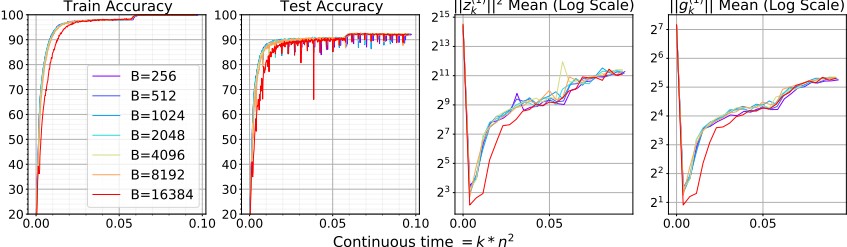

Figure 13: VGG-16 trained on CIFAR-10 using Adam are close for different batch sizes when the optimization hyperparameters are varied according to the proposed scaling rule for Adam (Definition 5.2). We use a baseline setting of $\eta = 10^{-3}$, $\epsilon = 10^{-8}$, and $(\beta_1, \beta_2) = (0.999, 0.999)$ for batch size 256. We use a weight decay factor of $10^{-4}$. We observe a gap of at most 3% among the different batch sizes under consideration.

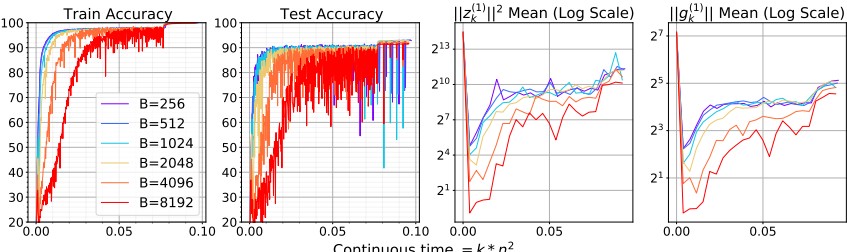

Figure 14: VGG-16 trained on CIFAR-10 using RMSprop are close for different batch sizes when the optimization hyperparameters are varied according to the proposed scaling rule for RMSprop (Definition 5.2). We use a baseline setting of $\eta = 10^{-3}$, $\epsilon = 10^{-30}$, and $\beta = 0.999$ for batch size 256. We use a weight decay factor of $10^{-4}$. We observe a gap of at most $3\%$ among the different batch sizes under consideration.

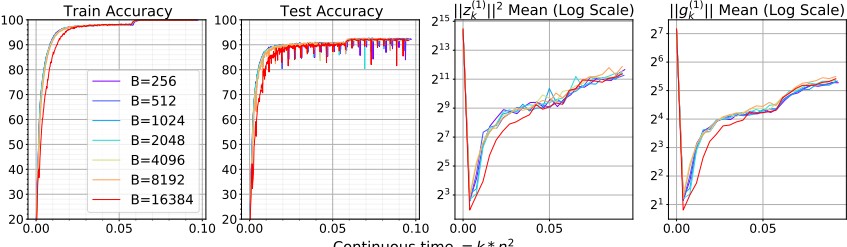

Figure 15: VGG-16 trained on CIFAR-10 using Adam are close for different batch sizes when the optimization hyperparameters are varied according to the proposed scaling rule for Adam (Definition 5.2). We use a baseline setting of $\eta = 10^{-3}$, $\epsilon = 10^{-30}$, and $(\beta_1, \beta_2) = (0.999, 0.999)$ for batch size 256. We use a weight decay factor of $10^{-4}$. We observe a gap of at most $3\%$ among the different batch sizes under consideration.

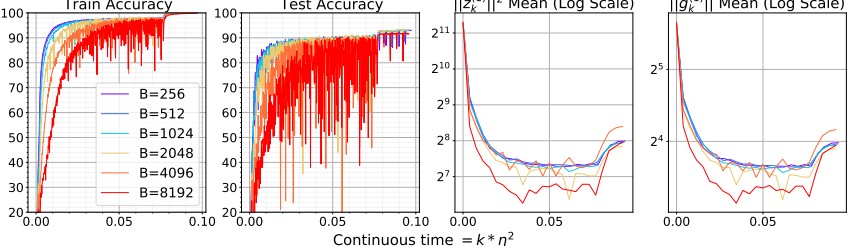

Figure 16: ResNet-50 trained on CIFAR-10 using RMSprop are close for different batch sizes when the optimization hyperparameters are varied according to the proposed scaling rule for RMSprop (Definition 5.2). We use a baseline setting of $\eta = 10^{-3}$, $\epsilon = 10^{-8}$, and $\beta = 0.999$ for batch size 256. $\epsilon = 10^{-30} \approx 0$ for all experiments. We use a weight decay factor of $10^{-4}$. We observe a gap of at most $3\%$ among the different batch sizes under consideration.

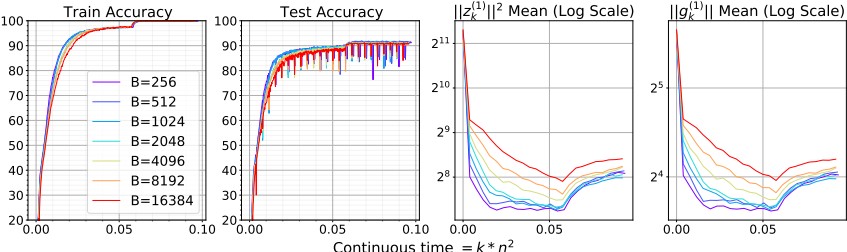

Figure 17: ResNet-50 trained on CIFAR-10 using Adam are close for different batch sizes when the optimization hyperparameters are varied according to the proposed scaling rule for Adam (Definition 5.2). We use a baseline setting of $\eta = 10^{-3}$ and $(\beta_1, \beta_2) = (0.999, 0.999)$ for batch size 256. $\epsilon = 10^{-30} \approx 0$ for all experiments. We use a weight decay factor of $10^{-4}$. We observe a gap of at most $3\%$ among the different batch sizes under consideration.

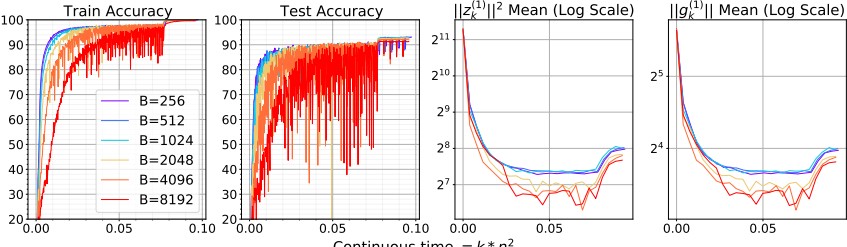

Figure 18: ResNet-50 trained on CIFAR-10 using RMSprop are close for different batch sizes when the optimization hyperparameters are varied according to the proposed scaling rule for RMSprop (Definition 5.2). We use a baseline setting of $\eta = 10^{-3}$, $\epsilon = 10^{-8}$, and $\beta = 0.999$ for batch size 256. We use a weight decay factor of $10^{-4}$. We observe a gap of at most $3\%$ among the different batch sizes under consideration.

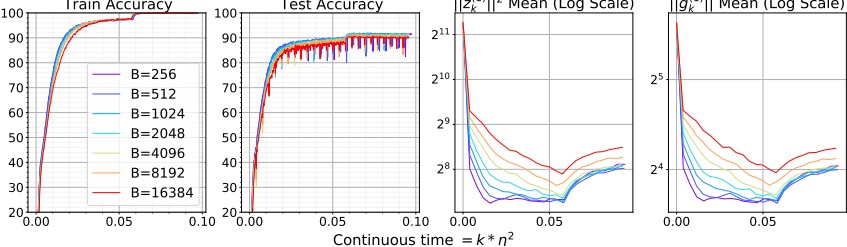

Figure 19: ResNet-50 trained on CIFAR-10 using Adam are close for different batch sizes when the optimization hyperparameters are varied according to the proposed scaling rule for Adam (Definition 5.2). We use a baseline setting of $\eta = 10^{-3}$, $\epsilon = 10^{-8}$, and $(\beta_1, \beta_2) = (0.999, 0.999)$ for batch size 256. We use a weight decay factor of $10^{-4}$. We observe a gap of at most $3\%$ among the different batch sizes under consideration.

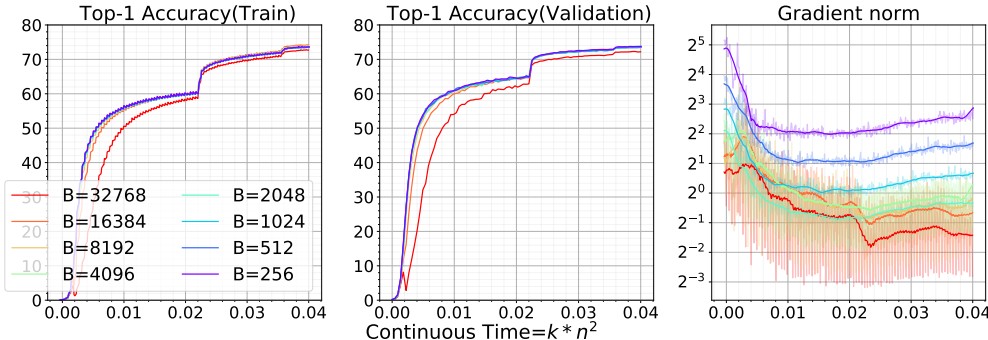

Figure 20: ResNet-50 trained on ImageNet using Adam are close for different batch sizes when the optimization hyperparameters are varied according to the proposed scaling rule for Adam (Definition 5.2). We use a baseline setting of $\eta = 3 \times 10^{-4}$, $\epsilon = 10^{-8}$, and $(\beta_1, \beta_2) = (0.999, 0.999)$ for batch size 256. We use a weight decay factor of $10^{-4}$. We achieve around 74% validation accuracy with batch size 256 and the accuracy drops by at most 1.5% at batch size 32768.

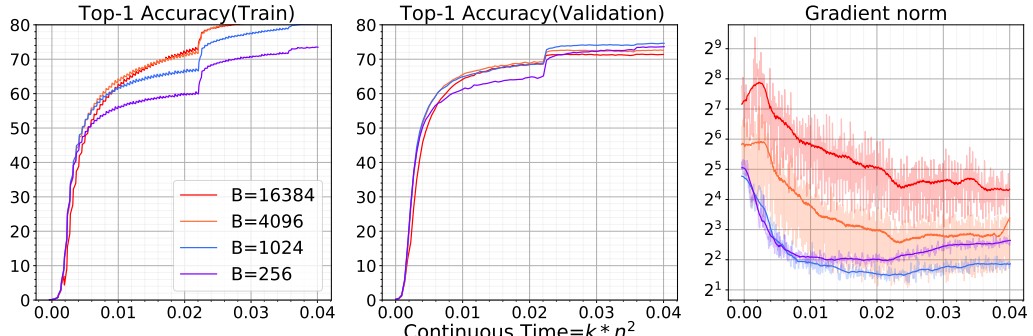

Figure 21: ResNet-50 trained on ImageNet using Adam are close for different batch sizes when the optimization hyperparameters are varied according to the proposed scaling rule for Adam (Definition 5.2). We use a baseline setting of $\eta = 3 \times 10^{-4}$ and $(\beta_1, \beta_2) = (0.999, 0.999)$ for batch size 256. $\epsilon = 10^{-30} \approx 0$ for all experiments. We use a weight decay factor of $10^{-4}$. We achieve around 74% validation accuracy with batch size 1024 and the accuracy drops by at most 3% at batch size 16384.

## I.1 Ablation study on the proposed scaling rule

**CIFAR-10.** We conduct an ablation study on whether all the parameters $\eta, \epsilon, \beta_1, \beta_2$ need to be scaled in our proposed scaling rule. To do so, we compare the performance of a ResNet-50 model trained with batch size 256 and hypeparameters ($\epsilon = 10^{-8}, \beta_1 = 0.999, \beta_2 = 0.999$) with the performance at a larger batch size, across 5 runs representing 5 different scaling rules: (a) Scale $\eta$, keeping others fixed, (b) Scale $\eta, \epsilon$, keeping others fixed, (c) Scale $\eta, \epsilon, \beta_1$, keeping others fixed, (d) Scale $\eta, \epsilon, \beta_2$, keeping others fixed, and (e) Scale $\eta, \epsilon, \beta_1, \beta_2$. Please check the behaviors of the different scaling rules at batch size 2048, 4096, 8192 and 16384 in Appendix I.2. We found that (e) consistently beats others in terms of the test functions and the validation accuracies at all batch sizes. The closest scaling rule (c) involved scaling only $\eta, \epsilon,$ and $\beta_1$ while keeping $\beta_2$ fixed.

## I.2 Ablation against linear scaling rules

**CIFAR-10.** We compared the proposed scaling rule against possible linear scaling rules, that scale the hyperparameters linearly with the increase in training batch size. We focused on ResNet-50 training with Adam. The linear scaling rules that we tried were: (a) Scale $\eta$ linearly, keeping $\beta_1, \beta_2, \epsilon$

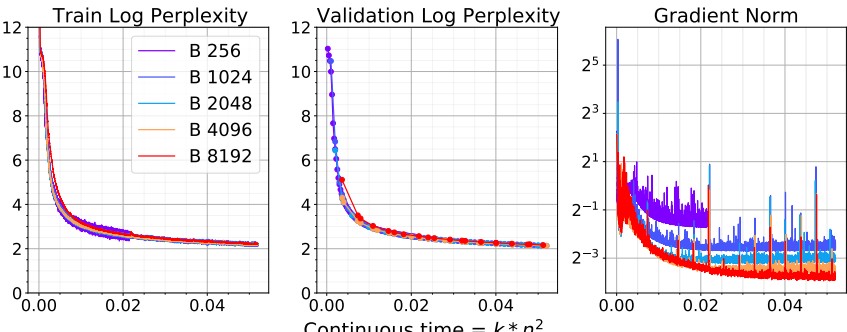

Figure 22: The train and validation log perplexities of RoBERTa-large trained on the Wiki+Books corpus using Adam (for $48$ hours) are close for moderate batch sizes using the Square Root Scaling Rule on Adam. $\eta = 10^{-3}$ and $(\beta_1, \beta_2) = (0.99375, 0.996)$ for batch size $1024$, $\epsilon = 2 \times 10^{-6}$ for batch size $1024$ and scaled likewise for other batch sizes. We achieve a validation log perplexity of $2.1 \pm 0.1$ for batch size $1024$, $2048$, $4096$ and $8192$. Training with batch size $256$ is computationally inefficient, but follows the same behavior during its 48-hour trajectory.

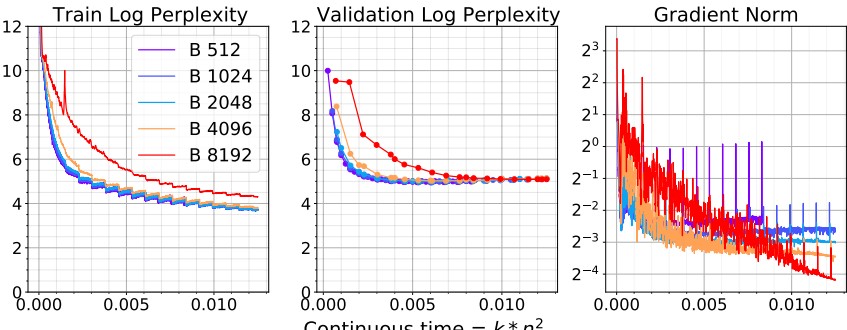

Figure 23: The train and validation perplexities of 12 layer GPT trained on the Wikitext corpus using Adam (for $48$ hours) are close for moderate batch sizes Square Root Scaling Rule on Adam on RoBERTa-large. $\eta = 10^{-3}$ and $(\beta_1, \beta_2) = (0.9875, 0.996)$ for batch size $1024$, $\epsilon = 2 \times 10^{-8}$ for batch size $1024$ and scaled likewise for other batch sizes. We achieve a validation log perplexity of $5 \pm 0.1$ for all the batch sizes under consideration. Moreover, we observe an alignment in the behavior of the trajectories across the different batch sizes (except $8192$ in the first half of the training).

fixed, (b) Scale $\eta, 1 - \beta_1$ linearly, keeping $\beta_2, \epsilon$ fixed, (c) Scale $\eta, 1 - \beta_2$ linearly, keeping $\beta_1, \epsilon$ fixed, and (d) Scale $\eta, 1 - \beta_1, 1 - \beta_2$ linearly, keeping $\epsilon$ fixed. Fig. 25 shows the behavior of these scaling rules at batch size $8192$ and $16384$, at different values of $\epsilon$.

The linear scaling rule in (d) seems to perform as well as the proposed Square Root scaling rule (Definition 5.2) in terms of validation accuracies. However, with a closer look on the train accuracy plots, we observe that the Square Root scaling rule tracks the smaller batch training trajectory better than the linear scaling rule. The linear scaling rules seem to catch up, only after the learning rate is decayed. Our hypothesis is that the CIFAR-10 dataset is simple enough for different scaling rules to work well.

**ImageNet.** We conduct ablation experiments on ResNet-50, trained with Adam on ImageNet, where we follow a linear scaling rule to scale the hyperparameters across batch sizes. Due to computational issues, we didn't conduct extensive experiments, as was done for CIFAR-10. The linear scaling rule for Adam is as follows: the hyperparameters $\eta, 1 - \beta_1$ and $1 - \beta_2$ are scaled by $\kappa$, when the batch size is scaled by $\kappa$, and $\epsilon$ isn't scaled. As was noted earlier, the definition of continuous time will change to $\eta \times \#\{\text{gradient steps}\}$. We keep the number of training epochs equal to $90$ as before, follow the

same learning rate schedule, and show the performance of the models in Figure 26. We observe that scaling the hyperparameters to larger batch training with the proposed LSR results in a big drop in validation accuracies.

## J  Experiment Configuration Details

### J.1  A note on learning rate schedule and warm-up

We have used a learning rate schedule and a warm-up phase in all our experiments. We have to admit that our current theorems do not directly apply to time-varying learning rates or batch sizes. But our experiments demonstrate that our scaling rules continue to hold for learning rate schedulers with a special warm-up, even if they go beyond the scope of our theoretical setting. Technically, the extensions of our theorems to time-varying learning rates or batch sizes are interesting, and we believe they can indeed be shown following the same proof strategy. The corresponding SDE approximations should have hyperparameters changing with time.

### J.2  CIFAR-10

There are $50000$ images in the training set and $10000$ images in the validation set of CIFAR-10 Krizhevsky et al..

**Architecture.**    We used the architecture of ResNet-56 from He et al. (2015a) without modification. We used the same architecture of VGG-16 with batch normalization from Simonyan and Zisserman (2014). However, we kept the final layer of the architecture fixed throughout training, to make the model 1-homogenous and avoid the optimization difficulties of 2-homogenous networks (Li and Arora (2020)).

**RMSprop.**    To fix a baseline to compare against, we first trained the models with batches of size $256$, sampled with replacement, at peak learning rate $\eta = 10^{-3}$ and $\beta = 0.999$. The model was trained for $500 \times \lfloor (50000/256) \rfloor = 97500$ gradient steps (or 500 epochs). We followed an initial warmup for the first $2\%$ of the total gradient steps. The learning rate schedule during the warmup phase is given by $\eta \times 10^{-3} \times (10^3)^{\# \text{ epochs } /10}$. We also followed a learning rate decay schedule, the learning rate was decayed by $0.1$ when the model reaches $80\%$ (400 epoch) and $90\%$ (450 epoch) of the total continuous time respectively. We did experiments at two values of $\epsilon$, small ($= 10^{-30}$) and large ($= 10^{-8}$).

We then made multiple runs of the same model with batches of varying sizes in $\{1024, 4096, 16384\}$, with the hyperparameters $\eta$, $\epsilon$ and $\beta$ modified appropriately according to the scaling rule. The number of gradient steps were modified to keep the total amount of continuous time same across all the batches (which amounted to 500 epochs by the equivalence between continuous time and the number of training epochs). The warmup schedule and the learning rate decay schedule were kept the same.

**Adam.**    We first trained the models with batches of size $256$, sampled with replacement, at peak learning rate $\eta = 10^{-3}$ and $(\beta_1, \beta_2) = (0.999, 0.999)$. The total continuous time of training (or number of epochs), the amount of continuous time in the warmup phase, and the learning rate schedule in the warmup phase were same as RMSprop. The only difference was the learning rate schedule after the warmup phase, the learning rate was decayed by $0.1$ when the model reaches $60\%$ of the total continuous time (or 300 epochs).

We then made multiple runs of the same model with batches of varying sizes in $\{1024, 4096, 16384\}$, with the hyperparameters $\eta$, $\epsilon$ and $\beta_1, \beta_2$ modified appropriately according to the scaling rule. The number of gradient steps were modified to keep the total amount of continuous time same across all the batches. The warmup schedule and the learning rate decay schedule were kept the same.

### J.3  ImageNet

There are $1281167$ images in the training set and $50000$ images in the validation set of ImageNet (Deng et al., 2009).

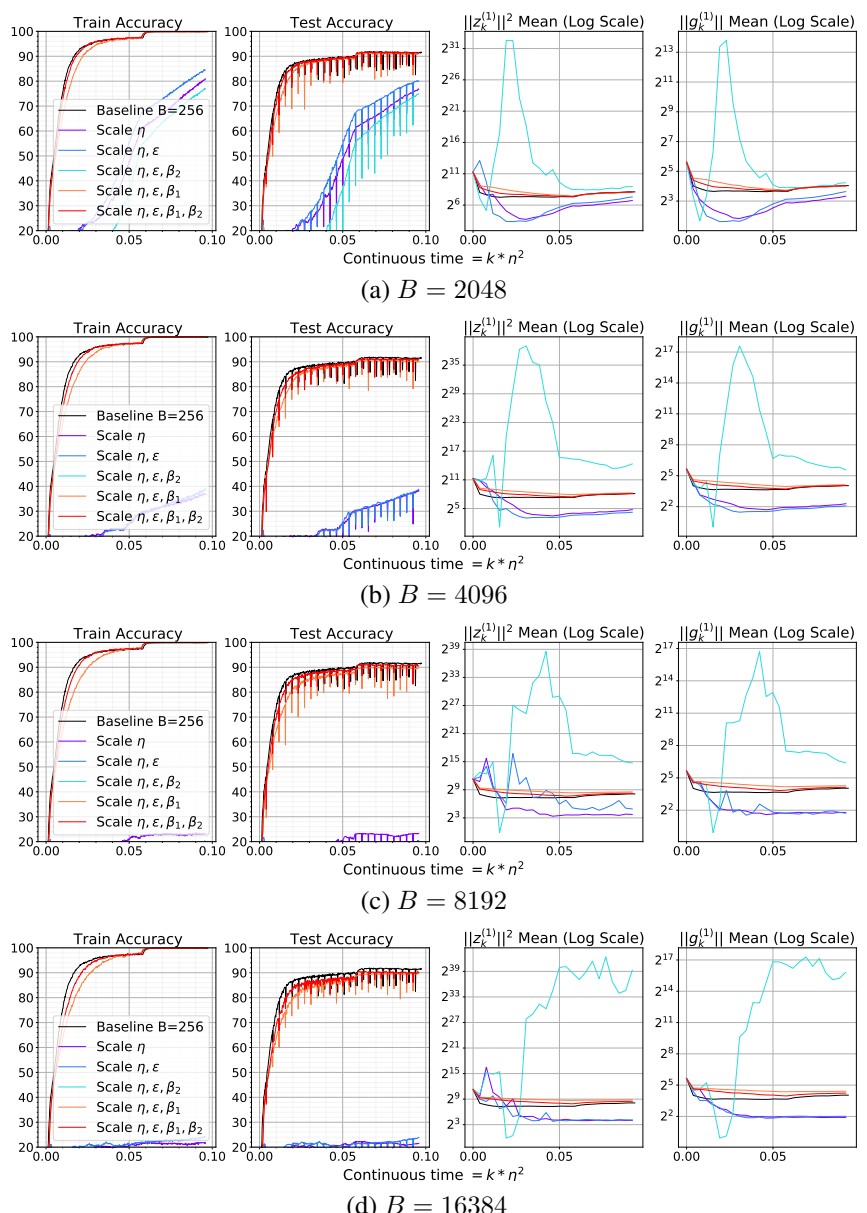

Figure 24: Ablation study for the square root scaling rule on Resnet-50 trained with Adam on CIFAR-10. We compare the performance of a model trained with batch size $256$ and hypeparameters ($\epsilon = 10^{-8}, \beta_1 = 0.999, \beta_2 = 0.999$) with the performance at a larger batch size, across $5$ runs representing $5$ variations of the square root scaling rule: (a) Scale $\eta$, keeping others fixed, (b) Scale $\eta, \epsilon$, keeping others fixed, (c) Scale $\eta, \epsilon, \beta_1$, keeping others fixed, (d) Scale $\eta, \epsilon, \beta_2$, keeping others fixed, and (e) Scale $\eta, \epsilon, \beta_1, \beta_2$. We use a weight decay of $10^{-4}$ in all the experiments. We observe that scaling all the hyperparameters consistently gives better performance at higher batch size. Scaling rule (c) is close second.

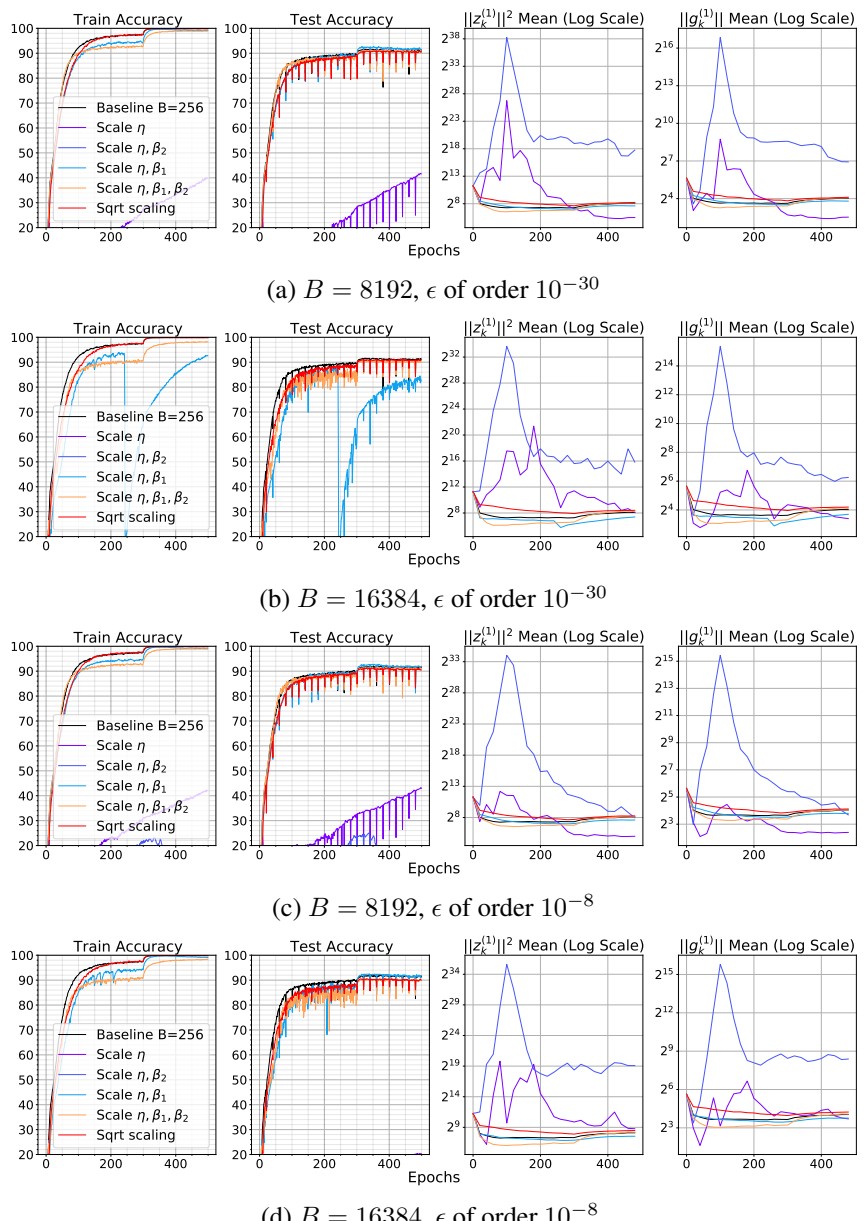

(a) $B = 8192$, $\epsilon$ of order $10^{-30}$

(b) $B = 16384$, $\epsilon$ of order $10^{-30}$

(c) $B = 8192$, $\epsilon$ of order $10^{-8}$

(d) $B = 16384$, $\epsilon$ of order $10^{-8}$

Figure 25: Ablation study against (possible) linear scaling rules on Resnet-50 trained with Adam on CIFAR-10. We compare the performance of a model trained with batch size 256 and hypeparameters ($\epsilon = 10^{-8}/10^{-30}$, $\beta_1 = 0.999$, $\beta_2 = 0.999$) with the performance at a larger batch size, across 5 runs representing 5 possible linear scaling rules: (a) Scale $\eta$ linearly, keeping $\beta_1$, $\beta_2$, $\epsilon$ fixed, (b) Scale $\eta$, $1 - \beta_1$ linearly, keeping $\beta_2$, $\epsilon$ fixed, (c) Scale $\eta$, $1 - \beta_2$ linearly, keeping $\beta_1$, $\epsilon$ fixed, and (d) Scale $\eta$, $1 - \beta_1$, $1 - \beta_2$ linearly, keeping $\epsilon$ fixed. We use a weight decay of $10^{-4}$ in all the experiments. We also compare the behavior of the linear scaling rules against the square root scaling rule. Since the continuous time definition varies across the scaling rules, we plot against the number of epochs trained. A closer look at the training accuracy plots shows that the square root scaling rule tracks the smaller batch training trajectory better than the linear scaling rule.

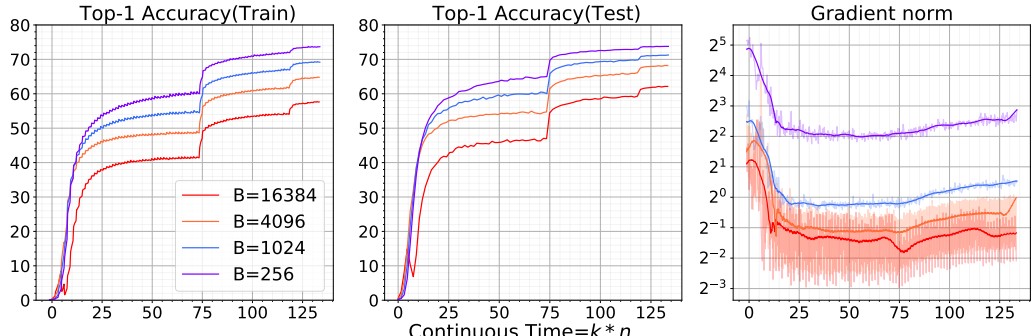

Figure 26: Ablation study against a (possible) linear scaling rule on Resnet-50 trained with Adam on Imagenet. We compare the performance of a model trained with batch size 256 and hypeparameters ($\eta = 3 \times 10^{-4}$, $\epsilon = 10^{-8}$, and $(\beta_1, \beta_2) = (0.999, 0.999)$) with the performance at a larger batch size, when the hyperparameters are scaled as follows: scale $\eta, 1 - \beta_1, 1 - \beta_2$ by $\kappa$, if the batch size is scaled by $\kappa$, keeping $\epsilon$ fixed. We use a weight decay factor of $10^{-4}$. We clearly observe a decrease in performance at larger batch size, in comparison to a model trained with square root scaling rule (see Figure 20).

**Architecture.** We trained a ResNet-50 (He et al., 2015a) model without modification.

**Adam.** To fix a baseline to compare against, we first trained the models with batches of size 256, sampled with replacement, at learning rate $\eta = 3 \times 10^{-4}$ and $(\beta_1, \beta_2) = (0.999, 0.999)$. The model was trained for a total of $90 \times \lfloor (1281167/256) \rfloor = 450360$ gradient steps (or 90 epochs). We followed an initial warmup for the first $\frac{1}{18}$ fraction of the total continuous time (or the first 5 epochs). The learning rate schedule during the warmup phase is increased linearly with epoch, i.e. the learning rate is given by $\eta \times 10^{-3} \times (10^3)^{\# \text{ epochs} /5}$. We then followed a learning rate decay schedule, where the learning rate was decayed by 0.1 when the model reaches $\frac{5}{18}$ fraction (or 50 epoch ) and $\frac{8}{9}$ fraction (or 80 epoch) of the total continuous time respectively. We use two different values for $\epsilon$, small $\epsilon$ ($= 10^{-30} \approx 0$) and a larger $\epsilon$ ($= 10^{-8}$).

### J.4 Books and Wikipedia (Academic BERT)

We use a combination of Bookcorpus (Zhu et al., 2015) plus English Wikipedia, which totals 16 GB of uncompressed text. We split the data uniformly with a ratio 9 : 1, to create training and validation datasets for pretraining.

**Architecture.** We pretrain a 24-layer RoBERTa Liu et al. (2019) model. We pretrain on sequences of length 128.

**Pre-training with Adam.** We use the code from Wettig et al. (2022). We follow the optimization recipe from Izsak et al. (2021) for efficient pre-training. To fix a baseline, we first trained our model with batch size 1024, with the optimization parameters given in table 2. In the warmup phase, the learning rate is increased linearly over the interval, i.e. the learning rate at step $k$ in the warmup phase is given by $\frac{k}{k_{\text{warmup}}}\eta$, where $k_{\text{warmup}}$ denotes the total number of warmup steps and $\eta$ denotes the peak learning rate. Moreover, after the warmup phase, the learning rate is decayed linearly to 0, i.e. the learning rate at step $k$ after the warmup phase is given by $\frac{k - k_{\text{warmup}}}{k_{\text{max}} - k_{\text{warmup}}}\eta$, where $\eta$ denotes the peak learning rate and $k_{\text{max}}$ denotes the maximum number of gradient steps intended for pretraining.

**Fine-tuning with Adam**: We also validate the performance of the pretrained models from the previous section on the GLUE(Wang et al., 2019) datasets. For each downstream task, we run an extensive grid search on the hyperparameters for finetuning each pretrained model. We focused on the following hyperparameters for grid search: batch size, the peak learning rate and the total number of training epochs. Please see Table 4 for the hyperparameter grid. The rest of the hyperparameters

| Hyperparameter | Value |
|---|---|
| Dropout | 0.1 |
| Attention Dropout | 0.1 |
| Warmup Steps | 5520 |
| Peak Learning Rate | $10^{-3}$ |
| Batch Size | 1024 |
| Weight Decay | $10^{-4}$ |
| Max Steps | 92000 |
| Learning Rate Decay | Linear |
| Adam $\beta_1$ | 0.99375 |
| Adam $\beta_2$ | 0.996 |
| Adam $\epsilon$ | $2 \times 10^{-6}$ |
| Gradient Clipping | 0.0 |
| Position embeddings | 128 |

Table 2: Optimization hyperparameters of baseline RoBERTa model during pretraining.

| Hyperparameter | Value |
|---|---|
| Dropout | 0.1 |
| Attention Dropout | 0.1 |
| Warmup Steps | 6% of total |
| Weight Decay | 0.1 |
| Learning Rate Decay | Linear |
| Adam $\beta_1$ | 0.9 |
| Adam $\beta_2$ | 0.98 |
| Adam $\epsilon$ | $10^{-6}$ |
| Gradient Clipping | 0.0 |

Table 3: Optimization hyperparameters of all pretrained RoBERTa models during finetuning.

are fixed for all the runs and are given in Table 3. We follow a similar learning rate schedule during the warmup phase as was used for pretraining: the learning rate at step $k$ in the warmup phase is given by $\frac{k}{k_{\mathrm{warmup}}}\eta$, where $k_{\mathrm{warmup}}$ denotes the total number of warmup steps and $\eta$ denotes the peak learning rate. Moreover, after the warmup phase, the learning rate is decayed linearly to $0$, i.e. the learning rate at step $k$ after the warmup phase is given by $\frac{k-k_{\mathrm{warmup}}}{k_{\max}-k_{\mathrm{warmup}}}\eta$, where $\eta$ denotes the peak learning rate and $k_{\max}$ denotes the maximum number of gradient steps intended for finetuning.

| Dataset | Finetune batch size | Peak Learning rate | Total training epochs |
|---|---|---|---|
| CoLA | $\{16, 32\}$ | $\{10^{-5}, 3 \times 10^{-5}, 5 \times 10^{-5}, 8 \times 10^{-5}\}$ | $\{3, 5, 10\}$ |
| SST-2 | $\{16, 32\}$ | $\{10^{-5}, 3 \times 10^{-5}, 5 \times 10^{-5}, 8 \times 10^{-5}\}$ | $\{3, 5, 10\}$ |
| MRPC | $\{16, 32\}$ | $\{10^{-5}, 3 \times 10^{-5}, 5 \times 10^{-5}, 8 \times 10^{-5}\}$ | $\{3, 5, 10\}$ |
| STS-B | $\{16, 32\}$ | $\{10^{-5}, 3 \times 10^{-5}, 5 \times 10^{-5}, 8 \times 10^{-5}\}$ | $\{3, 5, 10\}$ |
| RTE | $\{16, 32\}$ | $\{10^{-5}, 3 \times 10^{-5}, 5 \times 10^{-5}, 8 \times 10^{-5}\}$ | $\{3, 5, 10\}$ |
| QQP | $\{32\}$ | $\{5 \times 10^{-5}, 8 \times 10^{-5}\}$ | $\{3, 5\}$ |
| MNLI | $\{32\}$ | $\{5 \times 10^{-5}, 8 \times 10^{-5}\}$ | $\{3, 5\}$ |
| QNLI | $\{32\}$ | $\{5 \times 10^{-5}, 8 \times 10^{-5}\}$ | $\{3, 5\}$ |

Table 4: Hyperparameter grid for pretrained RoBERTa$_{\mathrm{LARGE}}$ on the downstream tasks.

| Hyperparameter | Value |
|---|---|
| Dropout | 0.1 |
| Attention Dropout | 0.1 |
| Warmup Steps | 1000 |
| Peak Learning Rate | $10^{-3}$ |
| Batch Size | 1024 |
| Weight Decay | $10^{-4}$ |
| Max Steps | 12500 |
| Learning Rate Decay | Linear |
| Adam $\beta_1$ | 0.9875 |
| Adam $\beta_2$ | 0.996 |
| Adam $\epsilon$ | $2 \times 10^{-8}$ |
| Gradient Clipping | 0.0 |
| Position embeddings | 128 |

Table 5: Optimization hyperparameters of baseline GPT model during pretraining.

## J.5 WikiText-103 (GPT)

WikiText-103 (Merity et al., 2017) is a dataset with 103 million tokens extracted from Wikipedia. We split the data uniformly with a ratio $9 : 1$, to create training and validation datasets for pretraining. We use Adam (Kingma and Ba, 2015) optimization algorithm.

**Architecture.** We pretrain a 12-layer GPT (Brown et al., 2020) model without modification. The model has 12 layers with hidden dimension 768, feedforward network dimension 3072, 12 attention heads in each attention layer, and attention head size 64. We pretrain on sequences of length 128 (unless stated otherwise).

**Adam.** We use the code from Wettig et al. (2022). To fix a baseline, we first trained our model with batch size 1024, with the optimization parameters given in Table 5. In the warmup phase, the learning rate is increased linearly over the interval, i.e. the learning rate at step $k$ in the warmup phase is given by $\frac{k}{k_{\text{warmup}}}\eta$, where $k_{\text{warmup}}$ denotes the total number of warmup steps and $\eta$ denotes the peak learning rate. Moreover, after the warmup phase, the learning rate is decayed linearly to 0, i.e. the learning rate at step $k$ after the warmup phase is given by $\frac{k - k_{\text{warmup}}}{k_{\text{max}} - k_{\text{warmup}}}\eta$, where $\eta$ denotes the peak learning rate and $k_{\text{max}}$ denotes the maximum number of gradient steps intended for pretraining.