# OpenReview forum: "On the SDEs and Scaling Rules for Adaptive Gradient Algorithms"
_NeurIPS.cc/2022/Conference — NeurIPS 2022 Accept_

### Official Review · Reviewer_QQca · 2022-07-09

**Rating:** 6
**Confidence:** 3
**Soundness:** 4 excellent
**Presentation:** 3 good
**Contribution:** 3 good

**Summary:**

The presented paper presents the SDE approximations for Adam and RMSProp optimizers and provides theoretical guarantees of their correctness. The derived SDE approximations revealed the theoretical explanations of the square-root scaling rule of learning rate with batch size increasing. To verify numerically the correctness of the proposed SDEs, authors extend the SVAG technique to the considered classes of SDEs. Numerical experiments consider computer vision and natural language processing models and confirm the correctness of the SDEs and square-root scaling rule.

**Questions:**

Please comment the following remarks
1) what is $\mathcal{G}$ in line 70?
2) please explain why the test functions given in line 319 are satisfied definition 2.4.?
3) how does the presented analysis correlate with results in https://arxiv.org/pdf/1706.02677.pdf%5B3%5D%20ImageNet and https://arxiv.org/pdf/1904.00962.pdf where the special warm up schedules were used?


**Limitations:**

The authors mention limitations in the assumptions before deriving the result SDEs (section 2.3).

**Strengths And Weaknesses:**

Strengths
1) authors present solid theoretical explanation of the empirical rule to manage learning rate with increasing of batch size
2) the expected approximation quality of SDEs is verified in large scale problems from CV and NLP
3) SVAG technique is modified for the considered SDEs to speed up simulations

Weaknesses:
1) the presented analysis assumes the not heavy-tailed noise in the gradient estimates, though some works demonstrate that noise in deep networks is indeed heavy-tailed, see e.g. https://tel.archives-ouvertes.fr/tel-03206456/document and https://arxiv.org/pdf/2106.05958.pdf -  that is left to future work
2) no new practically important consequences of the presented analysis. Researchers use the square root scaling rule previously, now we know why this rule works for adaptive method and why the difference with SGD exists.

---

> ### Author Response · Authors · 2022-08-02
> **Response to Reviewer QQca**
>
> **Why do the test functions given in Line 319 satisfy Definition 2.4?**
> Indeed, the test functions may be of polynomial growth but are not differentiable and thus don’t satisfy Definition 2.4. We studied these test functions because they are more mathematically well-studied during trajectory analysis than the test accuracy. We will update the paper to reflect this nuance.
>
> **How does the presented analysis correlate with previous large-batch training methods with special warm-up schedules?**
> The square root scaling rules in Definitions 5.1 and 5.2 prescribe how to change the optimization hyperparameters when changing the batch size (i.e., across different training runs). Learning rate schedules describe how to modify the LR over the course of a single training run. Therefore, the two are not incompatible; for example, the square root scaling rule can dictate how to scale each phase of the LR schedule when training with two different batch sizes.
>
> **What is $\mathcal{G}$ in Line 70?**
> Sorry for the confusion. $\mathcal{G}$ should be $\mathcal{G}_{\sigma}$, which refers to the noisy gradient oracle. We have updated it in the new version of the paper.

---

> > ### Comment · Reviewer_QQca · 2022-08-09
> > **Learning rate schedule and noise variance**
> >
> > Thank you for answers!
> > The question about large batch setting is related to the presented results since the larger batch size, the smaller variance of the noise in the gradient estimate. So, the question is can your analysis be applied to the variable variance of the gradient and if so, can we formally show the correctness of the LR schedulers for large batch setting? If it can not be applied, please specify the limitations.

---

> > > ### Author Response · Authors · 2022-08-10
> > > **Response to LR schedulers**
> > >
> > > Thanks for clarifying the question!
> > >
> > > **Can your analysis be applied to the variable variance of the gradient (due to the change of batch size), and if so, can we formally show the correctness of the LR schedulers in large-batch training?**
> > >
> > > The LR schedulers in the two papers the reviewer mentioned change LR with time, but not batch size. E.g., in the warm-up stage, the LR gradually increases, but the batch size is staying constant. Therefore, to show the correctness of the LR schedulers, the key is to extend our analysis to time-varying learning rates (i.e., $\eta$ changes with the time). The extension to time-varying batch sizes is not directly related to the second question on the LR schedulers.
> > >
> > > We have to admit that our current theorems do not directly apply to time-varying learning rates or batch sizes. But our experiments demonstrated that our scaling rules continue to hold for LR schedulers with a special warm-up (Appendix J: Lines 1033-1035, 1060-1063), even if they go beyond the scope of our theoretical setting. Technically, the extensions of our theorems to time-varying learning rates or batch sizes are interesting, and we believe they can indeed be shown following the same proof strategy. The corresponding SDE approximations should have hyperparameters changing with time.

---

### Official Review · Reviewer_6FQK · 2022-07-09

**Rating:** 7
**Confidence:** 3
**Soundness:** 4 excellent
**Presentation:** 4 excellent
**Contribution:** 3 good

**Summary:**

This paper derives SDEs corresponding to the dynamics of RMSProp and Adam. Based on the SDEs, the paper proposes to scale the learning rate by $\kappa^{-1/2}$ when the batch size is scaled by $\kappa$. Finally, the paper proposes a SVAG-type algorithm to simulate the training dynamics of the adaptive algorithms and

**Questions:**

1. In addition to the learning rate $eta$, parameteters $beta$, $beta_1$ and $\beta_2$ should also be changed if the batch size is scaled by a positive constant. How can this be interpreted?

2. Does the SVAG algorithm for adaptive algorithms have any practical applications other than validating the training dynamics?

3. It is mentioned in subsection 4.1 and inn the first paragraph of subsection 4.2 that the condition $sigma\gg\|g\|_2$ is crutial in deriving the scaling scheme. How is this condition related to the conditions of Theorem 4.2?

**Limitations:**

The authors properly address this issue in appendix A.2.

**Strengths And Weaknesses:**

Strengths:
1. The paper derives the scaling scheme for the learning rate of adaptive algorithms, which is very impressive.
2. In addition to the SDEs, an approximation bound is given.

Weaknesses:
1. Readers, including myself, may be unfamiliar with the SVAG algorithm if they do not have adequate knowledge of stochastic simulation. More introduction might be required in this part.
2. The SDEs and scaling schemes have been proposed in the literature, though these results have never been presented in such a rigorous way as this paper. The results of this paper might be a little incremental.

---

> ### Author Response · Authors · 2022-08-02
> **Response to Reviewer 6FQK**
>
> **The SDEs and scaling schemes have been proposed before in the literature.**
> We are not aware of this and would love to get a pointer. The square root scaling rules have been proposed through empirical search and through theoretical analysis near convergence, but we note that these earlier rules did not suggest changing the adaptive hyperparameters, which we found to be important (Appendix I.1).
>
> **What is the interpretation of scaling the beta parameters when scaling batch size?**
> In the finite regime, when we change the learning rate, we must scale the adaptivity hyperparameters accordingly to preserve the adaptive nature in the continuous time scale. As an extreme example, we can consider the limiting flows presented in [7]. Here, the authors show that fixing the beta parameters while taking the learning rate to 0 induces a signGD flow. In particular, the adaptivity of the algorithms, a defining characteristic of Adam and RMSprop, can no longer be studied.
>
> **Does the SVAG algorithm for validating the SDE have any other practical applications?**
> In Figure 3 and Appendix H, we see that the test accuracy always increases as we tune the hyperparameter $\ell$ in SVAG to approach the SDE approximation. Similar results were observed in [11]. This suggests that SVAG could be potentially used to closely track the SDE so that we can obtain better generalization in the end. However, this is not immediately practically useful because the computational expense of SVAG scales with $\ell^2$. It may be possible to circumvent or reduce this cost in the future.
>
> [11] Zhiyuan Li, Sadhika Malladi, and Sanjeev Arora. On the validity of modeling SGD with stochastic differential equations (SDEs). arXiv preprint arXiv:2102.12470, 2021.
>
> **How is the condition $\sigma \gg \|g\|_2$ related to the conditions of Theorem 4.2?**
> In the definition of the SDE for RMSprop (Definition 4.1), we set $\sigma_0 \triangleq \sigma \eta$ as a constant, and our final bound depends on this constant. To keep $c_0$ constant when $\eta \to 0$, $\sigma$ has to be scaled as $\sigma \sim 1 / \eta$ and thus $\sigma \gg \|g\|_2$. We will add this explanation in the final version.
>
> **More introduction on SVAG might be required.**
> We will add an additional section in the appendix discussing approximation schemes and more intuitions on how SVAG works.

---

### Official Review · Reviewer_AnTC · 2022-07-10

**Rating:** 8
**Confidence:** 3
**Soundness:** 3 good
**Presentation:** 3 good
**Contribution:** 4 excellent

**Summary:**

This paper attains the SDE approximations for two popular optimization algorithms RMSprop and Adam, and validates their applicability empirically. The SDE approximations generate square-root scaling rules, which are validated in the vision and language modeling domains.

**Questions:**

1.  [1] proposes an exponential learning rate schedule. How is the square root schedule proposed in the paper compared to the exponential schedule?
2. [2] employs Levy SDE approximation. Can the authors explain why "the=e quality of the Lévy SDE approximation was not formally guaranteed" and the advantage of using Ito SDE in the paper?

References

[1] Zhiyuan Li and Sanjeev Arora. An exponential learning rate schedule for deep learning. arXiv preprint arXiv:1910.07454, 2019.
[2] Pan Zhou, Jiashi Feng, Chao Ma, Caiming Xiong, Steven Chu Hong Hoi, Weinan E. Towards Theoretically Understanding Why Sgd Generalizes Better Than Adam in Deep Learning. NeurIPS 2020.

**Limitations:**

Yes.

**Strengths And Weaknesses:**

Strengths:
**1)** The paper is generally well-written and the presentation is clear.
**2)** The theoretical results are new and interesting. The empirical results are provided to validate the derived theories.
**3)** I did not check every detail of the proof but generally it is technically sound. The assumptions are also supported by empirical results.

Weaknesses
I did not find any obvious weakness of the paper.

---

> ### Author Response · Authors · 2022-08-02
> **Response to Reviewer AnTC**
>
> **How does the square root schedule compare to the exponential schedule?**
> The square root scaling rules in Definitions 5.1 and 5.2 prescribe how to change the optimization hyperparameters when changing the batch size (i.e., across different training runs). The exponential learning rate schedule describes how to modify the LR over the course of a single training run. Therefore, the two are not incompatible; for example, the square root scaling rule can dictate how to scale each phase of the exponential LR schedule when training with two different batch sizes.

---

### Official Review · Reviewer_fuyA · 2022-07-12

**Rating:** 5
**Confidence:** 3
**Soundness:** 4 excellent
**Presentation:** 3 good
**Contribution:** 3 good

**Summary:**

This paper derives SDE approximations of RMSProp and Adam when their hyperparameters changes according to a scaling rule. Experiments on VGG and ResNets were performed on CIFAR-10.

**Questions:**

Please see the "Strengths and Weaknesses" section.

**Limitations:**

As there are existing work studying similar problems, especially the Lévy process based SDEs, and some existing work suggests the gradient noises are heavy-tailed. It might be instructive to study whether SDEs in this paper makes more sense than existing work, and to compare and contrast their use cases in practice.

**Strengths And Weaknesses:**


Pros:
- The scaling rules in Theorem 5.3 is interesting.
- The SDE approximations of RMSProp and Adam are derived, and they appear to be correct.

Cons:

- Experiment results (e.g., Fig 1-3) do not have confidence bands. Furthermore, in Figure 1 (a) and (b) all curves are very close but they are quite far away in (c) and (d). Does this mean the scaling rule is more respected by Adam but not RMSProp? I found it a bit hard to interpret the results. Perhaps the authors can add some interpretations in the figure caption and in the main texts.
- Why in Figure 2(b) the curves do not start at the same point?
- The proof technique seems not new, the theorems look more like extensions to the stochastic modified equation literature. Perhaps I missed something. Can the authors discuss more in the main text how the new analyses differ in technique from those literatures?
- As some literature suggested, the gradient noise might be heavy-tailed, it is thus open to discuss which of the model (Gaussian vs. Lévy) makes more sense in practice — when would Gaussian noises occur in training practical neural nets?

---

> ### Author Response · Authors · 2022-08-02
> **Response to Reviewer fuyA**
>
> **How is your proof technique different from previous works?**
> The proof technique broadly follows the framework in [11, 15]: bound the difference between the SDE and the discrete trajectory for a single step and extend this to bound the error on a finite time interval. However, we now list some substantive improvements to existing ideas, which could be impactful because they extend existing techniques to adaptive gradient algorithms:
> 1. The SDEs we derive do not satisfy the Lipschitzness and smoothness conditions because the denominator can be *unbounded*. We thus derive well-behaved auxiliary SDEs (Theorems C.3 and D.2) that approximate RMSprop and Adam and show that the presented SDEs induce the same distribution of trajectories as the auxiliary SDEs.
> 2. Adam can induce a noisy and discontinuous trajectory in the initial steps because of the normalization constants, so we construct an SDE approximation that can track the dynamics after a few initial steps (see lines 233-236).
> 3. The drift and diffusion functions (see eq (5) in Appendix B) in the SDEs we propose are *time-dependent*, unlike the functions for SGD, so we needed to adapt the results from [11] (Lemmas B.4-B.6) to make them time-dependent.
>
> [11] Zhiyuan Li, Sadhika Malladi, and Sanjeev Arora. On the validity of modeling SGD with stochastic differential equations (SDEs). arXiv preprint arXiv:2102.12470, 2021.
>
> [15] Pan Zhou, Jiashi Feng, Chao Ma, Caiming Xiong, Steven C. H. Hoi, and Weinan E. Towards theoretically understanding why SGD generalizes better than ADAM in deep learning. CoRR, abs/2010.05627, 2020.
>
> **Do your experiments show that the scaling rule holds better for Adam than for RMSProp?**
> Yes. We have also repeated the experiments for Figure 1 with more random seeds. The figure shows that the trajectory of RMSprop exhibits a larger variance than Adam, which is likely why the scaling rule seems to hold better for Adam. We will smooth the trajectory and rerun more settings with additional seeds for the next version of the paper. We will also expand the discussion of the results.
>
> **Experiments do not have confidence bounds.**
> We have added confidence bounds for Figure 1. We did not have time to repeat the experiments for the other figures within the rebuttal period, but we will add them in the final version.
>
> **Why don't the curves in Figure 2(b) start at the same point?**
> The curves are plotted in continuous time, and we do not measure their performance at initialization. We measure the test error every 1000 updates, which corresponds to different continuous times based on the learning rate.

---

> > ### Comment · Reviewer_fuyA · 2022-08-08
> > **Thank you for the response**
> >
> > Thank you for answering my questions. It might be better to emphasize the main differences (Q1) in the main texts. I am not against accepting this work.

---

### Author Response · Authors · 2022-08-02
**Heavy-Tailed Noise**

We thank the reviewers for their helpful feedback. Here, we discuss the issue of heavy-tailed noise which was raised by several reviewers, and we will include a condensed version of this in the next version of the paper. In order to model the stochastic noise with an Ito process, we assume the gradient noise is slightly non-Gaussian but not heavy-tailed (Definitions 2.5-2.7), as is standard in Ito SDE literature [11, 15]. In lines 144-151, we briefly discuss the ongoing but currently inconclusive research on whether the real-world gradient noise is heavy-tailed or not. We later note that [15] considered a Levy SDE approximation for Adam but did not formally justify the approximation.

We start by noting that in the small (but finite) LR regime, corresponding to most realistic deep learning settings, we only require the gradient noise to have a bounded covariance. We now summarize various experiments that have shed light on the nature of gradient noise and their interpretations:

**Reviewer fuyA: “when would Gaussian noises occur in training practical neural nets?”** **Reviewer QQca: Links to experimental papers suggesting gradient noise is heavy-tailed**

Note that our assumptions (Definitions 2.6 and 2.7) allow some non-Gaussianity. Most empirical evidence about heavy-tailed noise relies on a tail-index estimator that’s shown to be faulty in [11, 16] (see lines 144-151 in our submission). Other evidence (e.g., Reviewer QQca’s second linked paper) fits Gaussian density functions to the gradient noise and visually inspects the quality of fit to determine the distribution cannot be modeled as Gaussian (Figure 1 in their paper), but it is unclear if the tails are heavy enough to violate our assumptions. Our SVAG simulation provably converges to the Ito SDE so long as the noise covariance is bounded. Hence the success of the SVAG simulation (Section 6 and Appendix H) suggests that modeling the noise with a Wiener process is sufficient for the SDE approximation to capture the properties of training that are most interesting to machine learning practitioners (e.g., final train and test accuracy) and theorists (e.g., gradient norm and other mathematically well-behaved test functions).

**Reviewer AnTC: What is the advantage of using Ito SDE (instead of the Levy SDE in [15])?**

The Levy SDE derived in [15] does not have formal approximation guarantees as ours do (Definition 2.4, Theorems 4.2 and 4.4). That work focuses primarily on a theoretical understanding of the escaping behaviors of SGD and Adam, so there is minimal empirical interest in characterizing the gradient noise as heavy-tailed or Gaussian-like. Note that there is no tractable simulation of Levy SDEs available, so we cannot directly test their applicability to deep learning as was done for Ito SDEs in Section 6 and Appendix H.

We remain interested in exploring the heavy-tailed analog for our approximations, but efficient simulation of such SDEs remains intractable and formal approximation guarantees are difficult to prove, limiting our ability to assess the utility of such approximations for practitioners and theorists.

[11] Zhiyuan Li, Sadhika Malladi, and Sanjeev Arora. On the validity of modeling SGD with stochastic differential equations (SDEs). arXiv preprint arXiv:2102.12470, 2021.

[15] Pan Zhou, Jiashi Feng, Chao Ma, Caiming Xiong, Steven C. H. Hoi, and Weinan E. Towards theoretically understanding why SGD generalizes better than ADAM in deep learning. CoRR, abs/2010.05627, 2020.

---

### Meta-Review · Area_Chair_fEZJ · 2022-08-23

**Recommendation:** Accept
**Confidence:** Certain

**Metareview:**

The paper attains the SDE approximations for two optimization algorithms RMSProp and Adam. The authors have addressed the concerns raised by the reviewers during the rebuttal period. All the reviewers agreed that the paper should be accepted at NeurIPS 2022. Please incorporate the reviewers’ suggestions in their detailed reviews and revise the final version of the paper properly.

**Award:**

No

---

### Decision · Program_Chairs · 2022-09-14

Accept